# TIME: A Multi-level Benchmark for Temporal Reasoning of LLMs in Real-World Scenarios

**Shaohang Wei[1], Wei Li[1], Feifan Song[1], Wen Luo[1]**
**Tianyi Zhuang[2], Haochen Tan[2], Zhijiang Guo[2], Houfeng Wang[1]**
[1]MoE Key Lab of Computational Linguistics,
School of Computer Science, Peking University
[2]Huawei Noah's Ark Lab
shaohang@stu.pku.edu.cn   wanghf@pku.edu.cn
{zhuangtianyi, haochen.tan}@huawei.com cartusguo@gmail.com

## Abstract

Temporal reasoning is pivotal for Large Language Models (LLMs) to comprehend the real world. However, existing works neglect the real-world challenges for temporal reasoning: (1) intensive temporal information, (2) fast-changing event dynamics, and (3) complex temporal dependencies in social interactions. To bridge this gap, we propose a multi-level benchmark TIME, designed for temporal reasoning in real-world scenarios. TIME consists of 38,522 QA pairs, covering 3 levels with 11 fine-grained sub-tasks. This benchmark encompasses 3 sub-datasets reflecting different real-world challenges: TIME-WIKI, TIME-NEWS, and TIME-DIAL. We conduct extensive experiments on reasoning models and non-reasoning models. And we conducted an in-depth analysis of temporal reasoning performance across diverse real-world scenarios and tasks, and summarized the impact of test-time scaling on temporal reasoning capabilities. Additionally, we release TIME-LITE, a human-annotated subset to foster future research and standardized evaluation in temporal reasoning.

|   |   |   |
|---|---|---|
| ⯃ | Github Repo | [GitHub Page] |
| 🤗 | TIME | [Huggingface Dataset] |
| 🤗 | TIME-Lite | [Huggingface Dataset] |
| ⧗ | Project Page | [Project Page & Leaderboard] |

## 1 Introduction

Time serves as the thread that weaves together complex events in the real world. Effective temporal reasoning is crucial for Large Language Models (LLMs) to process and comprehend complex events with human-like understanding, particularly in applications requiring integration of historical data and real-time progress tracking. Despite good capabilities of current LLMs across a wide range of reasoning tasks [26], including mathematical problem-solving [11, 14, 29] and code generation [22, 19, 15, 16, 25], they still face challenges in managing temporal understanding in reality.

Temporal reasoning in real-world contexts presents complex challenges: (1) the density of temporal information embedded within world knowledge, (2) the rapid evolution of event details over time, and (3) the complexity of temporal dependencies in social interactions, but existed benchmarks, like TimeBench[6] and TRAM[47] primarily focus onsimplified scenarios, such as basic temporal

39th Conference on Neural Information Processing Systems (NeurIPS 2025) Track on Datasets and Benchmarks.

commonsense and relationships within short texts and simple QA tasks. Consequently, a significant gap remains in exploring temporal reasoning in depth.

On the other hand, temporal reasoning constitutes a hierarchical framework of fine-grained abilities, which is also different from other reasoning tasks that focus on singular capabilities, but is still ignored by current works. For example, TReMu[12] involves only neuro-symbolic temporal reasoning while neglecting temporal computation, and TCELongBench[58] overlooks fundamental temporal concept understanding. In contrast, a robust evaluation framework should encompass both basic temporal abilities and complex event-event temporal reasoning, necessitating the development of a new comprehensive benchmark.

To address these limitations, we introduce **TIME**, a multi-level comprehensive benchmark for evaluating temporal reasoning in LLMs across diverse real-world scenarios, comprising 38,522 instances. TIME consists of three datasets: TIME-WIKI assesses temporal reasoning in knowledge-intensive scenarios, TIME-NEWS evaluates temporal understanding in rapidly evolving news contexts, and TIME-DIAL examines temporal reasoning in complex interactive settings with extensive temporal dependencies in very-long dialogs. Additionally, we construct TIME-LITE, a high-quality lightweight benchmark containing 938 carefully curated instances through manual annotation and verification, enabling efficient and reliable temporal reasoning evaluation. Both TIME and TIME-LITE feature a multi-level task structure: (1) basic temporal understanding and retrieval, (2) temporal expression reasoning, and (3) complex temporal relationship reasoning, with each level incorporating multiple fine-grained dimensions to comprehensively assess temporal reasoning capabilities.

Our main contributions can be summarized as follows:

- We introduce an innovative multi-level evaluation framework that systematically assesses temporal reasoning capabilities across different granularities in LLMs.
- We construct TIME, a comprehensive benchmark that captures the complexity of temporal reasoning in diverse real-world scenarios, including knowledge-intensive, dynamic events, and multi-session interactive contexts.
- We conduct a comprehensive evaluation and in-depth analysis on temporal reasoning for a wide range of LLMs.

## 2 Related Work

**Temporal Understanding in Natural Language** Temporal understanding in Natural Language Processing (NLP) has a rich history, initially focusing on extracting time expressions and temporal relationships [33, 42, 43, 44, 41, 28, 32, 36]. The advent of pre-trained language models facilitated the exploration of more complex phenomena, including implicit time reasoning [21, 38, 39] and cross-event temporal relationships [60, 48, 50, 18, 38, 39, 37]. Research has also addressed commonsense temporal knowledge [45, 59] and enhanced event-based temporal question-answering through knowledge graphs [51] or specialized language model training [35, 5, 55, 13]. Unlike these prior efforts that often target specific temporal aspects, TIME proposes a unified framework for a comprehensive evaluation of temporal understanding spanning time and events.

**Temporal Reasoning in Real-World Scenarios** Advancements in language models have facilitated deeper exploration of temporal reasoning, particularly concerning event ordering and causality. Benchmarks like TimeQA [4] assess time-sensitive question answering, while RealTimeQA [23], FreshLLM [46], and StreamingQA [27] address adaptation to dynamic knowledge. SituatedQA [56] and SituatedGen [57] evaluate the integration of temporal and geographical commonsense into text understanding. Domain-specific studies further explore temporal reasoning. In news, TCELongBench [58] probes temporal understanding of complex events, focusing on details, ordering, and prediction. For conversations, TimeDial [34] examines everyday temporal commonsense, and TReMu [12] tackles temporal localization and long-range dependencies in long-form dialogues. Unlike TCELongBench and TReMu, which concentrate on specific temporal aspects, TIME offers a comprehensive evaluation. It employs three progressive complexity levels and diverse subtasks for fine-grained analysis of temporal reasoning in extended contexts.

**Benchmarks for Temporal Reasoning** Existing studies evaluate LLM temporal reasoning using various benchmarks like TRAM [47], which assesses event sequence understanding but lacks challenge for current models. Other efforts include a benchmark using six existing datasets [20] and

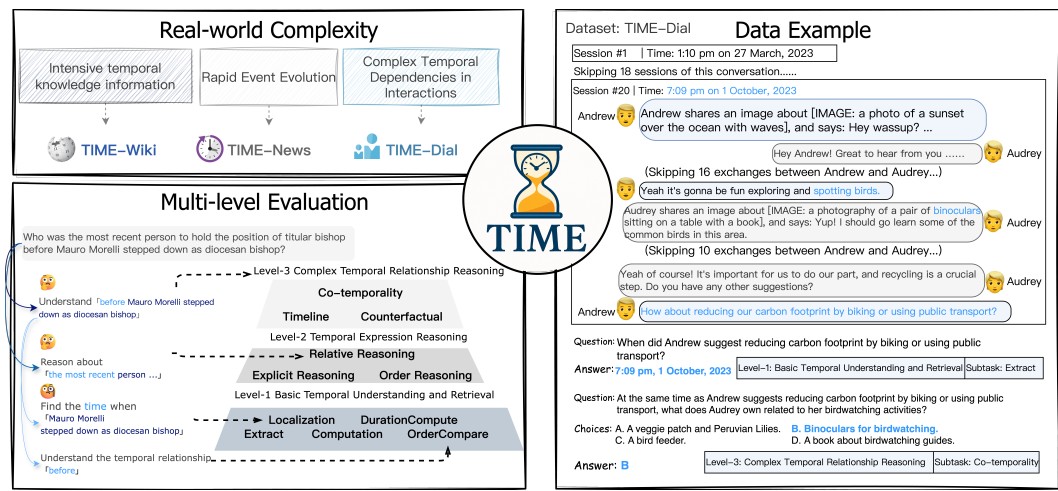

Figure 1: An overview of TIME. The top-left block illustrates three key challenges of real-world complexity and their corresponding dataset construction. The bottom-left quadrant depicts a three-level tasks. One data example from TIME-DIAL is shown on the right.

TimeBench [6], which aggregates 10 datasets across symbolic, common-sense, and event-temporal levels. However, these benchmarks have limitations: TimeBench's simpler tasks are less challenging for improving LLMs, and the diverse evaluation contexts and difficulty levels across datasets can introduce biases, hindering fair and consistent assessment. In contrast, TIME provides a unified evaluation context, enabling fair, fine-grained, and challenging assessment of LLM temporal reasoning in real-world scenarios.

## 3   TIME: Benchmark Construction

This section details the construction of TIME, covering task definition and design principles (§3.1), data source selection (§3.2), and dataset construction with quality control (§3.3). Finally, we introduce TIME-LITE, a high-quality, manually verified sub-dataset (§3.4).

### 3.1   Task Definition

TIME is designed for a fine-grained and comprehensive exploration of real-world temporal reasoning challenges. We structure these challenges into three progressive levels and propose various task formats that better capture the intrinsic nature of temporal reasoning in real-world scenarios.

#### 3.1.1   Design Principal

Real-world text information contains complex content, among which temporal information is a crucial clue. TIME aims to simulate the process by which humans utilize temporal concepts to better understand the complex and dynamic world information, and to measure the ability of LLMs to solve real-world problems using time. Humans tend to first accurately capture and understand temporal concepts (Level-1: Basic Temporal Understanding and Retrieval); secondly, it requires integrating contextual information to reason about implicit and ambiguous temporal expressions to locate event details (Level-2: Temporal Expression Reasoning); finally, the complex temporal relationships embedded in events are crucial for clarifying timelines and potential event causality (Level-3: Complex Temporal Relationship Reasoning). These three types of challenges are progressive and interconnected. The temporal reasoning tasks at three levels are detailed below:

**Level-1: Basic Temporal Understanding and Retrieval.** Level-1 requires models to establish fundamental temporal information processing capabilities. We design five subtasks organized through three complementary aspects: (1) `Extract` assesses direct retrieval of temporal expressions (time points, periods, relative time) from text, paired with (2) `Localization` that evaluates event-time mapping accuracy through temporal positioning of occurrences - together forming the basis of

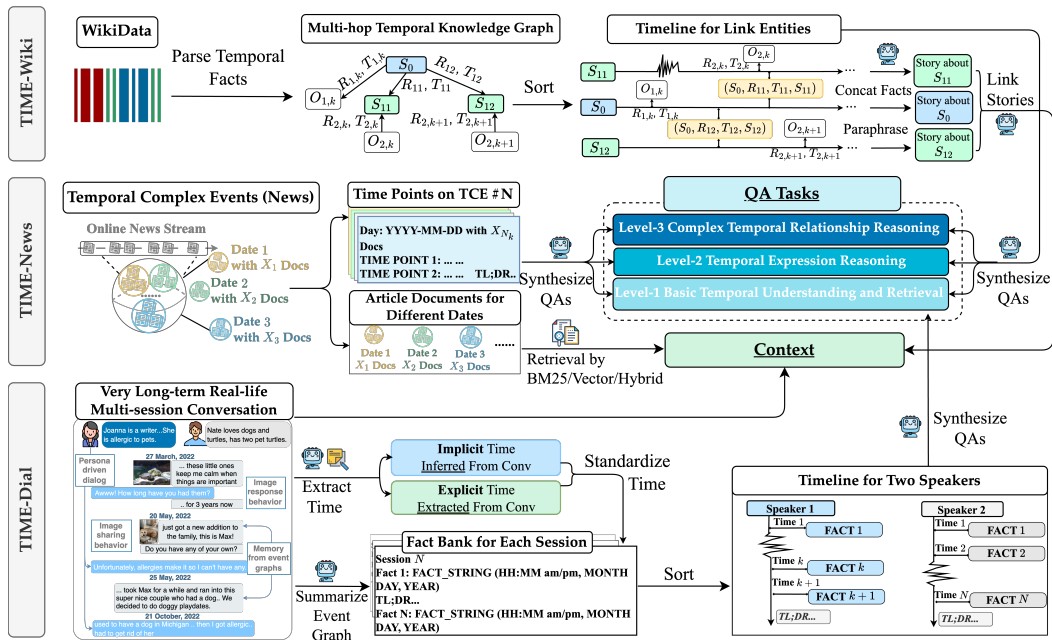

Figure 2: Dataset construction pipeline for TIME. In the process of QA synthesis for each sub-dataset, we first collect temporal facts (temporal knowledge graphs for TIME-WIKI, time points for TIME-NEWS, fact bank for TIME-DIAL). Then timelines are generated for QA data synthesis.

temporal information retrieval. For temporal quantification, we develop (3) `Computation` testing duration calculation between explicit time markers, combined with (4) `DurationCompare` measuring interval comparison capability between events. Finally, temporal sequencing is addressed through (5) `OrderCompare` that examines chronological ordering understanding. This tripartite structure evaluates core competencies through: basic temporal retrieval (`Extract+Localization`), quantitative temporal analysis (`Computation+DurationCompare`), and sequential relationship understanding (`OrderCompare`).

**Level-2: Temporal Expression Reasoning.** Level-2 requires models to locate event details through temporal expression reasoning. We design three subtasks: (1) `Explicit Reasoning` that demands inference based on unmentioned time points/ranges (e.g., Q: "What was Mauro Morelli's occupation between 1967-1973?"), (2) `Order Reasoning` that requires temporal positioning through ordinal expressions (e.g., "Mauro Morelli's second job"), and (3) `Relative Reasoning` that involves contextual interpretation of relative temporal references (e.g., "Where did Mauro Morelli work closest to Event A?"). These tasks collectively evaluate models' ability to understand complex event details under the premise of performing **multi-hop temporal reasoning**.

**Level-3: Complex Temporal Relationship Reasoning.** Level-3 requires models to comprehend and reason about complex temporal relationships among multiple events. We develop three subtasks: (1) `Co-temporality` that identifies overlapping temporal relationships between concurrent events (e.g., "Where did Elon Musk work concurrently with his OpenAI position?"), (2) `Timeline` that infers correct chronological ordering of multiple events (e.g., sorting 8 political events into temporal sequence), and (3) `Counterfactual Reasoning` that demands temporal inference under altered temporal premises contradicting the original context (e.g., "If Event X occurred 3 years later, how would it affect Event Y?"). These tasks collectively evaluate models' capacity to handle **multi-event temporal interactions** through co-occurrence analysis, timeline construction, and hypothetical temporal reasoning.

## 3.2 Data Source

**TIME-WIKI** We choose Wikidata as the data source, leveraging its extensive collection of continuously updated real-world temporal facts that capture time-evolving knowledge across diverse domains. Wikidata's structured temporal relations facilitate efficient extraction of temporal knowledge graphs.

To ensure comprehensive coverage of real-world knowledge, we systematically select 6 categories encompassing 34 representative Wikidata properties for fact retrieval, as detailed in Table 5.

**TIME-NEWS** We utilize online news articles as the data source. Previous work[58] provides large-scale, high-quality online news articles accompanied by corresponding timelines, effectively capturing multiple temporal complex events. A temporal complex event (TCE) refers to a sequence of interrelated events centered around a specific topic within a defined time period, such as the Israeli-Palestinian conflict that occurred between February and March 2015. As illustrated in Figure 5, each temporal complex event encompasses multiple dates, with event details evolving. The statistical characteristics of the TIME-NEWS data source are presented in Table 6.

**TIME-DIAL** We employ very long-term multi-session conversations as the data source, utilizing publicly available data from LOCOMO[31] and REALTALK[24]. Each conversation consists of multi-session dialogs between two distinct persona-driven speakers, incorporating image sharing behaviors, reflecting speaker personas and demonstrating complex temporal dependencies and contextual linkages across events. An example is illustrated in Figure 7, with detailed statistical analysis presented in Table 7.

### 3.3 Dataset Construction

As shown in Figure 2, for each data source(§3.2), we systematically collect temporal facts(§3.3) and then extract corresponding timelines(§3.3). Leveraging these timelines, we employ data synthesis methods to generate question-answer pairs(§3.3). To ensure data quality, we conduct human annotation on a randomly sampled subset, resulting in the high-quality TIME-LITE benchmark(§3.3). Details can be seen in Appendix A.

**Temporal Facts Collection** For TIME-WIKI, we employ the SLING framework to parse Wikidata, extracting temporal fact quadruples and constructing a multi-hop temporal knowledge graph (TKG), with detailed methodology in Appendix A.1.2, where each temporal fact is formalized as $(S_{i,j}, R_{i+1,k}, T_{i+1,k}, O_{i+1,k})$ and link entities (e.g., $S_0$, $S_{11}$, $S_{12}$ in Figure 2) serve as central nodes for collecting related temporal facts; for TIME-NEWS, we directly utilize high-quality time points from [58] as temporal facts; in TIME-DIAL, we leverage LLMs to summarize event graphs, extracting multiple temporal facts per session comprising speaker atomic facts and corresponding timestamps, while standardizing both implicit (e.g., "*two days ago*") and explicit (e.g., "*8:35 am, Feb 23, 2022*") temporal expressions through LLM processing and manual verification.

**Timeline Generation** The synthesis of question-answer pairs for subtasks such as `order reasoning`, `relative reasoning`, and `co-temporality` necessitates precise organization of timelines for individual entities or event groups. To this end, we systematically construct timelines across all three datasets. Specifically, for TIME-WIKI, we chronologically organize the temporal facts associated with link entities in the TKG. For TIME-DIAL, we separately arrange the temporal facts for each speaker in the conversation in chronological order. In the case of TIME-NEWS, we directly utilize the pre-existing time points from each temporal complex event as the timeline, as illustrated in Figure 5.

**Context Collection** We devise distinct context processing strategies tailored to each dataset's characteristics. For TIME-WIKI, we first aggregate temporal facts centered around link entities, reformulating them into coherent narrative segments, which are then integrated into natural and fluent contexts using `DeepSeek-V3`. For TIME-NEWS, given that the average token count per TCE exceeds 500,000, as shown in Figure 6, rendering full-text evaluation impractical, we employ Retrieval-Augmented Generation (RAG) to extract the most relevant text segments from associated articles as context, with implementation details elaborated in (§4.1). For TIME-DIAL, we directly utilize the original conversations as context to preserve their integrity and authenticity.

**QA Synthesis and Formats** We design distinct QA synthesis pipelines tailored to each subtask's characteristics, combining rule-based templates with `DeepSeek-V3` and `DeepSeek-R1` models, as detailed in Table 8. For TIME-WIKI and TIME-DIAL, we employ a rule-based approach grounded in timeline construction to establish logical relationships between questions and gold answers, followed by LLM-based natural language refinement. In TIME-NEWS, we leverage existing Time Points to generate timeline-aligned contexts centered on specific entities using `DeepSeek-V3`, which then serve as prompts for LLM-driven QA generation, as illustrated in Figure 10. Building upon the framework proposed in [58], our Time Points exhibit well-defined timelines and logical structures, enabling us

Table 1: Stats of TIME and TIME-LITE. # QA indicates the number of question-answer pairs.

| Dataset | # QA | Dataset | # QA |
|---|---|---|---|
| **TIME** | **38,522** | **TIME-LITE** | **943** |
| TIME-WIKI | 13,848 | TIME-LITE-WIKI | 322 |
| TIME-NEWS | 19,958 | TIME-LITE-NEWS | 299 |
| TIME-DIAL | 4,716 | TIME-LITE-DIAL | 322 |

to directly incorporate these contexts into prompts for high-quality QA synthesis. Implementation details are provided in Appendix A.4.

We design three primary question formats for different subtasks: free-form, single-choice, and multiple-choice, with their distribution detailed in Table 9. However, for TIME-NEWS and TIME-DIAL, the free-form format's standard answers often exhibit numerous synonymous expressions, leading to significant calibration errors in direct evaluation. To enhance evaluation accuracy while increasing data diversity[2], we adopt the STARC framework[1] to synthesize multiple-choice questions with misleading options, as illustrated in Figure 9. The detailed methodology for generating misleading options is presented in Appendix A.4.

**Quality Control** To validate the quality of our synthesized data and establish a high-quality subset for evaluation, we conducted a comprehensive manual annotation process. For data sampling, we employed a systematic approach to ensure representativeness across all sub-datasets. Using a fixed random seed of 42, we randomly sampled 30-40 QA pairs from each task within the three sub-datasets (TIME-WIKI, TIME-NEWS, and TIME-DIAL). This process yielded a total of 1,071 QA pairs for annotation, with 352 from TIME-WIKI, 359 from TIME-NEWS, and 360 from TIME-DIAL. To ensure annotation quality, we recruited annotators through professional forums and conducted rigorous qualification tests. From the initial pool of 8 professional annotators, we selected the top 3 performers based on both efficiency and quality metrics to establish our final human evaluation benchmark. We propose Word-level Similarity as a novel metric for evaluating annotation consistency, achieving a score of 0.6626, which demonstrates the high reliability of our annotated data. Details can be seen in Appendix A.5.

### 3.4 TIME-LITE

To facilitate efficient and reliable evaluation of temporal reasoning capabilities, we introduce TIME-LITE, a lightweight dataset derived from the manually annotated subset sampled in §3.3. Through multiple rounds of expert review and answer verification, we curated 945 high-quality QA pairs, all of which have undergone rigorous manual validation to ensure assessment accuracy and reliability.

To assess the quality of our synthesized data, we conducted a systematic comparison between the sampled data from §3.3 and its manually reviewed counterpart (TIME-LITE). The analysis revealed a high consistency rate of 89.13%, demonstrating the reliability of our data synthesis pipeline.

## 4 Evaluation

### 4.1 Experimental Setup

**Settings** We conducted comprehensive evaluations on the TIME and TIME-LITE datasets across 24 models (see Appendix C.1 for model details). All experiments employed greedy search decoding strategy. For the TIME-NEWS and TIME-LITE-NEWS dataset, we implemented the retrieval augmented generation (RAG) framework with three retrieval strategies: BM25, Vector, and Hybrid (detailed in Appendix C.3). Given the `Extract` task's limited effectiveness in assessing temporal retrieval capabilities under the RAG framework, we excluded it from our evaluation.

**Metrics** We evaluate free-form QA tasks with token-level metrics: Exact Match (EM) for the `Timeline` task, and F1 score for other free-form QA tasks. For single-choice and multiple-choice QA tasks, we use option-level F1 scores, emphasizing macro F1 for a comprehensive evaluation across all options. Details are shown in Appendix C.2.

Table 2: Results for TIME-WIKI. Abbreviations: Ext.L Extract, Loc.:Localization, Comp.: Computation, DC.: Duration Compare, OC.: Order Compare ER.: Explicit Reasoning, OR.: Order Reasoning, RR.: Relative Reasoning, Co-tmp.: Co-temporality, TL.: Timeline and CTF.: Counterfactual. Top-1 result for each blank are **bold**.

| Model | Level 1 | | | | | Level-2 | | | Level-3 | | |
|---|---|---|---|---|---|---|---|---|---|---|---|
| | Ext. | Loc. | Comp. | DC. | OC. | ER. | OR. | RR. | Co-tmp. | TL. | CTF. |
| *Non-reasoning Models* (TIME-WIKI) | | | | | | | | | | | |
| Llama-3.1-8B-Instruct | 53.16 | 75.41 | 9.79 | 50.89 | 65.49 | 28.96 | 31.72 | 24.53 | 31.36 | 0.92 | 28.60 |
| Qwen2.5-7B | 35.33 | 67.19 | 24.22 | 23.73 | 65.36 | 10.11 | 14.66 | 5.45 | 2.45 | 0.00 | 0.98 |
| Qwen2.5-14B | 33.58 | 71.26 | 20.53 | 50.64 | 66.49 | 7.42 | 15.37 | 20.68 | 17.11 | 0.00 | 27.95 |
| Qwen2.5-7B-Instruct | 57.58 | 65.30 | 32.34 | 52.22 | 68.75 | 44.76 | 35.48 | 26.79 | 36.68 | 1.08 | 38.42 |
| Qwen2.5-14B-Instruct | 71.02 | 74.49 | 26.37 | 63.50 | 82.76 | 52.93 | 38.94 | 30.34 | 33.68 | 2.62 | 43.16 |
| Qwen2.5-72B-Instruct | **81.70** | **83.84** | **41.37** | **66.64** | **84.22** | **70.13** | **44.84** | **35.23** | **51.17** | **4.08** | **50.68** |
| *Reasoning Models* (TIME-WIKI) | | | | | | | | | | | |
| Deepseek-R1-Distill-Llama-8B | 66.75 | **68.82** | **57.27** | 83.47 | 90.22 | 51.17 | 37.36 | 32.41 | 31.04 | 5.31 | 37.30 |
| Deepseek-R1-Distill-Qwen-7B | 54.89 | 65.04 | 56.63 | 77.85 | 85.71 | 48.88 | 32.53 | 30.57 | 29.74 | 0.54 | 37.38 |
| Deepseek-R1-Distill-Qwen-14B | 67.66 | 66.33 | 51.25 | 81.21 | 92.97 | 58.94 | **43.49** | 35.63 | 36.30 | 14.54 | 45.69 |
| QwQ-32B | **74.99** | 67.75 | 49.59 | **88.20** | **93.53** | **60.61** | 37.77 | **36.39** | **37.76** | **25.38** | **53.13** |
| *Advanced Models* (TIME-LITE-WIKI) | | | | | | | | | | | |
| Deepseek-V3 | 93.33 | **84.51** | 23.76 | 71.43 | 83.33 | 75.69 | 39.77 | 41.76 | 46.62 | 10.00 | 44.82 |
| Deepseek-R1 | 96.67 | 77.61 | 46.39 | 89.29 | **93.33** | 78.20 | **57.09** | **57.79** | 47.45 | **33.33** | **55.71** |
| GPT-4o | **98.89** | 83.24 | 33.82 | 67.86 | 90.00 | 80.68 | 45.83 | 46.56 | 45.45 | 20.00 | 50.72 |
| OpenAI o3-mini | 96.67 | 80.83 | **49.17** | **92.86** | 93.33 | **82.24** | 52.62 | 48.98 | **54.34** | 33.33 | 52.07 |

## 4.2 Result

### 4.2.1 Real-world Scenario Analysis

**Knowledge intensive events makes it challenging for capturing complex temporal expression and relationship.** As shown in Table 2, models face significant challenges in comprehending implicit temporal expressions and intrinsic temporal relationships between events. For o3-mini, it achieves only 52.62% and 48.98% on `Order Reasoning` and `Relative Reasoning` tasks respectively, and merely 54.34% on the `Co-temporality` task. In contrast, its performance on basic temporal retrieval and comprehension tasks (Level-1) approaches 80% for 4 tasks. This phenomenon suggests that the complex and diverse associations between temporal information and entities in knowledge-intensive scenarios substantially hinder models' ability to accurately correlate time with facts.

**Complex dynamic events constrain models' ability to comprehend basic temporal relationship and construct coherent timelines.** As shown in Table 3 (TIME-NEWS), models is challenged by comprehending fundamental temporal relationships, including time intervals and ordering, as well as constructing coherent timelines. For instance, the reasoning model o3-mini achieves a maximum performance of only 63.33% on both `Duration Compare` and `Order Compare` tasks. Notably, all models demonstrate limited capability in the `Timeline` task, which requires ordering three events, with performance not exceeding 30%. This suggests that the intricate details among complex events lead models to identify multiple similar but imprecise temporal points, resulting in erroneous predictions.

**Very-long multi-session dialog impairs the capability of time retrieval and event-time localization.** As shown in Table 4 (TIME-DIAL), the maximum accuracy of open-source vanilla models and test-time scaled models on `Extract` and `Localization` tasks is merely 40%, substantially lower than their performance on other datasets. This phenomenon can be attributed to two primary factors: first, the extensive dialog context (averaging over 15k tokens, as shown in Table 7) and multi-turn interactions significantly increase the difficulty of temporal localization; second, the frequent use of memory-based temporal expressions in daily dialogs (e.g., "Last Saturday"), which necessitate reasoning with the conversation timestamp to pinpoint the precise date, further hinders accurate timestamp identification.

### 4.2.2 Temporal Reasoning Tasks Analysis

**Time retrieval ability is significantly correlated with almost all aspects of temporal reasoning tasks.** To investigate the impact of basic temporal retrieval capabilities on temporal reasoning

Table 3: Results for TIME-NEWS. Top-3 articles are retrieved. Abbreviations follow Table 2.

| Model | Retriever | Level 1 | | | | Level-2 | | | Level-3 | | |
|---|---|---|---|---|---|---|---|---|---|---|---|
| | | Loc. | Comp. | DC. | OC. | ER. | OR. | RR. | Co-tmp. | TL. | CTF. |
| *Non-reasoning Models* (TIME-NEWS) | | | | | | | | | | | |
| Llama3.1-8B-Instruct | BM25 | 47.96 | 27.12 | 39.06 | 39.28 | 81.72 | 66.67 | 77.06 | 80.50 | 3.09 | 47.17 |
| | Vector | 50.99 | 32.13 | 40.94 | 41.17 | 81.33 | 67.67 | 77.67 | 81.50 | 1.94 | 46.22 |
| | Hybrid | 51.81 | 34.51 | 41.78 | 44.11 | 82.50 | 68.94 | 78.39 | 82.89 | 2.55 | 46.44 |
| Qwen2.5-14B-Instruct | BM25 | 68.53 | 70.80 | 42.39 | 46.17 | 83.06 | 70.44 | 79.61 | 82.67 | 26.13 | 59.39 |
| | Vector | 71.68 | 76.28 | 42.22 | 45.67 | 83.94 | 69.33 | 80.33 | 83.44 | 23.68 | **59.67** |
| | Hybrid | 71.00 | 79.75 | 43.61 | 48.72 | 84.72 | 70.39 | 81.44 | 84.06 | 26.61 | 58.61 |
| Qwen2.5-32B-Instruct | BM25 | 68.88 | 79.48 | **46.44** | 51.22 | 84.39 | 70.78 | 81.56 | 85.11 | **27.54** | 54.61 |
| | Vector | **71.76** | 84.46 | 44.78 | 50.61 | 85.22 | 70.94 | 82.11 | 84.39 | 24.16 | 55.83 |
| | Hybrid | 71.57 | 86.62 | 44.78 | **54.83** | **86.28** | **71.17** | **82.72** | **86.11** | 25.92 | 54.06 |
| *Reasoning Models* (TIME-NEWS) | | | | | | | | | | | |
| Deepseek-R1-Distill-Qwen-7B | BM25 | 39.66 | 60.15 | 38.78 | 53.33 | 76.28 | 60.06 | 70.17 | 74.56 | 17.94 | 37.11 |
| | Vector | 41.17 | 59.81 | **41.72** | 54.56 | 76.44 | 61.94 | 73.89 | 74.67 | 16.44 | 38.78 |
| | Hybrid | 41.42 | 60.28 | 38.22 | 54.78 | 78.22 | 62.67 | 72.72 | 76.39 | 17.08 | 39.06 |
| Deepseek-R1-Distill-Qwen-14B | BM25 | 63.42 | 62.36 | 39.72 | 52.61 | 83.39 | **70.33** | 80.83 | 83.78 | **21.82** | 62.72 |
| | Vector | 65.96 | 63.56 | 39.39 | 51.33 | 84.89 | 69.22 | 81.28 | 83.89 | 19.58 | **63.44** |
| | Hybrid | **66.11** | **66.29** | 39.39 | **54.94** | **85.61** | 69.89 | **82.67** | **85.00** | 21.10 | 62.00 |
| *Advanced Models* (TIME-LITE-NEWS) | | | | | | | | | | | |
| GPT-4o | BM25 | 79.26 | 10.56 | 43.33 | 43.33 | 76.67 | 70.00 | **93.33** | **93.33** | 24.14 | 43.33 |
| | Vector | 75.56 | 15.00 | 40.00 | 53.33 | **80.00** | 66.67 | 86.67 | 90.00 | 24.14 | 40.00 |
| | Hybrid | **80.56** | **20.00** | 33.33 | 46.67 | 73.33 | 66.67 | 86.67 | 90.00 | 13.79 | **46.67** |
| OpenAI o3-mini | BM25 | 72.59 | 12.78 | 56.67 | 60.00 | 73.33 | **83.33** | 86.67 | **93.33** | 27.59 | 33.33 |
| | Vector | 76.67 | 18.33 | **63.33** | **63.33** | **80.00** | 66.67 | 86.67 | 80.00 | 24.14 | 33.33 |
| | Hybrid | 77.94 | 16.67 | 56.67 | **63.33** | 76.67 | 63.33 | 80.00 | 86.67 | **27.59** | 36.67 |

tasks, we computed correlation coefficients between `Extract` and `Localization` tasks and other task performances based on 8 vanilla models across three TIME-LITE subsets. Specifically, we represented each task as a vector of model performance across various temporal reasoning tasks and performed agglomerative clustering using correlation coefficients as the distance metric. The results (shown in Figure 3) demonstrate that `Extract` and `Localization` tasks exhibit significant correlations (correlation coefficient > 0.5) with nearly all other tasks, confirming a strong relationship between basic temporal retrieval and higher-level temporal reasoning capabilities.

**Grasping timeline over multiple events is much challenging for long-range contexts.** As shown in Table 2, 3 and 4, all models demonstrate suboptimal performance on the `Timeline` task. Notably, small-scale vanilla models achieve accuracy below 10% on both TIME-WIKI and TIME-DIAL datasets. Even in the relatively simpler TIME-NEWS dataset, merely reordering three events poses a significant challenge. This difficulty stems from the `Timeline` task's requirement for simultaneous complex temporal information retrieval and global temporal ordering reasoning, which is substantially more challenging than basic tasks like `Order Compare` that only require understanding the sequence of two events.

### 4.2.3 Impact of Test-time Scaling

**Test-time scaling benefits temporal logical reasoning.** Test-time scaling enhances models' performance in complex logical reasoning tasks by strengthening their chain-of-thought capabilities. To systematically evaluate its impact on temporal reasoning, we compare R1-Distill models and their vanilla counterparts across multiple tasks (see Table 2 and 4). Experimental results demonstrate that Deepseek-R1-Distill-Qwen-14B significantly outperforms Qwen2.5-14B-Instruct in temporal reasoning tasks such as `Order Compare` and `Duration Compare`, as well as in handling complex temporal-event relationships in the `Counterfactual` task, achieving performance improvements of 24.44%, 11.33%, and 12.0% respectively on the TIME-DIAL dataset. Our analysis further reveals that advanced test-time scaled models, including o3-mini and Deepseek-R1, consistently outperform their non-reasoning counterparts in logical reasoning-based tasks, demonstrating the effectiveness of test-time scaling in enhancing complex reasoning capabilities.

**Test-time scaling is not consistently effective for time retrieval and event location.** Experimental results reveal significant performance variations of test-time scaling models across different datasets.

Table 4: Results for TIME-DIAL. Abbreviations follow Table 2.

| Model | Level 1 | | | | | Level-2 | | | Level-3 | | |
|---|---|---|---|---|---|---|---|---|---|---|---|
| | Ext. | Loc. | Comp. | DC. | OC. | ER. | OR. | RR. | Co-tmp. | TL | CTF. |
| *Non-reasoning Models* (TIME-DIAL) | | | | | | | | | | | |
| Llama-3.1-8B-Instruct | 27.45 | **38.61** | 9.05 | **48.44** | **52.67** | 38.22 | **46.22** | **57.33** | **72.00** | 0.00 | 38.00 |
| Qwen2.5-7B-Instruct | 36.51 | 30.91 | 23.25 | 41.11 | 41.33 | 31.11 | 34.22 | 44.44 | 58.00 | 0.22 | **46.44** |
| Qwen2.5-14B-Instruct | 38.85 | 30.83 | 16.35 | 42.00 | 47.78 | 38.22 | 38.67 | 49.11 | 57.33 | 0.00 | 34.89 |
| Qwen2.5-32B-Instruct | **40.67** | 33.56 | **23.45** | 40.89 | **52.67** | **43.33** | 36.67 | 46.00 | 63.11 | **0.67** | 40.44 |
| *Reasoning Models* (TIME-DIAL) | | | | | | | | | | | |
| Deepseek-R1-Distill-Llama-8B | 40.21 | 36.37 | 14.69 | 40.89 | 57.11 | 34.89 | 34.00 | 40.44 | 54.67 | **0.44** | 42.22 |
| Deepseek-R1-Distill-Qwen-14B | **40.40** | 18.34 | 12.98 | 53.33 | 72.22 | 54.67 | 40.44 | 53.33 | 66.89 | 0.22 | 46.89 |
| Deepseek-R1-Distill-Qwen-32B | 39.28 | **35.79** | 22.87 | 58.22 | 75.33 | 57.56 | **41.78** | 54.89 | 72.67 | 0.22 | **49.78** |
| *Advanced Models* (TIME-LITE-DIAL) | | | | | | | | | | | |
| Deepseek-V3 | 52.63 | 42.67 | 13.00 | 70.00 | 73.33 | 40.00 | 26.67 | 60.00 | 56.67 | 3.33 | 43.33 |
| Deepseek-R1 | **65.00** | 48.56 | 22.61 | **73.33** | **86.67** | 76.67 | 53.33 | 66.67 | **76.67** | 10.00 | 53.33 |
| GPT-4o | 61.08 | **52.98** | 14.00 | 40.00 | 76.67 | 60.00 | 43.33 | 66.67 | 76.67 | 0.00 | 46.67 |
| OpenAI o3-mini | 41.41 | 45.30 | **29.90** | 56.67 | **86.67** | 76.67 | **60.00** | **70.00** | 70.00 | **10.00** | 46.67 |

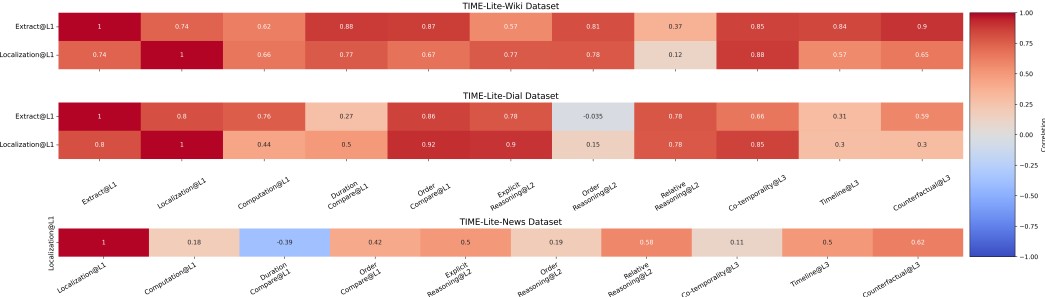

Figure 3: Task correlation heatmap highlighting the relationship between `Extract` and `Localization` tasks and other temporal reasoning tasks. Note: `Extract` task is excluded from TIME-LITE-NEWS evaluation.

On TIME-WIKI(shown in Table 2), Deepseek-R1-Distill-Qwen-14B underperforms Qwen2.5-14B-Instruct by 3.36% and 8.16% in `Extract` and `Localization` tasks respectively. Conversely, on TIME-DIAL (shown in 4), it achieves a 1.55% improvement in `Extract` but suffers a 12.49% decline in `Localization`. This discrepancy stems from the temporal information retrieval mechanism of test-time scaling models: their systematic context traversal strategy benefits multi-session dialog scenarios but may lead to overthinking cycles after retrieval errors, hindering error correction (see case in Appendix D).

#### 4.2.4 Impact of Retrievers in TIME-NEWS

Experimental results (shown in Table 3) demonstrate that the choice of retriever significantly impacts temporal reasoning performance. Taking GPT-4o as an example, its performance with the Hybrid retriever is over 10% lower than with BM25 and Vector retrievers in the `Timeline` task. Similarly, a 10% performance gap exists across different retrievers in the `Order Compare` task. This finding suggests that accurate temporal fact retrieval is crucial for processing dynamic information, directly affecting the effectiveness of complex event reasoning. Notably, in `Explicit Reasoning` and `Order Reasoning` tasks, the performance differences among models under the same retriever setting are significantly reduced, indicating that the retriever plays a dominant role in temporal reasoning for these tasks, even overshadowing the inherent capabilities of different models.

## 5 Conclusion

TIME presents a comprehensive benchmark for evaluating temporal reasoning in LLMs, featuring three hierarchical levels with 11 subtasks that systematically assess temporal understanding. Our benchmark captures real-world complexities through knowledge associations, temporal dynamics,

and long-term interactions. We introduce TIME-LITE, a fully human-annotated subset for efficient evaluation. Extensive experiments across diverse models reveal that test-time scaling significantly enhances logical reasoning while showing varied effects on time retrieval. These findings provide critical insights for advancing temporal reasoning capabilities. TIME establishes a foundation for rigorous evaluation and deeper understanding of temporal reasoning, paving the way for future advancements in this essential NLP capability.

## Acknowledgments

This work was supported by Beijing Natural Science Foundation (No. L253020) and National Natural Science Foundation of China (62036001). The corresponding author is Houfeng Wang.

Special thanks are extended to Jingyuan Ma, Ze Meng, Zeqi Zhao, Min Zhu and Yangtian Yi for their important contributions to the quality control of the dataset. We also thank the anonymous reviewers for their helpful comments and suggestions.

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

## A    Benchmark Construction

### A.1    TIME-WIKI Construction

#### A.1.1    Data Source

We utilize Wikidata as the data source for constructing TIME-WIKI. We downloaded the WikiData Dump[12] from November 1, 2024 from the WikiMedia[3].

**WikiData**    WikiData is a free and open collaborative knowledge base that serves as a central storage for the structured data of Wikimedia projects. It organizes information into items, each identified by a unique Q-number, and describes them using statements composed of properties and values. This structured data, akin to a large-scale knowledge graph, provides a valuable resource for Natural Language Processing tasks such as entity linking, relation extraction, and building comprehensive linguistic resources by connecting text to real-world entities and their relationships.

Wikidata contains abundant real-world temporal facts that can be extracted to form multi-hop structured temporal knowledge graphs, which can then be transformed into unstructured temporal contexts for evaluating models' temporal fact comprehension capabilities.

#### A.1.2    Temporal Knowledge Graph Construction

Using SLING[4], we parsed temporal facts under given relations based on predefined rules, ultimately generating structured temporal knowledge graph data. Subsequently, we enhanced the temporal knowledge graph data to obtain 1,300 multi-hop temporal knowledge graphs as the data source for QA construction.

To comprehensively reflect world knowledge, we selected six categories of relations when constructing temporal facts from Wikidata: (1) Education, employment, and organizational affiliation, (2) Family relations, (3) Geographical location relations, (4) Naming relations, (5) Significant event, and (6) Role/Identity relations, as shown in Table 5. For each relation, we meticulously designed corresponding natural language templates to facilitate the subsequent synthesis of contexts containing complex temporal facts using LLMs.

**SLING**    SLING is a task processing pipeline designed for downloading and processing Wikipedia and Wikidata dumps. It leverages freely available dump files from Wikimedia and converts them into the SLING frame format through its workflow task system. Specifically, following the construction process of TempLAMA[9], we first installed the SLING toolkit (version 3.0.0) and downloaded two key datasets: the SLING-formatted WikiData KB and the WikiData to Wikipedia mapping. We then used SLING to extract facts and relations with temporal properties **P580 (start time)**[5], **P582 (end time)**[6], and **P585 (point in time)**[7], filtering out non-entity objects and null objects. We mapped entity names to Wikipedia page titles, generating structured temporal knowledge graph data. The resulting temporal knowledge graph data contains fact relation sets from both subject and object perspectives.

**Data Augmentation of Temporal Knowledge Graph**    We first filtered entities associated with at least three facts from the SLING-generated temporal knowledge graph. We then removed facts with duplicate temporal information and alternately combined (s, r, o) relations from both subject and object perspectives. Specifically, we constructed a 2-hop temporal knowledge graph where we randomly selected 2-3 object entities as *intermediate linking entities* at each hop. Each *intermediate linking entity* then served as a new subject entity, concatenating new temporal fact relations. Finally,

---

[1]`https://dumps.wikimedia.org/enwiki/20241101/enwiki-20241101-pages-articles-multistream.xml.bz2`

[2]`https://dumps.wikimedia.org/enwiki/20241101/enwiki-20241101-pages-articles-multistream-index.txt.bz2`

[3]WikiMedia: `https://dumps.wikimedia.org`

[4]SLING: `https://github.com/ringgaard/sling`

[5]P580 *start time*: `https://www.wikidata.org/wiki/Property:P580`

[6]P582 *end time*: `https://www.wikidata.org/wiki/Property:P582`

[7]P585 *point in time*: `https://www.wikidata.org/wiki/Property:P585`

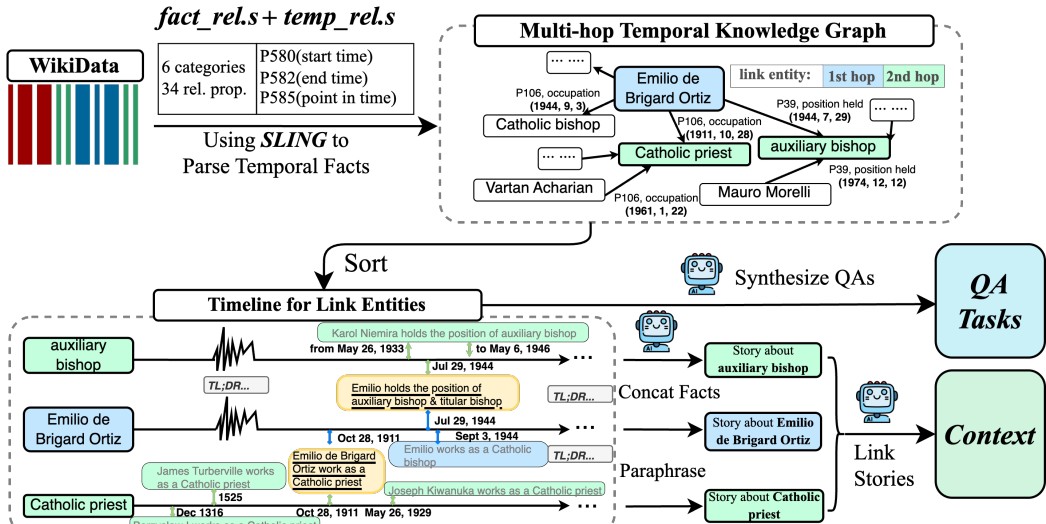

Figure 4: An overview of the TIME-WIKI benchmark construction pipeline. Beginning with Wikidata as the data source, temporal facts are parsed using SLING. These facts are then used to construct multi-hop temporal knowledge graphs. Timelines for link entities are generated from these graphs by sorting temporal facts. Finally, these timelines are used to synthesize question-answer (QA) pairs, and corresponding context is generated by concatenating and paraphrasing stories derived from the timelines, forming the final QA tasks.

we selected 1,300 high-quality multi-hop temporal knowledge graphs as the data source for subsequent QA construction.

### A.1.3 Collect Context for Evaluation for TIME-WIKI

To provide comprehensive world knowledge contexts for evaluation, we synthesize evaluation contexts for LLMs by leveraging the previously constructed Temporal Knowledge Graph. Specifically, we first build timelines for link entities from the temporal knowledge graph as structured raw data. For each link entity, we chronologically concatenate temporal facts and paraphrase them into coherent stories. Finally, we prompt LLMs to generate contexts centered around three interconnected link entities. Below we present our few-shot prompts for paraphrasing facts into stories and concatenating stories for TIME-WIKI, followed by an example of the final concatenated context.

Table 5: Selected Wikidata Properties Categorized by Semantic Type, including Property IDs, Query Frequencies, and Natural Language Templates. Categories are highlighted by different colors: Education, employment, and organizational affiliation , Family relations , Geographical location relations , Naming relations , Significant event , and Role/Identity relations . *Num.* represents the number of facts with the corresponding relation that are extracted using SLING.

| WikiData ID | Relation | Num. | Template |
| --- | --- | --- | --- |
| P39 | position held | 165479 | <subject> holds the position of <object> |
| P54 | member of sports team | 903683 | <subject> plays for <object> |
| P69 | educated at | 68121 | <subject> attended <object> |
| P102 | political party | 16871 | <subject> is a member of <object> |
| P108 | employer | 63298 | <subject> works for <object> |
| P127 | owned by | 8065 | <object> owns <subject> |
| P286 | head coach | 10389 | <object> becomes the head coach of <subject> |
| P106 | occupation | 33209 | <subject> works as a <object> |
| P463 | member of | 41425 | <subject> becomes a member of <object> |
| P1416 | affiliation | 938 | <subject> becomes affiliated with <object> |
| P22 | father | 6 | <object> becomes the father of <subject> |
| P40 | child | 10 | <subject> has a child <object> |
| P101 | spouse | 175 | <subject> is the spouse of <object> |
| P17 | country | 37344 | <subject> becomes a part of the country: <object> |
| P131 | located in the administrative territorial entity | 82221 | <subject> becomes a part of <object> |
| P276 | location | 2065 | <subject> becomes a part of <object> |
| P551 | residence | 7771 | <subject> resides in <object> |
| P740 | location of formation | 5 | <subject> was formed in <object> |
| P159 | headquarters location | 5619 | <subject> has its headquarters in <object> |
| P138 | named after | 3558 | <object> names after <subject> |
| P155 | follows | 272 | <subject> follows <object> |
| P156 | followed by | 290 | <object> follows <subject> |
| P793 | significant event | 4436 | <subject> experienced the significant event <object> |
| P50 | author | 47 | <object> writes <subject> |
| P57 | director | 62 | <object> directes <subject> |
| P86 | composer | 92 | <object> composes <subject> |
| P170 | creator | 21 | <object> creates <subject> |
| P175 | performer | 132 | <object> performs <subject> |
| P676 | lyrics by | 8 | <object> wrote the lyrics for <subject> |
| P110 | illustrator | 11 | <object> illustrates <subject> |
| P162 | producer | 31 | <object> produces <subject> |
| P58 | screenwriter | 29 | <object> writes <subject> as a screenwriter |
| P161 | cast member | 495 | <object> plays the role of <subject> |
| P1431 | executive producer | 32 | <object> produces <subject> as the executive producer |

## Few-shot Prompt for Paraphrasing Facts into Stories for TIME-WIKI

**[Instruction:]** You are a very skilled storyteller. Below are the facts I've extracted from WikiData. Please craft a complete story centered around {link_entity}.

**[Requirements:]**

1. Generate content that accurately conveys all the facts I have provided without omitting any of them.

2. Particular attention must be paid to the accuracy of time expressions, and the temporal relationships between sentences must be coherent and natural.

3. The story should be coherent, natural, and smooth, consistently revolving around the core facts, with concise and precise language, avoiding any unnecessary digressions.

**[Example:]** [Core entity:] Mauro Morelli

[Facts:] Mauro Morelli holds the position of diocesan bishop from May 25, 1981 to March 30, 2005.. Mauro Morelli holds the position of titular bishop in December 12, 1974.. Mauro Morelli holds the position of auxiliary bishop in December 12, 1974.. Mauro Morelli works as a Catholic priest in April 28, 1965.. Mauro Morelli works as a Catholic bishop in January 25, 1975.

[Generated Story:] Mauro Morelli's journey in the Catholic Church is a testament to his unwavering dedication and service. Born into a world of faith, his path was set early on, leading him to become a significant figure in the Church's hierarchy.

On April 28, 1965, Mauro Morelli took his first major step in his religious career by being ordained as a Catholic priest. This ordination marked the beginning of his lifelong commitment to serving the Church and its followers. For nearly a decade, he devoted himself to his priestly duties, gaining the respect and admiration of his parishioners. ... ...

**[Example:]** (Here is another example with the same format of the example above.)

**[TODO:]** Below are the facts and the link entity. Please generate a story in accordance with the aforementioned requirements and with reference to the examples provided.

**[Link entity:]**{link_entity}

**[Facts:]**{raw_text}

**[Generated story:]**

---

## Prompt for Concatenating Stories for TIME-WIKI

**[Instruction:]** You are adept at weaving multiple complete stories together to form a coherent and comprehensive narrative. I will provide you with several stories and the connections between them. I hope you can retain all the facts from the stories without any omissions or errors in time or events.

**[Requirements:]**

1. The content you generate must not omit any time-related information from the texts being concatenated.

2. Particular attention must be paid to the accuracy of time expressions, and the temporal relationships between sentences must be coherent and natural.

3. When concatenating, pay attention to the inherent logical connections between the given multiple stories, and refer to the provided linking facts to generate transitional paragraphs that seamlessly connect the different stories.

**[Link facts:]** {linked_facts}

**[Stories:]** {stories}

**[Generated Story:]**

> **One Example for the Concatenated Story for TIME-WIKI**
>
> Mauro Morelli's life was a testament to unwavering dedication and service to the Catholic Church. His journey began with a profound commitment to his faith, which shaped his path toward becoming a respected leader within the Church's hierarchy. On April 28, 1965, Mauro Morelli was ordained as a Catholic priest, marking the beginning of his spiritual and pastoral vocation. For nearly a decade, he devoted himself to his priestly duties, guiding and nurturing the communities he served. His compassion, wisdom, and deep connection to his faith earned him the trust of his parishioners and the attention of his superiors.
>
> Recognizing his leadership qualities, the Church appointed Mauro Morelli as both a titular bishop and an auxiliary bishop on December 12, 1974. These dual roles signified a new chapter in his journey, as he took on greater responsibilities in assisting the diocesan bishop and overseeing pastoral and administrative tasks. His dedication to these roles showcased his ability to balance spiritual guidance with effective governance. Just over a month later, on January 25, 1975, Mauro Morelli was consecrated as a Catholic bishop. This formal consecration solidified his position as a key figure within the Church, empowering him to lead with authority and grace. His work as a bishop further deepened his impact on the communities he served, as he continued to champion the values of the Church.
>
> ***Skipping 3 paragraphs here...***
>
> As we reflect on Mauro Morelli's life and the broader history of auxiliary and titular bishops, we are reminded of the importance of service, humility, and faith. The roles of these bishops, though sometimes overlooked, are a testament to the Church's enduring commitment to its mission of love, guidance, and spiritual care. Their contributions, woven together, form a testament to the enduring legacy of leadership within the Catholic tradition.

## A.2   TIME-NEWS Construction

### A.2.1   Data Source

TIME-NEWS investigates temporal reasoning in real-world complex dynamics. Prior work typically employs news articles to analyze intricate temporal event sequences [30, 58]. We utilize the open-source dataset from [58] as our data source, selected for its extensive scale, accessibility, and rich temporal information. This dataset leverages LLMs to systematically extract and analyze event chains within temporal complex events (TCEs), where each TCE comprises multiple news articles spanning extended periods.

Specifically, dataset [58] contains 2,289 TCEs, with each event averaging 29.31 articles and spanning 17.44 days. The open-source data provides both the corpus and corresponding outline for each TCE, as shown in Figure 5. We randomly sampled 600 temporal complex events (TCEs) with at least 5 distinct dates and at most 10 distinct dates as the data source for TIME-NEWS, with detailed statistics presented in Table 6 and the number of dates distribution presented in Figure 6. For each TCE, we utilize its outline for QA synthesis. During evaluation, we employ the corpus as the retrieval source, where for each question (i.e., query), we specifically retrieve the top-k relevant news articles.

Table 6: Statistics of the 600 selected Temporal Complex Events (TCEs) in TIME-NEWS. Note that "Time Span" means the span between the earliest and the latest date of the TCE. "Art." and "Tok." are abbreviations for "Article" and "Token" respectively. Token counts are calculated using *tiktoken's cl100k_base* encoder.

|      | # Art. / TCE | # Art. / Date | # Tok. / TCE | # Tok. / Art. | Time Span / TCE | # Dates / TCE |
|------|--------------|---------------|--------------|---------------|-----------------|---------------|
| Avg. | 871.46       | 116.95        | 527,418.05   | 605.21        | 405.87          | 7.45          |
| Max. | 2068         | 304           | 1,232,706    | 3,405         | 8,900           | 10            |
| Min. | 344          | 18            | 236,521      | 31            | 4               | 5             |

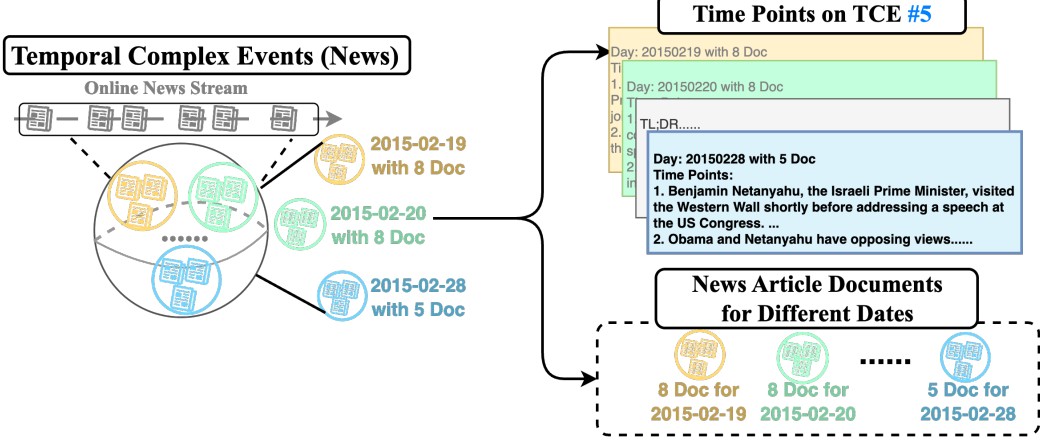

Figure 5: One example of temporal complex events in dataset[58]

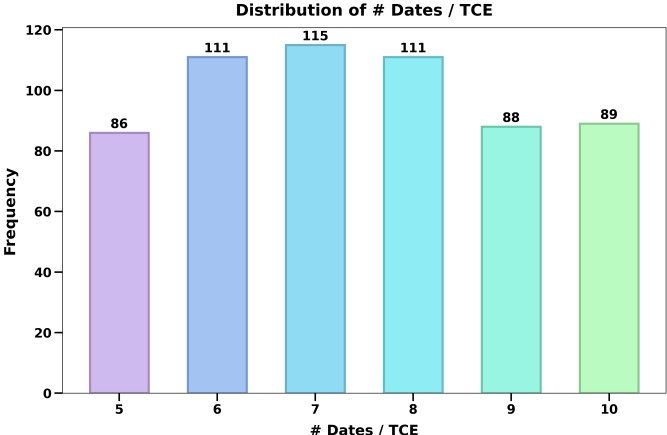

Figure 6: Frequency distribution of length of days in 600 selected Temporal Complex Events (TCEs) in TIME-NEWS

## A.3 TIME-DIAL Construction

In the data construction pipeline of TIME-DIAL, we utilize very long-term real-life multi-session conversations [31, 24] as the evaluation context. Our pipeline begins with summarizing event graphs from conversations. We then employ a combination of LLM extraction and manual verification to extract explicit temporal information and infer implicit temporal relations, which are subsequently standardized into unified temporal expressions. The event graphs are then temporally ordered to construct individual timelines for each speaker. Finally, we synthesize question-answer pairs for 11 subtasks based on these timeline representations. The complete construction pipeline is illustrated in Figure 8.

### A.3.1 Data Source

Real-world interpersonal interactions provide a crucial context for temporal reasoning research. Human conversations exhibit distinct characteristics of long-term continuity and persistence: interlocutors frequently reference previously discussed or mutually known information based on shared memory, enabling natural topic continuation. Moreover, authentic dialogues typically involve numerous sessions with significant temporal spans, containing rich fine-grained information such as discourse markers, brief turns, and subjective expressions. Notably, the processing of temporal information in ultra-long conversational contexts remains an under-explored research area.

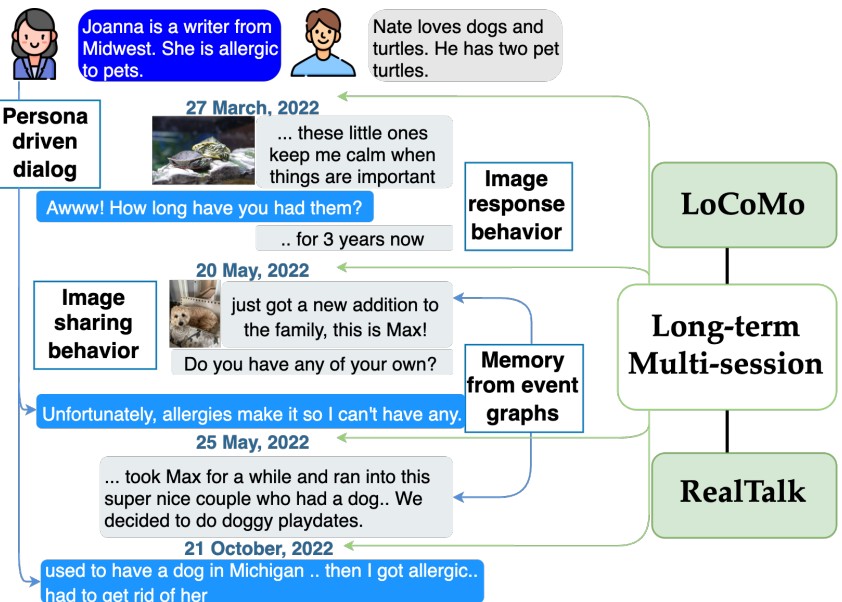

Figure 7: A long-term multi-turn conversation example from the LOCOMO dataset [31]. The interlocutors, Joanna and Nate, are each assigned distinct persona characteristics. The conversation consists of multiple session units, each timestamped and containing several dialogue turns. Throughout the interaction, the participants engage in image sharing and response behaviors. Notably, certain dialogue segments require the speakers to leverage previously established shared memory to facilitate conversational progression. REALTALK [24] maintains an identical data format to the example illustrated in this figure.

We select LoCoMo[31] and RealTalk[24] as our data sources (illustrated in Figure 7) based on the following considerations: First, both datasets contain ultra-long conversations composed of multiple sessions, with persona-driven dialogues generated through data synthesis techniques that align with human characteristics. Second, the datasets feature not only natural language exchanges but also multimodal content such as images, with corresponding responses, closely mirroring real-world conversational scenarios. Given our focus on temporal reasoning evaluation in natural language contexts, we utilize image captions rather than raw image data for analysis.

**LOCOMO** LOCOMO [31] addresses the scarcity of datasets for evaluating long-term memory in open-domain dialogues, which traditionally span few sessions. It was created using a machine-human pipeline where LLM-based agents generate dialogues grounded in personas and, crucially for our work, temporal event graphs. These conversations are extensive, averaging 300 turns and 9K tokens over as many as 35 sessions. This characteristic makes LOCOMO a valuable source for TIME-DIAL, providing the very long-term, multi-session conversational data needed to construct complex temporal reasoning tasks. While LOCOMO agents can share and react to images, for TIME-DIAL, we utilize the textual captions of these images to maintain focus on natural language-based temporal reasoning. As of February 2025, LOCOMO has released only 35 conversations in [8], which we incorporate as part of our data source.

**REALTALK** REALTALK [24] provides a corpus of authentic, multi-session dialogues collected from real-world messaging app interactions over a 21-day period. This dataset captures genuine human conversational dynamics and long-term interaction patterns, offering a benchmark against true human interactions rather than synthetic data. For the construction of TIME-DIAL, REALTALK serves as an invaluable source of very long-term, real-life conversational contexts. These characteristics are instrumental for developing tasks that require understanding complex temporal relationships and

---

[8]LOCOMO-35 is downloaded from Google Drive: https://drive.google.com/file/d/1JimNyO4eryOIjz6dZwLaqWs7glumE2v-/view

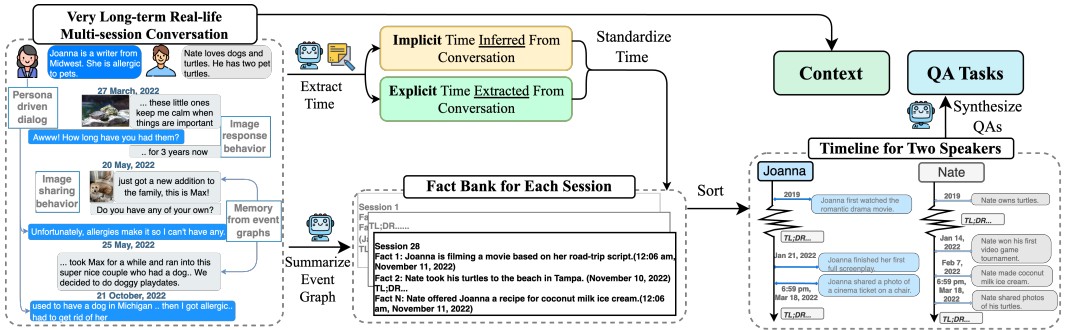

Figure 8: An overview of the construction pipeline of TIME-DIAL. We employ real-world long-term multi-turn conversations as evaluation contexts. First, we extract and summarize event graphs from the conversations. Then, through a combination of LLM extraction and manual verification, we identify explicit temporal information and infer implicit temporal relations, standardizing them into unified temporal expressions. Subsequently, we temporally order the event graphs to construct individual timelines for each speaker. Finally, we generate question-answer pairs covering 11 subtasks based on these timeline representations.

recalling information across extended dialogues, aligning with our objective to evaluate nuanced temporal reasoning capabilities within multi-session conversations.

Table 7: Statistics of LOCOMO-35, REALTALK, and TIME-DIAL datasets. Note: Token counts are calculated using *tiktoken's cl100k_base* encoder. LOCOMO-35 is the open-source subset of LOCOMO as of February 2025. $C$ represents "Conversation".

| Dataset | # $C$ | # Session / $C$ | # Token / $C$ | # Turn / $C$ | # Image / $C$ |
|---|---|---|---|---|---|
| LOCOMO-35[31] | 35 | 20.49 | 14509.91 | 431.23 | 94.94 |
| REALTALK[24] | 10 | 21.90 | 20581.60 | 894.40 | 31.30 |
| TIME-DIAL | 45 | 20.80 | 15859.18 | 534.16 | 80.80 |

## A.4 QA Synthesis

We synthesize QA pairs by integrating timelines and rules, utilizing `DeepSeek-V3` and `DeepSeek-R1`. However, the methodologies employed for data synthesis (as shown in Table 8) and task formats (as presented in Table 9) vary across different tasks. The approaches and details of QA synthesis are elaborated in §A.4.1, §A.4.2, and §A.4.3, respectively.

Table 8: Overview of LLM Utilization Strategies in Question Answering (QA) Construction. The symbols in the cells denote the construction methodology: $\mathcal{R}$: Purely **r**ule-based and template-driven QA generation. $\mathcal{L}$: QA generation fully reliant on **L**arge Language Models (LLMs). $\mathcal{H}_{\mathcal{Q}}$: **H**ybrid for **Q**uestion generation, i.e. rule-based extraction of question logic and answers, with LLMs employed for question phrasing. The suffix '$+\mathcal{M}$' indicates the additional use of LLMs specifically for generating **m**isleading options. Light blue shaded cells denote data generated by the `DeepSeek-R1`, while uncolored cells correspond to the `DeepSeek-V3`.

| Dataset | Level 1 | | | | | Level-2 | | | Level-3 | | |
|---|---|---|---|---|---|---|---|---|---|---|---|
| | Extract | Local. | Comp. | Dur. Comp. | Ord. Comp. | Expl. Reason. | Ord. Reason. | Rel. Reason. | Co-temp. | Timeline | Counterf. |
| TIME-WIKI | $\mathcal{H}_{\mathcal{Q}}+\mathcal{M}$ | $\mathcal{H}_{\mathcal{Q}}$ | $\mathcal{H}_{\mathcal{Q}}$ | $\mathcal{R}$ | $\mathcal{R}$ | $\mathcal{H}_{\mathcal{Q}}$ | $\mathcal{H}_{\mathcal{Q}}$ | $\mathcal{H}_{\mathcal{Q}}$ | $\mathcal{H}_{\mathcal{Q}}$ | $\mathcal{R}$ | $\mathcal{H}_{\mathcal{Q}}$ |
| TIME-NEWS | $\mathcal{L}+\mathcal{M}$ | $\mathcal{L}$ | $\mathcal{L}$ | $\mathcal{L}$ | $\mathcal{L}$ | $\mathcal{L}+\mathcal{M}$ | $\mathcal{L}+\mathcal{M}$ | $\mathcal{L}+\mathcal{M}$ | $\mathcal{L}+\mathcal{M}$ | * | $\mathcal{H}_{\mathcal{Q}}+\mathcal{M}$ |
| TIME-DIAL | $\mathcal{H}_{\mathcal{Q}}+\mathcal{M}$ | $\mathcal{H}_{\mathcal{Q}}$ | $\mathcal{H}_{\mathcal{Q}}$ | $\mathcal{R}$ | $\mathcal{R}$ | $\mathcal{H}_{\mathcal{Q}}$ | $\mathcal{H}_{\mathcal{Q}}+\mathcal{M}$ | $\mathcal{H}_{\mathcal{Q}}+\mathcal{M}$ | $\mathcal{H}_{\mathcal{Q}}+\mathcal{M}$ | $\mathcal{R}$ | $\mathcal{H}_{\mathcal{Q}}+\mathcal{M}$ |

*Not applicable; data directly reused from TCELongBench[58] examples.

**DeepSeek-V3 and DeepSeek-R1** `DeepSeek-V3`[8] and `DeepSeek-R1`[7] are both released by DeepSeek, representing state-of-the-art non-reasoning and reasoning models, respectively. These models offer superior performance at cost-effective rates, making them widely adopted for data

Table 9: Overview of Question Answering (QA) formats across datasets and task categories. The QA formats are denoted by calligraphic letters: $\mathcal{F}$ (**free-form**), $\mathcal{S}$ (**single-choice**), and $\mathcal{M}$ (**multiple-choice**). A light green cell background indicates that the task's gold answer includes a refusal option, and models are permitted to decline answering during evaluation. Refusal options include answers such as *"There is no answer.", "None of the above."*

| Dataset | Level 1 | | | | | Level-2 | | | Level-3 | | |
|---|---|---|---|---|---|---|---|---|---|---|---|
| | Extract | Local. | Comp. | Dur. Comp. | Ord. Comp. | Expl. Reason. | Ord. Reason. | Rel. Reason. | Co-temp. | Timeline | Counterf. |
| TIME-WIKI | $\mathcal{M}$ | $\mathcal{F}$ | $\mathcal{F}$ | $\mathcal{S}$ | $\mathcal{S}$ | $\mathcal{F}$ | $\mathcal{F}$ | $\mathcal{F}$ | $\mathcal{F}$ | $\mathcal{F}$ | $\mathcal{F}$ |
| TIME-NEWS | $\mathcal{M}$ | $\mathcal{F}$ | $\mathcal{F}$ | $\mathcal{S}$ | $\mathcal{S}$ | $\mathcal{S}$ | $\mathcal{S}$ | $\mathcal{S}$ | $\mathcal{S}$ | $\mathcal{F}$ | $\mathcal{S}$ |
| TIME-DIAL | $\mathcal{M}$ | $\mathcal{F}$ | $\mathcal{F}$ | $\mathcal{S}$ | $\mathcal{S}$ | $\mathcal{S}$ | $\mathcal{S}$ | $\mathcal{S}$ | $\mathcal{S}$ | $\mathcal{F}$ | $\mathcal{S}$ |

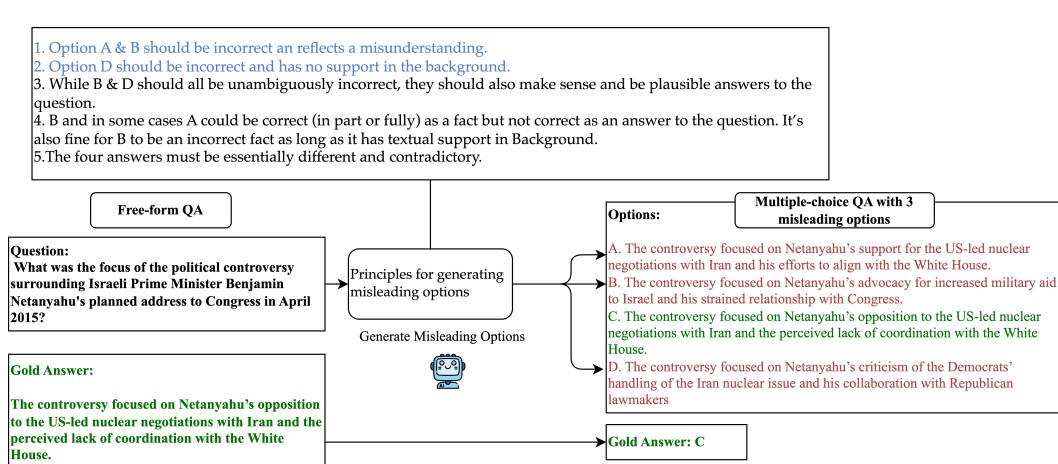

Figure 9: An illustration of the misleading option generation process. This example demonstrates a TCE question from TIME-NEWS regarding Netanyahu's planned address to Congress in April 2015, which falls under the explicit reasoning task category. Based on the gold answer from the free-form QA format, we employ an LLM to generate misleading options following our predefined principles, resulting in a single-choice QA format.

synthesis and various downstream applications. After evaluating both performance and cost considerations, we primarily employed `DeepSeek-V3` for synthesizing QA pairs across most tasks. However, for certain Level-2 and Level-3 tasks in the TIME-DIAL dataset, which involve processing numerous input instances and require complex reasoning capabilities, we utilized the advanced Large Reasoning Model `DeepSeek-R1` to ensure high-quality QA synthesis. Specifically, `DeepSeek-R1` was deployed for generating the `Order Reasoning`, `Relative Reasoning`, `Co-temporality`, and `Counterfactual` tasks within the TIME-DIAL dataset. Note that the DeepSeek-V3 used in data synthesis is the version released in December 2024.

**QA Formats**  The TIME dataset incorporates three distinct QA formats: **free-form**, **single-choice**, and **multiple-choice**. The free-form format requires short-answer generation, exemplified by questions like "*When did Nicola Agnozzi become an auxiliary bishop?*" with concise answers such as "*April 2, 1962.*" This format is particularly suitable for questions with limited synonymous answer variations, as demonstrated by temporal expressions which inherently possess low ambiguity. The single-choice format presents exactly one correct option among the provided choices, while the multiple-choice format may contain one or more correct options. For multiple-choice questions, we expect evaluated models to comprehensively identify all correct options in their responses.

**Misleading Options Generation**  Building upon the STARC framework [1] for generating diverse misleading options, we modify the principles for creating such distractors. The specific guidelines are illustrated in Figure 9. Each time we prompt the model, it generates three misleading options based on the given principles, the original question, and the gold answer. These options are designed to satisfy distinct types of misleading requirements according to the specified guidelines.

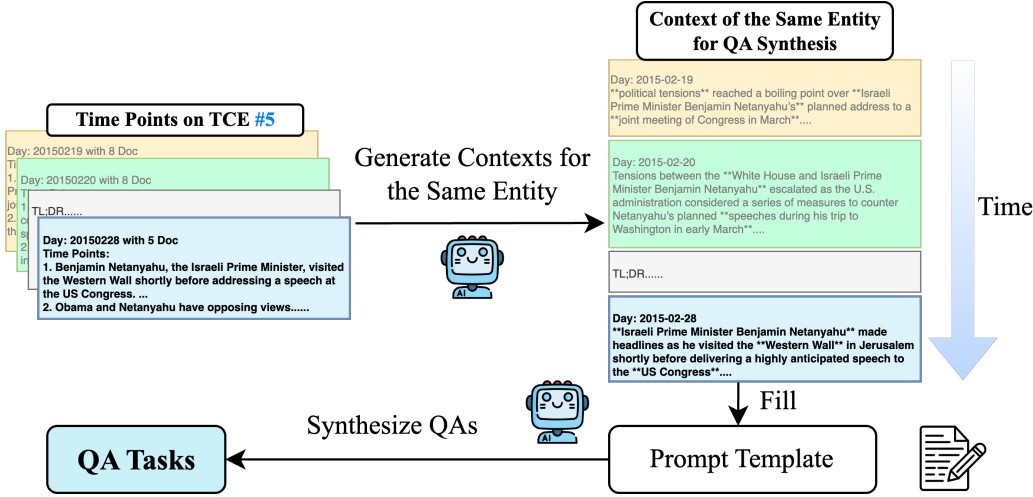

Figure 10: An illustration of the QA synthesis process based on time points in TIME-NEWS. We first utilize an LLM to generate contexts for the same entity, which are then organized into timeline contexts. Subsequently, for each QA task generation, we select relevant time points and their corresponding contexts to populate the prompt template, thereby synthesizing the QAs.

### A.4.1 Level 1: Basic Temporal Understanding and Retrieval

At this level, we primarily evaluate models' capabilities in temporal information retrieval and comprehension. We design tasks focusing on temporal information extraction, including directly extracting time expressions from context (`Extract`) and determining event occurrence times given specific events (`Localization`). Additionally, we assess fundamental temporal understanding through tasks such as duration calculation (`Computation`), duration comparison (`Duration Compare`), and event ordering (`Order Compare`).

**Extract**  For the `Extract` task, we first provide a set of authentic time points extracted from the source data. We then instruct the LLM to generate five novel time expressions that are distinct from all existing ones. Subsequently, we randomly select 0-4 authentic time points to form multiple-choice options, ensuring their randomness. This approach combines rule-based selection with LLM-generated distractors, thereby maintaining high QA quality. The task is formulated using a fixed question template: *Which of the following are time expressions mentioned in the context? (Note: There may be one or more correct options. If you think NONE of the time expressions in options A/B/C/D are mentioned, then you can choose E. Do not choose E together with other options.)*

> **Prompt for generating fake time expressions (`Extract`) (TIME-WIKI)**
>
> **[Rules:]** Given a list of time expressions, please generate FIVE new time expressions randomly. You should follow the instructions below:
> 1. The 5 new time expressions should totally different from all of the the given time expressions.
> 2. Each new time expression should be in the format of <Month> <Day>, <Year>. For example: "May 4, 1998", "April 1992", "1934".
> 3. The 5 new time expressions should closely resemble the given time expression in format and structure, creating a high level of confusion, yet they must represent entirely different times. For instance, you could alter the year while keeping the month and day unchanged, such as changing May 2, 1922 to May 2, 1923; or you could modify the month and day while retaining the same year, for example, transforming May 2, 1922 into July 2, 1922. Additionally, you might consider changing the day while keeping the month and year consistent, like adjusting May 2, 1922 to May 3, 1922. Another approach could involve altering the month and year but keeping the day the same, such as changing May 2, 1922 to May 2, 1921. Lastly, you could shift the entire date by a consistent interval, for example, moving May 2, 1922 to June 3, 1923, ensuring that each change introduces a subtle yet distinct variation from the original time expression.
> **[Example:]** Here are some examples showing the writing style. NOTE that the content of the examples are irrelevant to the question you will generate.
> **[Given time expressions:]** ['1656', '1763', '1782', '1784', '1787', '1815', 'September 20, 1817', 'January 20, 1848', ..., 'March 30, 2005', 'June 27, 1965', '1973', '2009', 'May 25, 1963', '1980', '2008', 'June 11, 1949', '1983', '1984', '2007', 'August 14, 2004', 'October 7, 2014', 'November 11, 2021', 'October 14, 2024', 'May 31, 1971', 'October 28, 1975']
> **[New 5 time expressions:]** ['1682', 'June 12, 1974', 'May 8, 1907', 'May 11, 1949', 'May 25, 2011']
> *Skipping 2 examples here......*
> **[Output:]** Now please write 5 new time expressions following the instructions and examples above. You should output the 5 new time expressions along with its answer, in the format of "["YY", "MM, YY", "MM DD, YY"]". NOTE that the time expressions should be in chronological order. Now the given time expressions are:
> **[Given time expressions:]** {Given_time_expressions}
> **[New 5 time expressions:]**

**Localization** We select three facts from the temporal knowledge graph and employ LLMs to generate corresponding QA pairs.

> **Prompt for generating QA. (`Localization`) (TIME-WIKI)**
>
> **[Rules:]** Given 3 facts, please generate one question along with its gold answer for each given fact. You should follow the instructions below:
>
> 1. The answer MUST be short and concise, avoiding using redundant words or repeating the information in the question.
>
> 2. You should output the question and its answer without any other explanation, such as "Question: xxx? Answer: xxx."
>
> 3. I will give you 3 facts in each line, and then you should output the question and its answer each line in the same sequence. So your output should be 3 lines of "Question: xxx? Answer: xxx.".
>
> 4. The question can be phrased in different ways, such as 'When is...?', 'What is the time for...?', 'What time did...?', and so on.
>
> 5. The answer should be "From xxx to xxx." or directly "xxx."
>
> **[Example:]** Here are some examples showing the writing style. NOTE that the content of the examples are irrelevant to the question you will generate.
>
> **\* [Given facts:]**
>
> Debra Hamel worked at Wesleyan University from 1998 to 2001.
>
> Oliver Marcy attended Wesleyan University in 1846.
>
> Debra Hamel completed her studies at Yale University in 1993.
>
> **[Generated QA:]**
>
> Question: When did Debra Hamel work at Wesleyan University? Answer: From 1998 to 2001.
>
> Question: What time did Oliver Marcy attend Wesleyan University? Answer: 1846.
>
> Question: What is the time for Debra Hamel to complete her studies at Yale University? Answer: 1993.
>
> *Skipping 1 example here......}*
>
> **[Output:] Now please write a question following the instructions and examples above. You should output the question along with its answer, in the format of "Question: xxx? Answer: xxx.". NOTE that the answer should be "From xxx to xxx." or directly "xxx."**
>
> **[Given facts:]**
>
> **{given_facts**
>
> **[Generated QA:]**

**Computation** We provide multiple pairs of temporal facts and utilize LLMs to generate corresponding questions. The temporal computations are systematically derived through script-based rules to ensure accuracy and consistency.

Prompt for generating QA. (`Computation`) (TIME-WIKI)

**[Rules:]** Each time, I will provide you with several pairs of text snippets, with each pair occupying one line. For each line containing a pair of text snippets, you need to generate a question. You should follow the instructions below:

1. The question should be based on the snippet pair.

2. Each text snippet pair includes two snippets. Each snippet is composed of a fact and a 'Happen/Begin/End' time. You should translate the snippets into a format like: ... ...

3. Please refer to the following examples and learn the patterns well.

**[Example:]** Here are one example showing the writing style. NOTE that the content of the examples are irrelevant to the question you will generate.

**[Snippet pairs:]**

José Manuel Pasquel served as an auxiliary bishop. Happen time: January 20, 1848. José Manuel Pasquel became a priest. Happen time: September 20, 1817.

Antonius Grech Delicata Testaferrata was ordained. Happen time: October 19, 1845. Diego Fabbrini played for Watford F.C.. Begin time: 2013.

Máximo Alcócer played for the Bolivia men's national football team. Begin time: 1957. Diego Fabbrini played for Udinese Calcio. 1944.

**[Questions:]**

What was the duration from the time José Manuel Pasquel became a priest until he served as an auxiliary bishop?

How long was it between Antonius Grech Delicata Testaferrata was ordained and Diego Fabbrini began to play for Watford F.C.?

How much time passed from Máximo Alcócer began to play for the Bolivia men's national football team to Diego Fabbrini end playing for Udinese Calcio?

**[Instruction:]**

Now please write a question following the instructions and examples above. You should output the question only for each line. NOTE that there is NO any prefix like "Question:", just output the question string.

**[Snippet pairs:]**

{snippet_pairs}

**[Questions:]**

**Duration Compare** The `Duration Compare` task is designed to evaluate models' capability in comparing the lengths of two time intervals. Specifically, we represent each time interval by the span between two distinct event timestamps. To construct questions for this task, we extract pairs of non-overlapping events from the timeline to form two comparable time intervals. A sample question template is: *"Which of the following two durations is longer? *Duration 1:* Between {fact1_1} and {fact1_2} *Duration 2:* Between {fact2_1} and {fact2_2}"* For the TIME-WIKI benchmark, we have developed alternative question templates based on temporal facts, including: *"Which fact lasted longer, Fact 1: {fact1} or Fact 2: {fact2}?", "Which of the two events, Fact 1: {fact1} or Fact 2: {fact2}, had a longer duration?", "Compare the duration of Fact 1: {fact1} and Fact 2: {fact2}. Which one was longer?"*

**Order Compare** The `Order Compare` task evaluates models' capability to comprehend temporal ordering between two time points. To avoid direct comparison of timestamps, we utilize event occurrence times as the comparison points. By leveraging a collection of temporal facts, we can directly select two facts and embed them into predefined question templates, as illustrated below.

"For Fact1: {fact1} and Fact2: {fact2}, which one happened earlier?"

"Which started earlier, Fact1: {fact1} or Fact2: {fact2}?"

"Which ended earlier, Fact1: {fact1} or Fact2: {fact2}?"

"Did Fact1: {fact1} start before Fact2: {fact2} ended?"

"Did Fact1: {fact1} end before Fact2: {fact2} started?"

"Did Fact1: {fact1} start before Fact2: {fact2} happened?"

### A.4.2 Level 2: Temporal Expression Reasoning

This level comprises three distinct subtasks that collectively assess the model's capability in performing multi-hop reasoning over temporal expressions. Each question can only be correctly answered if the LLM successfully conducts accurate multi-hop reasoning on the temporal expressions themselves.

**Explicit Reasoning** In this task, we employ temporal expressions composed of explicit time points in the questions. Notably, these explicit time points do not exist in the original context. We first randomly select temporal facts as ground truth answers, then transform their time points to generate non-existent temporal references. Finally, we utilize these modified time points to formulate questions. Below demonstrates the prompt template for generating `Explicit Reasoning` task questions in the TIME-WIKI dataset.

---

Generate Questions by Modifying Time Expressions (`Explicit Reasoning`) (TIME-WIKI)

**[Rules:]** Given the original questions and their corresponding updated time expressions, generate new questions by replacing the time expressions in the original questions. Follow these guidelines:
1. Modify only the time expressions; leave all other parts of the questions unchanged.
2. Output only the questions; do not include any answers.
3. Present each question on a separate line.
4. If the temporal expression in the given original question is a time point, such as "on September 9, 2002," then each time I will provide you with a time period expressed with "from ... to ...", for example, "from April 6, 1999 to June 2003." What you need to pay attention to is that your revised question should carry a tone of uncertainty. For instance, you should change "Which team did Ted play for on September 9, 2002?" to "Which team might Ted have played for from April 6, 1999 to June 2003?"
**[Example:]** Here are some examples showing the writing style. NOTE that the content of the examples are irrelevant to the question you will generate.
**[Original Question:]**
What position did Alexandre da Sagrada Família hold on August 8, 1782?
**[New Time Expression:]**
from April 1772, to June 1784
**[New Question:]**
What position have Alexandre da Sagrada Família held from April 1772, to June 1784?
*Skipping some examples...*

**[Output:]** Now please write a question following the instructions and examples above.
**[Original Question:]**
{original_question}
**[New Time Expression:]**
{new_time_expression}
**[New Question:]**

---

**Order Reasoning** This task evaluates the model's comprehension of ordinal temporal expressions, such as "the second time serving as a professional basketball player" or "the last time attending a ballet performance." Correctly interpreting these expressions requires the LLM to fully understand the timeline to accurately identify the specific time point referenced. For task construction, we first establish a timeline for the same entity. Then, we identify sub-timelines sharing the same factual relationship and select the k-th fact in chronological order to generate questions. Below are two prompt templates for question generation in TIME-WIKI, given specific facts and their corresponding timeline orders.

## Generate Question based on Subject-oriented Fact Timeline (`Order Reasoning`) (TIME-WIKI)

**[Rules:]** Given a sentence describing a simple fact and an order number, please generate one question along with its answer. You should follow these instructions:

1. The question MUST target the subject in the factual statement. For example, given "Bruno Aguiar plays for Portugal national under-21 football team from 2001 to 2004." with order number "3", generate "Who is the third person affiliated with Portugal national under-21 football team?" with answer "Bruno Aguiar".

2. Formulate questions exclusively based on the provided factual content and numerical order.

3. Exclude temporal expressions from generated questions while maintaining factual integrity.

4. Craft unambiguous questions using diverse interrogative structures (e.g., "Which individual...", "What entity...") that require contextual analysis rather than lexical matching.

5. Ensure answers contain only the factual subject without explanatory content or repetition.

6. Present results strictly as: "Question: xxx? Answer: xxx." with continuous formatting(e.g. with no line breaks).

**[Example:]** Here are some examples showing the writing style. NOTE that the content of the examples are irrelevant to the question you will generate.

**[Given fact:]**
Julius Babatunde Adelakun holds the position of diocesan bishop from April 13, 1973 to November 4, 2009.
**[Order number:]**
2
**[Generated QA:]**
Question: Who is the second person to hold the position of diocesan bishop? Answer: Julius Babatunde Adelakun.
*Skipping some examples here...*
**[Output:]** Now please write a question and its answer following the instructions and examples above. You should output the question along with its answer, in the format of "Question: xxx? Answer: xxx.". NOTE that the answer should be as short as possible.
**[Given fact:]**
{given_fact}
**[Order number:]**
{order_number}
**[Generated QA:]**

**[Rules:]** Given a sentence describing a simple fact and an order number, please generate one question along with its answer. You should follow the instructions below:
1. This question MUST be directed at the object in the fact. For example, given a fact "Bruno Aguiar plays for Portugal national under-21 football team from 2001 to 2004.", and an order number"3", your question should be "What was the third team Bruno Aguiar was affiliated with during her professional career?" and the answer should be "Portugal national under-21 football team".
2. The question should be derived directly from the factual content.
3. The question must exclude time expressions present in the original fact.
4. Phrase questions unambiguously using varied interrogative patterns (e.g., "Which team...", "What position...", "What organization...") while avoiding simple string matching.
5. The answer MUST contain only the factual object without explanations or repetitions.
6. Output strictly in the format: "Question: xxx? Answer: xxx." with no line breaks.
**[Example:]** Here are some examples showing the writing style. NOTE that the content of the examples are irrelevant to the question you will generate.
**[Given fact:]**
Julius Babatunde Adelakun holds the position of diocesan bishop from April 13, 1973 to November 4, 2009.
**[Order number:]**
2 **[Generated QA:]**
Question: What was the second position Julius Babatunde Adelakun held in his role? Answer: diocesan bishop.

**[Output:]** Now please write a question and its answer following the instructions and examples above. You should output the question along with its answer, in the format of "Question: xxx? Answer: xxx.". NOTE that the answer should be as short as possible.
**[Given fact:]**
{given_fact}
**[Order number:]**
{order_number}
**[Generated QA:]**

**Relative Reasoning** This task evaluates models' reasoning capabilities with relative temporal expressions. For instance, "within the last two weeks before Trump's second official election as US President" and "within 7 months and 2 days after Xiao Ming's official graduation" demonstrate temporal reasoning based on specific event anchors. Such expressions pose significant challenges to LLMs' temporal reasoning abilities. To construct QA pairs for this task, we first provide ground truth temporal fact statements along with their corresponding relative temporal expressions. Notably, these relative temporal expressions are pre-extracted from all temporal facts, each consisting of a factual statement and a relative temporal expression. The QA generation methodology is detailed in the following prompt examples.

**Generate Question based on Subject-oriented Fact Timeline (`Relative Reasoning`) (TIME-WIKI)**

**[Rules:]** Given a sentence describing a simple fact and a time expression, please generate one question. You should follow the instructions below:

1. The question MUST be directed at the subject in the fact. For example, given a fact "Bruno Aguiar plays for Portugal national under-21 football team.", and the time expression "Assuming today is January, 2002 | before today", your question should be "Assuming today is January, 2002, who was the most recently player that played for Portugal national under-21 football team before today?".

2. The question should come from the facts.

3. The question should be unambiguous and challenging, avoiding simple string matching. NO sub-questions allowed.

4. You should output the question without any other explanation. You should output the question directly.

**[Example:]** Here are some examples showing the writing style. NOTE that the content of the examples are irrelevant to the question you will generate.

**Example 1:**
**[Given fact:]**
Julius Babatunde Adelakun holds the position of diocesan bishop.
**[Time Expression:]**
Assuming today is January, 2002 | before today
**[Generated Question:]**
Assuming today is January 2002, who was the most recent person that held the position of diocesan bishop before today?

**Example 2:**
**[Given fact:]**
Gerolamo Castaldi holds the position of diocesan bishop.
**[Time Expression:]**
before Francesco Marmaggi works as a Catholic priest
**[Generated Question:]**
Who was the most recent person to hold the position of diocesan bishop before Francesco Marmaggi works as a Catholic priest?

**Example 3:**
**[Given fact:]**
Wlodzimierz Roman Juszczak holds the position of diocesan bishop.
**[Time Expression:]**
before September 9, 2009 | within the span of 3 years 2 months 28 days
**[Generated Question:]**
Who was the most recent person to hold the position of diocesan bishop before September 9, 2009, within the span of 3 years 2 months 28 days?

**[Output:]** Now please write a question following the instructions and examples above. You should directly output the question.
**[Given fact:]**
{given_fact}
**[Time Expression:]**
{time_expression}
**[Generated Question:]**

Generate Question based on Object-oriented Fact Timeline (`Relative Reasoning`) (TIME-WIKI)

**[Rules:]** Given a sentence describing a simple fact and a time expression, please generate one question. You should follow the instructions below:

1. The question MUST be directed at the object in the fact. For example, given a fact "Bruno Aguiar plays for Portugal national under-21 football team.", and the time expression "Assuming today is January, 2002 | before today", your question should be "Assuming today is January, 2002, which team did Bruno Aguiar play for?".

2. The question should come from the facts.

3. The question should be unambiguous and challenging, avoiding simple string matching. NO sub-questions allowed.

4. You should output the question without any other explanation. You should output the question directly.

**[Example:]** Here are some examples showing the writing style. NOTE that the content of the examples are irrelevant to the question you will generate.

**Example 1:**
**[Given fact:]**
Julius Babatunde Adelakun holds the position of diocesan bishop.
**[Time Expression:]**
Assuming today is January, 2002 | before today
**[Generated Question:]**
Assuming today is January, 2002, what position did Julius Babatunde Adelakun most recently hold before today?

**Example 2:**
**[Given fact:]**
Gerolamo Castaldi holds the position of diocesan bishop.
**[Time Expression:]**
before Francesco Marmaggi works as a Catholic priest
**[Generated Question:]**
What position did Gerolamo Castaldi most recently hold before Francesco Marmaggi worked as a Catholic priest?

**Example 3:**
**[Given fact:]**
Wlodzimierz Roman Juszczak holds the position of diocesan bishop.
**[Time Expression:]**
before September 9, 2009 | within the span of 3 years 2 months 28 days
**[Generated Question:]**
What position did Wlodzimierz Roman Juszczak most recently hold within the 3 years, 2 months, and 28 days prior to September 9, 2009?

**[Output:]** Now please write a question following the instructions and examples above. You should directly output the question.
**[Given fact:]**
{given_fact}
**[Time Expression:]**
{time_expression}
**[Generated Question:]**

### A.4.3 Level 3: Complex Temporal Relationship Reasoning

At this level, we focus on evaluating the model's capability to comprehend both implicit and explicit temporal relationships between events. Specifically, we assess the model from three perspectives: (1) temporal co-occurrence between events (`Co-temporality`), (2) complete reordering of multiple distinct events (`Timeline`), and (3) the model's simultaneous understanding of both the original

context and the question when temporal expressions in the question contradict the source text (`Counterfactual`).

**Co-temporality**    This task evaluates the model's ability to comprehend temporal co-occurrence between two events. For instance, the question "When Sam Altman co-founded OpenAI, what positions did Elon Musk hold?" implicitly assumes the temporal overlap between "Sam Altman founding OpenAI" and "Elon Musk serving as co-founder of OpenAI and CEO of Tesla and SpaceX". To construct questions for this task, we provide the LLM with two key elements: a condition fact that serves as the temporal reference, and a query fact that forms the basis for question generation. The following demonstrates the prompt template.

---

Generate Question based on Subject-oriented Fact Timeline (`Co-temporality`) (TIME-WIKI)

**[Rules:]** Given a condition fact and a query fact, please generate one question. You should follow these instructions:

1. The question MUST target the subject in the factual statement. For example, given the condition fact "Mauro Morelli holds the position of diocesan bishop." with query fact "William Weigand holds the position of diocesan bishop.", generate "When Mauro Morelli holds the position of diocesan bishop, who held the position of diocesan bishop?"

2. The question should come from the facts.

3. The question should be unambiguous and challenging, avoiding simple string matching. NO sub-questions allowed.

4. You should output the question without any other explanation. You should output the question directly.

**[Example:]** Here are some examples showing the writing style. NOTE that the content of the examples are irrelevant to the question you will generate.

**Example 1:**

**[Given fact:]**

Julius Babatunde Adelakun holds the position of diocesan bishop from April 13, 1973 to November 4, 2009.

**[Generated QA:]**

Question: Who served as diocesan bishop from April 13, 1973 to November 4, 2009?

Answer: Julius Babatunde Adelakun.

*Skipping the rest of the examples.*

**[Output:]** Now please write a question following the instructions and examples above. You should directly output the question.

**[Condition fact:]**

{condition_fact}

**[Query fact:]**

{query_fact}

**[Generated Question:]**

---

When Mauro Morelli held the position of diocesan bishop, what position did William Weigand hold?

**Timeline**  The Timeline task is designed to evaluate a model's ability to chronologically reorder multiple facts within a given context. In TIME-WIKI and TIME-DIAL, we assess the model's capability to determine temporal relationships among eight distinct facts, generating only one `Timeline` task question per timeline and context. In contrast, for TIME, we directly utilize multiple reordering questions from TCELongBench[58].

For constructing `Timeline` task questions, we employ existing timelines containing eight temporal facts (without explicit temporal expressions or timestamps). Using a Python program, we first compute the chronological order of events, then randomly shuffle the order of the eight facts, with the correct temporal sequence serving as the ground truth. In this question construction process, we solely rely on question templates to generate the final output.

The question template is as follows:

**Counterfactual**  To thoroughly evaluate models' understanding of temporal relationships in context, we modify the `Explicit Reasoning` task by counterfactually altering temporal expressions in questions, making them contradict the temporal information in the provided real context. During

evaluation, we instruct models to strictly adhere to the new temporal expressions in their responses. This approach eliminates direct reliance on contextual information (i.e., surface-level event-event correlations), enabling a fair assessment of models' genuine comprehension of temporal sequences.

Specifically, we construct task questions by directly modifying temporal expressions in `Explicit Reasoning` questions while preserving all other event details. After altering the temporal conditions, we employ a Python program with rule-based matching to determine whether the original answer remains valid under the new conditions. If the original answer no longer satisfies the new conditions, the correct answer becomes "There is no answer"; otherwise, the original gold answer remains unchanged.

We employ two distinct prompt strategies for question construction. The first prompt requires the model to generate a new temporal expression that replaces the original one while maintaining the same answer as before. The second prompt instructs the model to generate a new temporal expression that results in a different answer from the original.

---

Generate Question based on False Fact Premise and the new answer is the same as the original answer (`Counterfactual Question`) (TIME-WIKI)

**[Scenario:]** You are an annotator who is exceptionally skilled at generating false temporal facts and premises. First, you carefully comprehend the question I provided you and its corresponding correct answer. Based on the question and answer, you cleverly imagine an "if" clause that represents a hypothesis. This hypothesis must contradict the facts in the given context but must simultaneously ensure that the provided answer remains correct. Your hypothesis only needs to modify the temporal elements to differ from the original, such as altering the year, month, or day. You need to add the imagined "if" clause to the original question and only output the question itself with the added "if" clause.
**[Example:]** Here are some examples showing the writing style. NOTE that the content of the examples are irrelevant to the question you will generate.

**[Given info:]**
Question: Who might have worked as a Catholic priest before April 25, 1830?
Answer: Alexandre da Sagrada Família
**[Generated new question:]**
Who might have worked as a Catholic priest before April 25, 1830, if Alexandre da Sagrada Família worked as a Catholic priest in 1815?
*Skipping the rest of the examples...*

**[NOTE:]**
Now please write a question following the instructions and examples above. You should ONLY output the question, and there should be no other output.
**[Given info:]**
Context: {story}
Question: {question}
Answer: {answer}
**[Generated new question:]**

---

### A.5 Quality Control

For data sampling, we employed a systematic approach to ensure representativeness across all sub-datasets. Using a fixed random seed of 42, we randomly sampled 30-40 QA pairs from each task within the three sub-datasets (TIME-WIKI, TIME-NEWS, and TIME-DIAL). This process yielded a total of 1,071 QA pairs for annotation, with 352 from TIME-WIKI, 359 from TIME-NEWS, and 360 from TIME-DIAL (detailed sampling procedures are provided in Appendix A.5). To ensure annotation quality, we recruited annotators through professional forums and conducted rigorous qualification tests. From the initial pool of 8 professional annotators, we selected the top 3 performers based on both efficiency and quality metrics to establish our final human evaluation benchmark.

**Annotation Principles**    To ensure the quality of questions and answers, we established four annotation principles for evaluators to assess each QA pair systematically. For each data instance, we provide both the question and its corresponding gold answer. Evaluators are required to verify: (1) `Question Answerability` and (2) `Answer Correctness` (shown in ?) as fundamental criteria. Additionally, for the specialized `Timeline` and `Counterfactual` tasks, we introduced two specific evaluation principles: `Correctness Answer Ranking` and `Counterfactual Conditional Answer Correctness` respectively.

---

Checking Principle For `Question Answerability`

**Description:** Can the question be answered based on the provided context (without considering the standard answer provided above)?
**Options:**
[A]: Yes, it can be directly found and answered in the context.
[B]: Yes, but the question can only be fully answered through inference.
[C]: The question cannot be answered based on the context; the context does not provide relevant information.

---

Checking Principle For `Answer Correctness`

**Description:** Please double-check, is the standard answer correct according to the original text?
**Options:**
[A]: Yes, the answer is completely correct, answering the question fully and accurately.
[B]: Cannot be determined, but it has a certain possibility of being completely correct (perhaps you lack sufficient information to make a judgment).
[C]: The answer is not completely correct; it contains obvious errors or is irrelevant to the question.

---

Checking Principle For `Correctness Answer Ranking`

**Description:** Is the ranking of the answers correct? (Ranking: Is the order of events correct?)
**Options:**
[A]: Yes, the order of all events is accurate.
[B]: Incorrect, the order of events is not completely correct.

---

Checking Principle For `Counterfactual Conditional Answer Correctness`

**Description:** Determine 1. whether the hypothetical premise in the given question contradicts the facts in the original context. 2. Regarding the hypothesis, if we disregard whether it contradicts the context and only consider the given question itself, is the given reference answer correct according to the given original context?
**Options:**
[A]: Both 1 and 2 are met.
[B]: Only 1 is met, but 2 is not.
[C]: Neither 1 nor 2 is met.

---

**Worker Recruitment**    To ensure high-quality annotations, we established rigorous selection criteria for workers: (1) holding at least a bachelor's degree from accredited universities; (2) demonstrating English proficiency through TOEFL, IELTS, or equivalent certifications; (3) possessing strong information retrieval skills using search engines; and (4) successfully passing our customized Qualification Test (QT) for temporal reasoning tasks. These criteria guarantee annotators' competence in English comprehension and temporal reasoning capabilities. Regarding compensation, each worker received a base payment of $110, with an average rate of $0.103 per annotation instance.

**Qualification Test**    To ensure the quality of temporal reasoning annotations, we meticulously designed a Qualification Test (QT) to assess workers' capabilities in comprehending temporal

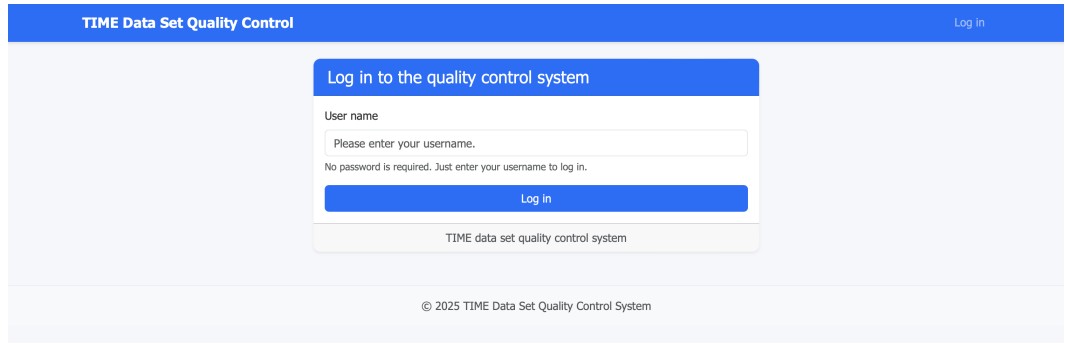

Figure 11: The login page of the annotation website.

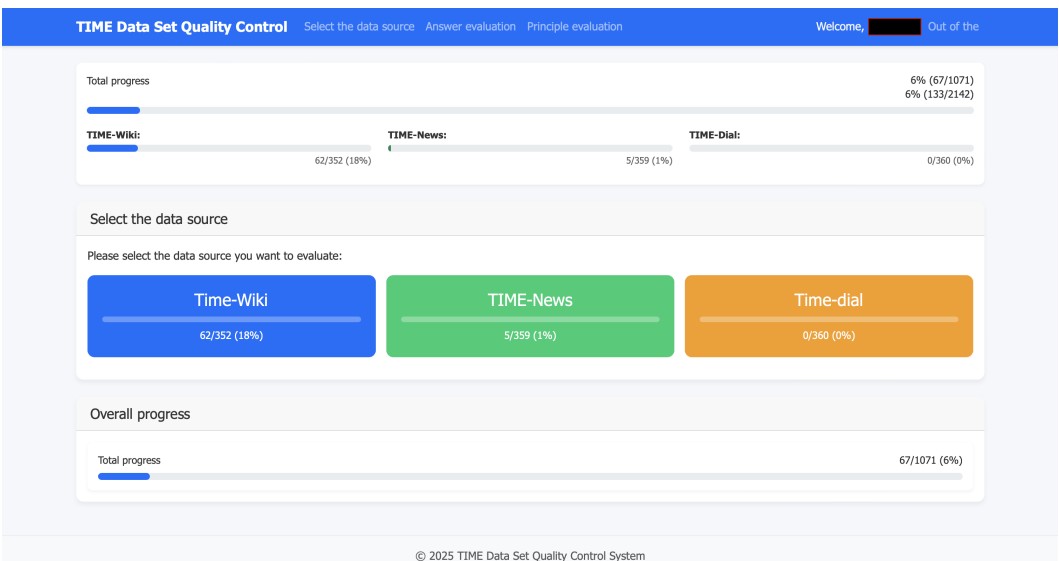

Figure 12: The overview page of the annotation website.

information and relationships across short, medium, and long contexts. Prior to administering the QT, we provided workers with comprehensive guidelines detailing the annotation requirements and evaluation criteria. Following the test administration, we manually reviewed all submissions to identify and exclude workers who demonstrated insufficient annotation proficiency. From the initial pool of 15 candidates, 8 successfully passed the QT and proceeded to the subsequent annotation phase.

**Two-stage Annotation**    We implemented a two-stage annotation process to ensure data quality. In the first stage, we collected annotators' responses as the human performance benchmark. In the second stage, annotators evaluated each QA pair based on predefined assessment principles. When annotators deemed a question unanswerable, they could select option C under the answerability principle, indicating that no possible answer could be inferred from the given context. The detailed annotation interface design is illustrated in Figure 11, 12, 13, 14.

**Annotation Quality Control**    Since our annotation platform uniformly employs a fill-in-the-blank format for manual answer annotation across both cloze and multiple-choice questions, we designed a novel annotation scheme to characterize inter-annotator agreement.

We adopt word-level similarity as the evaluation metric for annotation consistency. This method quantifies agreement by computing the lexical overlap between two annotators' responses to the same question. The approach not only measures the overall consistency level but also captures gradations of disagreement, thereby providing an objective basis for assessing annotation quality.

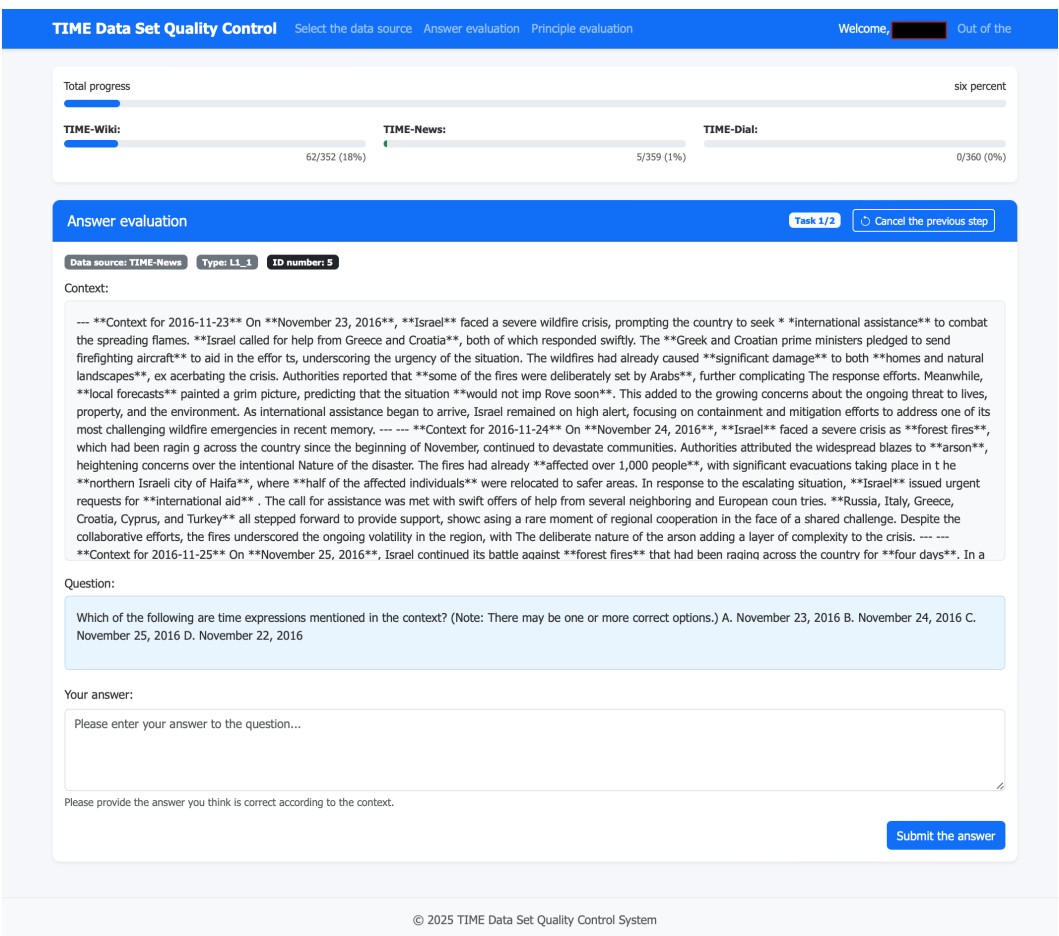

Figure 13: The first stage of data annotation.

First, we automatically pair JSON files following identical paths from multiple annotators' datasets. For each file pair, we extract the manually annotated answers (Human Prediction Answer) corresponding to the same question ID. The word-level similarity is calculated using the Jaccard similarity coefficient from set theory, defined as:

$$Sim(A, B) = \frac{|A \cap B|}{|A \cup B|}$$

This metric yields values in the range [0, 1], where 1 indicates perfect agreement and 0 denotes complete disagreement. Based on word-level similarity, we compute the following evaluation metrics: Average Word-level Agreement: The mean word-level similarity across all question-answer pairs Exact Match Rate: The proportion of samples with perfect word-level similarity (1.0) Agreement Distribution: The sample distribution across similarity intervals: [0-0.2, 0.2-0.4, 0.4-0.6, 0.6-0.8, 0.8-1.0] Quartiles: The Q1, median, and Q3 values of word-level similarity.

## A.6 Dataset Statistics

We present comprehensive statistics of instance counts for each subtask in both TIME and TIME-LITE datasets, as detailed in Table 10.

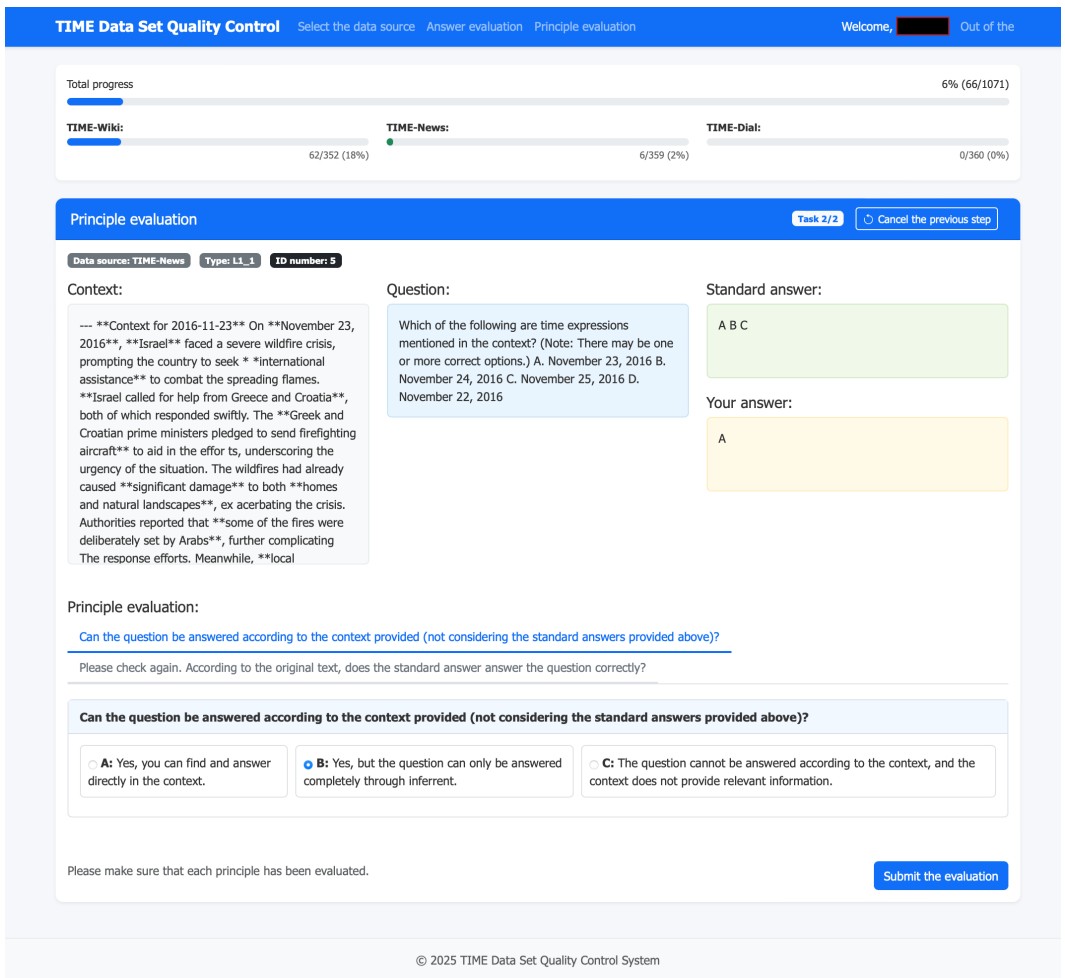

Figure 14: The second stage of data annotation.

Table 10: Dataset statistics. The table displays the number of instances for each dataset and task category. Task abbreviations are: Ext. (Extract), Loc. (Localization), Comp. (Computation), D.C. (Duration Compare), O.C. (Order Compare); E.R. (Explicit Reasoning), O.R. (Order Reasoning), R.R. (Relative Reasoning); C.T. (Co-temporality), T.L. (Timeline), C.F. (Counterfactual).

| Dataset | All Tasks | Level 1 | | | | | Level 2 | | | Level 3 | | |
|---|---|---|---|---|---|---|---|---|---|---|---|---|
| | | Ext. | Loc. | Comp. | D.C. | O.C. | E.R. | O.R. | R.R. | C.T. | T.L. | C.F. |
| **TIME** | **38522** | 1480 | 3546 | 3376 | 3401 | 3549 | 3537 | 3538 | 3537 | 3513 | 5508 | 3537 |
| TIME-WIKI | 13848 | 1261 | 1299 | 1126 | 1151 | 1299 | 1287 | 1288 | 1287 | 1263 | 1300 | 1287 |
| TIME-NEWS | 19958 | 0 | 1800 | 1800 | 1800 | 1800 | 1800 | 1800 | 1800 | 1800 | 3758 | 1800 |
| TIME-DIAL | 4716 | 219 | 447 | 450 | 450 | 450 | 450 | 450 | 450 | 450 | 450 | 450 |
| **TIME-LITE** | **943** | 60 | 90 | 78 | 86 | 90 | 90 | 90 | 90 | 90 | 89 | 90 |
| TIME-LITE-WIKI | 322 | 30 | 30 | 24 | 28 | 30 | 30 | 30 | 30 | 30 | 30 | 30 |
| TIME-LITE-NEWS | 299 | 0 | 30 | 30 | 30 | 30 | 30 | 30 | 30 | 30 | 29 | 30 |
| TIME-LITE-DIAL | 322 | 30 | 30 | 24 | 28 | 30 | 30 | 30 | 30 | 30 | 30 | 30 |

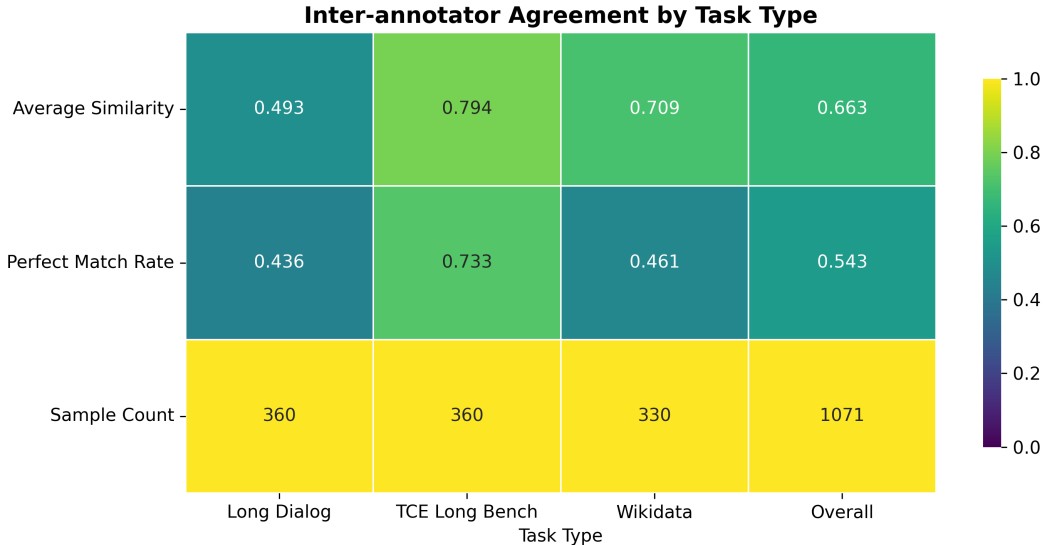

Figure 15: Task Type Comparison Heatmap: A comparison of annotation consistency across three distinct task types (TIME-WIKI, TIME-NEWS, and TIME-DIAL). The heatmap employs a color gradient from dark to light to visually represent the differences in average similarity and exact match rates among the task types.

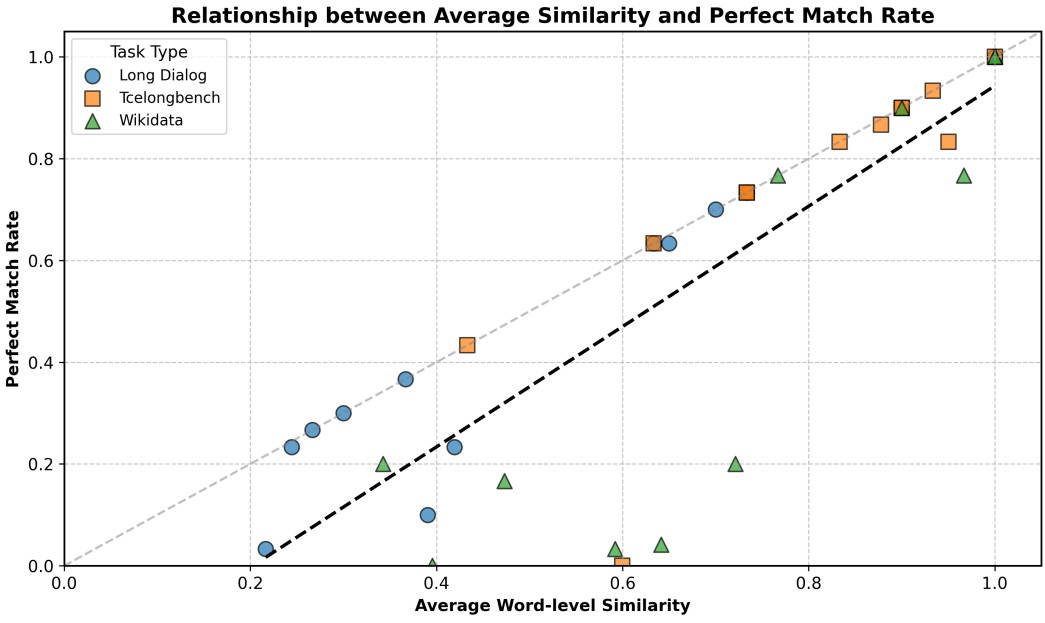

Figure 16: Scatter Plot of Average Similarity vs. Exact Match Rate: This plot illustrates the relationship between the average token-level similarity and the exact match rate for each file, with different colors and markers distinguishing different task types. A regression line is added to indicate the correlation between the two metrics.

Table 11: Average context token counts for each dataset and task category. Token counts are derived using the 'cl100k_base' encoder from the 'tiktoken' library. Task abbreviations are: Ext. (Extract), Loc. (Localization), Comp. (Computation), D.C. (Duration Compare), O.C. (Order Compare); E.R. (Explicit Reasoning), O.R. (Order Reasoning), R.R. (Relative Reasoning); C.T. (Co-temporality), T.L. (Timeline), C.F. (Counterfactual). A dash (—) indicates that data was not available for the corresponding combination. Note that for TIME-NEWS and TIME-LITE-NEWS, the context token counts represent the average token counts of the top-3 chunks retrieved by three distinct retrievers.

| Dataset | Level 1 | | | | | Level 2 | | | Level 3 | | |
|---|---|---|---|---|---|---|---|---|---|---|---|
| | Ext. | Loc. | Comp. | D.C. | O.C. | E.R. | O.R. | R.R. | C.T. | T.L. | C.F. |
| **TIME** | | | | | | | | | | | |
| TIME-WIKI | 1157.39 | 1156.37 | 1159.06 | 1188.26 | 1156.37 | 1155.05 | 1150.47 | 1150.58 | 1158.69 | 1156.00 | 1155.05 |
| TIME-NEWS | — | 1473.98 | 1568.53 | 1527.80 | 1499.17 | 1502.02 | 1511.58 | 1496.59 | 1441.80 | 1561.75 | 1515.70 |
| TIME-DIAL | 20862.69 | 20709.06 | 20667.67 | 20667.67 | 20667.67 | 20667.67 | 20667.67 | 20667.67 | 20667.67 | 20667.67 | 20667.67 |
| **TIME-LITE** | | | | | | | | | | | |
| TIME-LITE-WIKI | 1332.37 | 1332.37 | 1284.88 | 1328.86 | 1332.37 | 1332.37 | 1332.37 | 1332.37 | 1332.37 | 1332.37 | 1332.37 |
| TIME-LITE-NEWS | — | 1498.64 | 1572.68 | 1500.36 | 1425.72 | 1361.89 | 1494.90 | 1458.50 | 1490.09 | 1523.03 | 1423.47 |
| TIME-LITE-DIAL | 21665.93 | 21665.93 | 21665.93 | 21665.93 | 21665.93 | 21665.93 | 21665.93 | 21665.93 | 21665.93 | 21665.93 | 21665.93 |

# B    Benchmark Details

## B.1    QA Examples

### B.1.1    TIME-WIKI

---

QA Example. (Extract) (TIME-WIKI) (Level 1: Basic Temporal Understanding and Retrieval)

**Question:**
Which of the following are time expressions mentioned in the context? (Note: There may be one or more correct options. If you think NONE of the time expressions in options A/B/C/D are mentioned, then you can choose E. Do not choose E together with other options.)
A. 1930
B. 2011
C. 1929
D. 1932
E. None of the above.
**Answer:**
A B

---

QA Example. (Localization) (TIME-WIKI) (Level 1: Basic Temporal Understanding and Retrieval)

**Question:**
When did Nicola Agnozzi become an auxiliary bishop?
**Answer:**
April 2, 1962

---

QA Example. (Computation) (TIME-WIKI) (Level 1: Basic Temporal Understanding and Retrieval)

**Question:**
What was the duration from the time George Omaira began to serve as an auxiliary bishop until Nicola Agnozzi became an auxiliary bishop? (Hint: Please answer in the form of Month Day, Year. e.g. 1 year 2 months 3days, or 2 days, or 9 months, or 3 months 15 days.)
**Answer:**
362 years

---

**QA Example.** (`Duration Compare`) (TIME-WIKI) (Level 1: Basic Temporal Understanding and Retrieval)

**Question:**
Which of the two events, Fact 1: George Omaira served as an auxiliary bishop. or Fact 2: Maxim Hermaniuk served as an auxiliary bishop., had a longer duration?
A. Fact 1 lasts longer.
B. Fact 2 lasts longer.
C. They last almost the same amount of time.
**Answer:**
A

---

**QA Example.** (`Order Compare`) (TIME-WIKI) (Level 1: Basic Temporal Understanding and Retrieval)

**Question:**
For Fact1: Mauro Morelli was appointed as both a titular bishop and an auxiliary bishop. and Fact2: Pedro Bantigue y Natividad was appointed as an auxiliary bishop., which one happened earlier?
A. Fact 1 happened earlier.
B. Fact 2 happened earlier.
C. They happen at almost the same time.
**Answer:**
B

---

**QA Example.** (`Explicit Reasoning`) (TIME-WIKI) (Level 2: Temporal Expression Reasoning)

**Question:**
Who might have held the position of auxiliary bishop from 1902 to February 31, 1916?
**Answer:**
Edward Joseph Hanna

---

**QA Example.** (`Order Reasoning`) (TIME-WIKI) (Level 2: Temporal Expression Reasoning)

**Question:**
What was the third position Mauro Morelli held in his role?
**Answer:**
auxiliary bishop

---

**QA Example.** (`Relative Reasoning`) (TIME-WIKI) (Level 2: Temporal Expression Reasoning)

**Question:**
Who was the most recent person to hold the position of titular bishop before Mauro Morelli stepped down as diocesan bishop?
**Answer:**
José Antonio Eguren

QA Example. (`Co-temporality`) (TIME-WIKI) (Level 3: Complex Temporal Relationship Reasoning)

**Question:**
When José Antonio Eguren held the position of auxiliary bishop, what position did Mauro Morelli hold?
**Answer:**
diocesan bishop

QA Example. (`Timeline`) (TIME-WIKI) (Level 3: Complex Temporal Relationship Reasoning)

**Question:**
Below are 8 facts. You need to sort these facts in chronological order. Requirements: Your output format must be numbers enclosed in parentheses without any other symbols or whitespace. For example: (1)(5)(2)(7)(3)(8)(6)(4)

(1) Mauro Morelli holds the position of auxiliary bishop.
(2) Mauro Morelli holds the position of titular bishop.
(3) Jean-Claude Miche holds the position of titular bishop.
(4) Belchior Carneiro Leitão holds the position of titular bishop.
(5) Timothy Norton holds the position of titular bishop.
(6) Nicola Agnozzi holds the position of titular bishop.
(7) Edward Joseph Hanna holds the position of titular bishop.
(8) John Joseph Swint holds the position of auxiliary bishop.
**Answer:**
(4)(3)(7)(8)(6)(2)(1)(5)

QA Example. (`Counterfactual`) (TIME-WIKI) (Level 3: Complex Temporal Relationship Reasoning)

**Question:**
Which institutions might Victorina Durán have attended from 1915 to 1934, if she graduated from the Madrid Royal Conservatory in 1915?
**Answer:**
Royal Academy of Fine Arts of San Fernando

## B.1.2 TIME-NEWS

QA Example. (`Localization`) (TIME-NEWS) (Level 1: Basic Temporal Understanding and Retrieval)

**Question:**
When is Israeli Prime Minister Benjamin Netanyahu scheduled to address the US Congress?
**Answer:**
March 3, 2015

QA Example. (Computation) (TIME-NEWS) (Level 1: Basic Temporal Understanding and Retrieval)

**Question:**
How many days passed between the initial opposition to Netanyahu's speech on February 19, 2015, and the announcement of his address to Congress on February 27, 2015? (Hint: Please answer in the form of Month Day, Year. e.g. 1 year 2 months 3days, or 2 days, or 9 months, or 3 months 15 days.)
**Answer:**
8 days

QA Example. (Duration Compare) (TIME-NEWS) (Level 1: Basic Temporal Understanding and Retrieval)

**Question:**
Which of the following two durations is longer? *Duration 1:* The time span between when political tensions escalated over Netanyahu's planned address to Congress and when the White House announced measures to counter Netanyahu's speeches. *Duration 2:* The time span between when the White House considered limiting communication with Israel and when the White House began strategizing a response to Netanyahu's upcoming speech.
A. Duration 1 is longer.
B. Duration 2 is longer.
C. The two durations are approximately the same length. **Answer:**
B

QA Example. (Order Compare) (TIME-NEWS) (Level 1: Basic Temporal Understanding and Retrieval)

**Question:**
For Fact1: Democratic Party representatives echoed calls to postpone Netanyahu's address. and Fact2: The White House explored strategies to counter Netanyahu's speeches during his trip to Washington., which one happened earlier?
A. Fact 1 happened earlier.
B. Fact 2 happened earlier.
C. They happen at almost the same time.
**Answer:**
A

QA Example. (Explicit Reasoning) (TIME-NEWS) (Level 2: Temporal Expression Reasoning)

**Question:**
What was the focus of the political controversy surrounding Israeli Prime Minister Benjamin Netanyahu's planned address to Congress in April 2015?
A. The controversy focused on Netanyahu's support for the US-led nuclear negotiations with Iran and his efforts to align with the White House.
B. The controversy focused on Netanyahu's advocacy for increased military aid to Israel and his strained relationship with Congress.
C. The controversy focused on Netanyahu's opposition to the US-led nuclear negotiations with Iran and the perceived lack of coordination with the White House.
D. The controversy focused on Netanyahu's criticism of the Democrats' handling of the Iran nuclear issue and his collaboration with Republican lawmakers **Answer:**
C

## QA Example. (`Order Reasoning`) (TIME-NEWS) (Level 2: Temporal Expression Reasoning)

**Question:**
What was the first action taken by liberal Democrats in response to Netanyahu's planned address to Congress?
A. They issued a public statement criticizing the speech.
B. They held a press conference to announce their support for the speech.
C. They signed a letter requesting the delay of the speech.
D. They organized a protest against the speec
**Answer:**
C

## QA Example. (`Relative Reasoning`) (TIME-NEWS) (Level 2: Temporal Expression Reasoning)

**Question:**
Who publicly reaffirmed their endorsement of Netanyahu on February 21, 2015, within the context of his upcoming speech to Congress?
A. Susan Rice, Samantha Power.
B. Hillary Clinton, Bernie Sanders.
C. Barack Obama, Joe Biden.
D. Jeb Bush, Ted Cruz
**Answer:**
D

## QA Example. (`Co-temporality`) (TIME-NEWS) (Level 3: Complex Temporal Relationship Reasoning)

**Question:**
At the same time as Israeli Prime Minister Benjamin Netanyahu's planned address to a joint meeting of Congress, what did liberal Democrats formally request?
A. the endorsement of Netanyahu's address
B. the rescheduling of Netanyahu's address
C. the cancellation of Netanyahu's address
D. the delay of Netanyahu's address
**Answer:**
D

## QA Example. (`Timeline`) (TIME-NEWS) (Level 3: Complex Temporal Relationship Reasoning)

**Question:**
Below are 3 facts. You need to sort these facts in chronological order. Requirements: You must output a sequence of uppercase letters separated by commas, such as 'A,B,C', without any other characters.

A. Democratic leaders expressed concern about the announcement of the Israeli Prime Minister's speech to the joint meeting of the House of Representatives and the Senate without consulting neither the White House nor the Democrats.
B. A number of Democrats have expressed opposition to Israeli Prime Minister Benjamin Netanyahu's planned address to a joint meeting of Congress in March.
C. Some Democrats criticize Benjamin Netanyahu, who is m̈istaken to agree to iẗänd p̈laying politics with the critical issue of Israel's security.
**Answer:**
B,A,C

QA Example. (Counterfactual) (TIME-NEWS) (Level 3: Complex Temporal Relationship Reasoning)

**Question:**
If no liberal Democrats signed the letter requesting the postponement of Netanyahu's address to Congress from January 2015 to March 2015, who signed the letter?
A. The National Iranian American Council drafted the letter.
B. A bipartisan group of lawmakers initiated the letter.
C. Conservative Democrats signed the letter.
D. There is no answer
**Answer:**
D

### B.1.3   TIME-DIAL

QA Example.  (Extract) (TIME-DIAL) (Level 1:  Basic Temporal Understanding and Retrieval)

**Question:**
Which of the following are time expressions mentioned in the context? (Note: There may be one or more correct options. And the time expressions are mentioned directly or indirectly in the context.)
A. April 17, 2021
B. 2018
C. March 16, 2020
D. March 14, 2019
**Answer:**
C

QA Example. (Localization) (TIME-DIAL) (Level 1: Basic Temporal Understanding and Retrieval)

**Question:**
When is Debra Ryan working on starting her own business?
**Answer:**
8:35 pm, February 21, 2020

QA Example. (Computation) (TIME-DIAL) (Level 1: Basic Temporal Understanding and Retrieval)

**Question:**
How long was it between Debra Ryan going skydiving and India Brown attending a street art fest in Brazil? (Please answer using natural time expressions that combine appropriate units based on duration length, e.g. 2̈ months 4 daysf̈or 64 days or 6̈4 daysf̈or shorter spans)
**Answer:**
19 days

**QA Example.** (`Duration Compare`) (TIME-DIAL) (Level 1: Basic Temporal Understanding and Retrieval)

**Question:**
Which of the following two durations is longer? *Duration 1:* Between Debra Ryan is learning to play the guitar. and Debra Ryan visited Adventure Land during the weekend trip. *Duration 2:* Between Debra Ryan rode a roller coaster called T̈he Wild Rideät Adventure Land. and India Brown found flowers by a lake in the park.
A. Duration 1 is longer.
B. Duration 2 is longer.
C. The two durations are approximately the same length.

**Answer:**
A

---

**QA Example.** (`Order Compare`) (TIME-DIAL) (Level 1: Basic Temporal Understanding and Retrieval)

**Question:**
For Fact1: India Brown became a Queen fan. and Fact2: India Brown found flowers by a lake in the park., which one happened earlier?
A. Fact 1 happened earlier.
B. Fact 2 happened earlier.
C. They happen at almost the same time.

**Answer:**
A

---

**QA Example.** (`Explicit Reasoning`) (TIME-DIAL) (Level 2: Temporal Expression Reasoning)

**Question:**
What notable artistic or outdoor activities did India Brown participate in between April 1, 2020, and April 9, 2020?
A. India Brown attended a street art fest in Brazil.
B. India Brown took a photo of a feather and shells on a beach.
C. India Brown went hiking and sketching at a nearby national park.
D. India Brown received positive feedback on her artwork
**Answer:**
B

---

**QA Example.** (`Order Reasoning`) (TIME-DIAL) (Level 2: Temporal Expression Reasoning)

**Question:**
What was India Brown's third teaching engagement in 2020?
A.Running a painting workshop for kids.
B.Teaching art at an orphanage in Cambodia.
C.Conducting a live demonstration for her college art club.
D.Instructing a pottery class at a local studi
**Answer:**
A

QA Example. (`Relative Reasoning`) (TIME-DIAL) (Level 2: Temporal Expression Reasoning)

**Question:**
What was India Brown's most recent job before 12:00 am, March 09, 2020?
A.India Brown is working on a new series of abstract artworks based on her trip.
B.India Brown is working as a travel guide based on her trip experiences.
C.India Brown is working on a new painting technique learned at a street art festival.
D.India Brown is testing watercolors for her new series of abstract artworks
**Answer:**
A

QA Example. (`Co-temporality`) (TIME-DIAL) (Level 3: Complex Temporal Relationship Reasoning)

**Question:**
At the same time as Debra Ryan is learning to play the guitar, what collection does India Brown have?
A.India Brown has a collection of soap sculptures.
B.India Brown has a collection of watercolor paintings.
C.India Brown has a collection of CDs.
D.India Brown has a collection of vinyl records
**Answer:**
C

QA Example. (`Timeline`) (TIME-DIAL) (Level 3: Complex Temporal Relationship Reasoning)

**Question:**
Below are 8 facts. You need to sort these facts in chronological order. Requirements: Your output format must be numbers enclosed in parentheses without any other symbols or whitespace. For example: (1)(5)(2)(7)(3)(8)(6)(4)

(1) India Brown is working on a new painting technique learned at a street art festival.
(2) India Brown shared an image of a mural made by kids.
(3) India Brown had her first art show at a local gallery.
(4) India Brown became a Queen fan.
(5) India Brown got invited to exhibit at a local gallery.
(6) India Brown took a photo of a feather and shells on a beach.
(7) India Brown sketched a waterfall during a hike.
(8) India Brown received positive feedback on her artwork.

**Answer:**
(4)(5)(1)(7)(6)(2)(8)(3)

## C Experiment Details

### C.1 Models

#### C.1.1 Vanilla Models

We primarily evaluate vanilla models, including the base and instruction-tuned versions of Qwen2.5 series [52, 53], the instruction-tuned Llama-3.1 model [10], as well as state-of-the-art models such as Deepseek-V3 [8] and GPT-4o [17]. These models are evaluated without any test-time computation scaling specifically designed to enhance their reasoning capabilities.

#### C.1.2 Test-time Scaled Models

We primarily select Deepseek-R1[7], OpenAI o3-mini, QwQ-32B[40], and Deepseek-R1 Distilled Models[7] as our test-time scaled models. These models enhance their logical reasoning capabilities through reinforcement learning or direct distillation from advanced test-time scaled models. They demonstrate strong reasoning performance not only in mathematical and coding domains but also exhibit generalizable reasoning abilities across diverse fields.

**Deepseek-R1 Distilled Models**  We conduct experiments using distilled models from Deepseek-R1 [7], specifically Deepseek-R1-Distilled-Qwen-7B, Deepseek-R1-Distilled-Qwen-14B, Deepseek-R1-Distilled-Qwen-32B, Deepseek-R1-Distill-Llama-8B, and Deepseek-R1-Distill-Llama-70B. These models are derived through knowledge distillation from Deepseek-R1, with their base architectures being Qwen2.5-Math-7B [54], Qwen2.5-14B, Qwen2.5-32B [52], Llama-3.1-8B, and Llama-3.3-70B-Instruct [10], respectively.

### C.2 Evaluation Metrics

We employ distinct evaluation metrics tailored to different task formats, as detailed in Table 9. Following the evaluation protocol established in [6], we adopt the following metrics: (1) For free-form QA tasks, we utilize token-level Exact Match (EM) and F1 scores. Specifically, we apply token-level EM for the `Timeline` task, while employing token-level F1 for other free-form QA tasks. (2) For single-choice and multiple-choice questions, we implement option-level F1 scores, with a particular focus on macro option-level F1 to ensure comprehensive evaluation across all options.

**Token-level Exact Match and F1 Score**  The token-level Exact Match is a binary metric that evaluates the complete match between predicted and ground truth answers:

$$\text{Exact Match} = \begin{cases} 1.0, & \text{if } \text{pred\_answer.lower().strip()} = \text{gold\_answer.lower().strip()} \\ 0.0, & \text{otherwise} \end{cases}$$

This strict metric assigns a score of 1 only when the predicted answer exactly matches the ground truth (ignoring case and leading/trailing whitespace), and 0 otherwise.

The token-level F1 Score measures answer similarity through lexical overlap between predicted and ground truth answers, computed as follows:

1. Tokenization and normalization: Convert answers to lowercase, remove punctuation, and tokenize by whitespace.

2. Calculate shared tokens:

$$c = \sum_{t \in \text{tokens}} \min(\text{freq}_{\text{gold}}(t), \text{freq}_{\text{pred}}(t))$$

where $t$ represents unique tokens, and $\text{freq}_{\text{gold}}(t)$ and $\text{freq}_{\text{pred}}(t)$ denote the frequency of token $t$ in ground truth and predicted answers respectively.

3. Compute precision:

$$\text{precision} = \frac{c}{|\text{pred\_tokens}|}$$

4. Compute recall:

$$\text{recall} = \frac{c}{|\text{gold\_tokens}|}$$

5. Calculate F1 score:

$$\text{F1} = \frac{2 \times \text{precision} \times \text{recall}}{\text{precision} + \text{recall}}$$

with F1 = 0 when precision + recall = 0.

This lexical overlap-based F1 score captures partially correct answers, making it more lenient than Exact Match.

**Option-level F1 Score**   For multiple-choice questions, the Option-level F1 score evaluates the match between predicted and ground truth options:

1. Extract options: Normalize options (e.g., "A B C" or "A,B,C") into standardized option sets.

2. Compute confusion matrix:

$$\text{TP} = |\text{pred\_options} \cap \text{gold\_options}|$$
$$\text{FP} = |\text{pred\_options} - \text{gold\_options}|$$
$$\text{FN} = |\text{gold\_options} - \text{pred\_options}|$$

3. Calculate precision and recall:

$$\text{precision} = \frac{\text{TP}}{|\text{pred\_options}|}$$
$$\text{recall} = \frac{\text{TP}}{|\text{gold\_options}|}$$

4. Compute Option-level F1 score:

$$\text{pair\_level\_f1} = \begin{cases} \frac{2 \times \text{precision} \times \text{recall}}{\text{precision} + \text{recall}}, & \text{if } \text{pred\_options} \subseteq \text{gold\_options} \\ 0, & \text{if } \exists x \in \text{pred\_options} : x \notin \text{gold\_options} \end{cases}$$

**Macro and Micro F1 Scores**   The Macro F1 Score averages F1 scores across all questions:

$$\text{Macro F1} = \frac{1}{n} \sum_{i=1}^{n} \text{F1}_i$$

where $n$ is the total number of questions and $\text{F1}_i$ is the F1 score for the $i$-th question. This approach assigns equal weight to each question, making it robust to imbalanced option distributions across the dataset.

In contrast, Micro F1 aggregates all TP, FP, and FN across questions before computing the overall score:

$$\text{micro\_precision} = \frac{\sum_{i=1}^{n} \text{TP}_i}{\sum_{i=1}^{n} (\text{TP}_i + \text{FP}_i)}$$
$$\text{micro\_recall} = \frac{\sum_{i=1}^{n} \text{TP}_i}{\sum_{i=1}^{n} (\text{TP}_i + \text{FN}_i)}$$
$$\text{micro\_f1} = \frac{2 \times \text{micro\_precision} \times \text{micro\_recall}}{\text{micro\_precision} + \text{micro\_recall}}$$

## C.3 Retriever for Evaluating TIME-NEWS

We employ three distinct retrievers, each retrieving the top-3 text chunks based on the given question, with a maximum of 500 words per chunk.

**BM25**    BM25 is a bag-of-words retrieval model that computes relevance scores based on term frequency (TF), inverse document frequency (IDF), and document length, without considering word order. It improves upon traditional TF calculation by preventing unbounded growth and introduces two key parameters: a document length normalization parameter (typically b) and a TF saturation parameter (typically k1) for finer score adjustment. The primary strength of BM25 lies in its effective handling of keyword matching and its ability to assign appropriate weights to both common and rare terms.

**Vector**    The core of vector retrieval lies in high-quality text embedding models that capture deep semantic information. Unlike keyword-based methods such as BM25, vector retrieval excels at handling synonyms, near-synonyms, and complex semantic relationships, enabling it to retrieve documents that are semantically relevant to the query even when they do not contain exact keyword matches. We employ the state-of-the-art BGE-M3 [3] text embedding model as our vector retriever.

**Hybrid**    The hybrid retrieval approach aims to combine the strengths of keyword-based retrieval (e.g., BM25) and semantic vector retrieval to achieve superior performance compared to individual methods, typically enhancing both recall and accuracy. Specifically, we first conduct initial retrieval by invoking both BM25 and vector retrievers to obtain the top-5 candidate results for each query, followed by result merging and deduplication. Our merging strategy prioritizes BM25 results while supplementing with unique results from vector retrieval. Subsequently, we perform document re-ranking using BGE-Reranker-Base [49]. Finally, we select the top-3 documents from the (re-)ranked candidates. Notably, before generating chunks, we sort documents in ascending order by date. This temporal sorting may override previous relevance-based rankings (whether from BM25, vector similarity, or reranker scores), with its impact contingent on whether the application prioritizes timeliness.

**Computation Resource**    All experiments are done on 4 NVIDIA A800 GPUs with 80GB memory for each GPU.

## C.4 Prompt Templates for Evaluation

---

Evaluation Prompt Template for **free-form** tasks (exluding `Counterfactual` and `Computation`)

**Context:** {context}
You need to answer the following question based on the given context. If you can infer the answer from the context, please output your answer directly, keeping it concise and accurate, without any explanatory text. **If you are certain there are multiple answers in the context that satisfy the question, please output all answers, one per line (i.e., separate each answer with a line break).** And you will never refuse to answer any question.

**Question:** {question}
Therefore, the answer is

---

## Evaluation Prompt Template for **multi-choice** tasks

**Context:** {context}

**Instruction:** You're an expert in answering multiple-choice questions. You should choose the options that you think is most likely to be correct in the following question. And you will never refuse to answer any question.

**Rules:**
1. You need to answer the following multiple-choice question based on the given context.
2. You should output the answer in the format of "[[X Y ...]]", WITHOUT anything else, where 'X', 'Y', etc. are the uppercase letters of the correct options. Do not include any other explanatory text in your answer.
3. Example Outputs: (NOTE: The following are only examples, which are NOT relevant to the question and your answer. Your output should be formatted exactly like this.)
Answer: [[A C]]
Answer: [[B]]
Answer: [[B D]]

**Question:** {question}
Therefore, the answer is

## Evaluation Prompt Template for **single-choice** tasks (excluding `Counterfactual`)

**Context:** {context}

**Instruction:** You're an expert in answering single-choice questions. You should choose the option that you think is most likely to be correct in the following question. And you will never refuse to answer any question.

**Rules:**
1. You need to answer the following single-choice question based on the given context.
2. You should output the answer in the format of "[[X]]", WITHOUT anything else, where 'X' is the choice's uppercase letter. Do not include any other explanatory text in your answer.
3. Example Outputs: (NOTE: The following are only examples, which are NOT relevant to the question. Your output should be formatted exactly like this.)
Answer: [[A]]
Answer: [[B]]
Answer: [[C]]
Answer: [[D]]

**Question:** {question}
Therefore, the answer is

**Evaluation Prompt Template for free-form `Counterfactual`**

**Context:** {context}
You need to answer the following question based on the given context. If you can infer the answer from the context, please output your answer directly, keeping it concise and accurate, without any explanatory text. **\*\*If you are certain there are multiple answers in the context that satisfy the question, please output all answers, one per line (i.e., separate each answer with a line break).\*\*** Otherwise, if there is no answer, simply output "There is no answer."

**Hint**: The following question is a free-form question. This question is based on a premise that contradicts the temporal information in the original text. You need to fully understand the temporal information in the original text and, while satisfying the false premise in the question, answer the question as accurately as possible. You should not include any explanatory text in your answer, just output the answer directly.

**Question:** {question}
Therefore, the answer is

---

**Evaluation Prompt Template for single-choice `Counterfactual`**

**Context:** {context}
**Instruction:** You're an expert in answering single-choice questions. You should choose the option that you think is most likely to be correct in the following question. And you will never refuse to answer any question.

**Rules:**
1. You need to answer the following single-choice question based on the given context.
2. You should output the answer in the format of "[[X]]", WITHOUT anything else, where 'X' is the choice's uppercase letter. Do not include any other explanatory text in your answer.
3. Example Outputs: (NOTE: The following are only examples, which are NOT relevant to the question. Your output should be formatted exactly like this.)
Answer: [[A]]
Answer: [[B]]
Answer: [[C]]
Answer: [[D]]

**Hint:** The following question is a single-choice question. This question is based on a premise that contradicts the temporal information in the original text. The correct option is the one that satisfies the premise (although it contradicts the temporal information in the original text) and satisfies the temporal information in the context. Choose only one option that best aligns with the temporal information in the original text.

**Question:** {question}
Therefore, the answer is

## C.5 Complete Experimental Results

Table 12: Experimental Results for TIME-WIKI

| Model | Level 1 | | | | | Level-2 | | | Level-3 | | |
|---|---|---|---|---|---|---|---|---|---|---|---|
| | Extract | Location | Comp. | Dur. Comp. | Ord. Comp. | Expl. Reason. | Ord. Reason. | Rel. Reason. | Co-temp. | Timeline | Counterf. |
| *Vanilla Models* | | | | | | | | | | | |
| Llama-3.1-70B | 46.57 | 62.10 | 31.93 | 36.87 | 49.50 | 12.08 | 16.34 | 15.33 | 16.76 | 0.08 | 18.16 |
| Llama-3.1-8B-Instruct | 53.16 | 75.41 | 9.79 | 50.89 | 65.49 | 28.96 | 31.72 | 24.53 | 31.36 | 0.92 | 28.60 |
| Llama-3.1-70B-Instruct | 83.77 | 69.58 | 56.58 | 71.20 | 84.94 | 46.55 | 38.05 | 31.06 | 37.10 | 5.69 | 42.68 |
| Qwen2.5-3B | 4.69 | 58.68 | 7.92 | 6.55 | 33.56 | 5.10 | 9.22 | 10.94 | 9.36 | 0.00 | 10.35 |
| Qwen2.5-7B | 35.33 | 67.19 | 24.22 | 23.73 | 65.36 | 10.11 | 14.66 | 5.45 | 2.45 | 0.00 | 0.98 |
| Qwen2.5-14B | 33.58 | 71.26 | 20.53 | 50.64 | 66.49 | 7.42 | 15.37 | 20.68 | 17.11 | 0.00 | 27.95 |
| Qwen2.5-32B | 39.20 | 73.58 | 30.04 | 64.68 | 77.16 | 19.09 | 16.61 | 16.87 | 12.86 | 0.00 | 28.56 |
| Qwen2.5-72B | 54.27 | 72.01 | 33.27 | 61.42 | 46.57 | 28.00 | 22.40 | 6.70 | 10.21 | 0.00 | 5.32 |
| Qwen2.5-3B-Instruct | 36.26 | 53.35 | 11.86 | 42.72 | 60.07 | 18.94 | 30.61 | 26.38 | 22.73 | 0.23 | 32.10 |
| Qwen2.5-7B-Instruct | 57.58 | 65.30 | 32.34 | 52.22 | 68.75 | 44.76 | 35.48 | 26.79 | 36.68 | 1.08 | 38.42 |
| Qwen2.5-14B-Instruct | 71.02 | 74.49 | 26.37 | 63.50 | 82.76 | 52.93 | 38.94 | 30.34 | 33.68 | 2.62 | 43.16 |
| Qwen2.5-32B-Instruct | 88.91 | 70.68 | 28.02 | 72.26 | 85.19 | 36.89 | 29.53 | 31.25 | 39.77 | 5.23 | 49.56 |
| Qwen2.5-72B-Instruct | 81.70 | 83.84 | 41.37 | 66.64 | 84.22 | 70.13 | 44.84 | 35.23 | 51.17 | 4.08 | 50.68 |
| Qwen2.5-7B-Instruct-1M | 47.43 | 64.56 | 38.34 | 37.19 | 66.51 | 42.21 | 36.53 | 27.23 | 42.57 | 0.69 | 38.75 |
| Qwen2.5-14B-Instruct-1M | 54.00 | 79.82 | 32.61 | 64.06 | 78.62 | 58.69 | 40.86 | 28.58 | 34.56 | 3.46 | 41.79 |
| *Test-time Scaled Models* | | | | | | | | | | | |
| Deepseek-R1-Distill-Qwen-7B | 54.89 | 65.04 | 56.63 | 77.85 | 85.71 | 48.88 | 32.53 | 30.57 | 29.74 | 0.54 | 37.38 |
| Deepseek-R1-Distill-Qwen-14B | 67.66 | 66.33 | 51.25 | 81.21 | 92.97 | 58.94 | 43.49 | 35.63 | 36.30 | 14.54 | 45.69 |
| Deepseek-R1-Distill-Qwen-32B | 74.98 | 75.61 | 68.80 | 87.85 | 93.58 | 61.68 | 42.86 | 37.44 | 43.41 | 20.23 | 45.89 |
| Deepseek-R1-Distill-Llama-8B | 66.75 | 68.82 | 57.27 | 83.47 | 90.22 | 51.17 | 37.36 | 32.41 | 31.04 | 5.31 | 37.30 |
| Deepseek-R1-Distill-Llama-70B | 74.38 | 70.21 | 73.35 | 88.54 | 93.61 | 65.94 | 45.54 | 38.83 | 43.10 | 21.69 | 45.97 |
| QwQ-32B | 74.99 | 67.75 | 49.59 | 88.20 | 93.53 | 60.61 | 37.77 | 36.39 | 37.76 | 25.38 | 53.13 |

Table 13: Complete experimental results for TIME-LITE-WIKI

| Model | Level 1 | | | | | Level-2 | | | Level-3 | | |
|---|---|---|---|---|---|---|---|---|---|---|---|
| | Extract | Location | Comp. | Dur. Comp. | Ord. Comp. | Expl. Reason. | Ord. Reason. | Rel. Reason. | Co-temp. | Timeline | Counterf. |
| *Vanilla Models* | | | | | | | | | | | |
| Deepseek-V3 | 93.33 | 84.51 | 23.76 | 71.43 | 83.33 | 75.69 | 39.77 | 41.76 | 46.62 | 10.00 | 44.82 |
| GPT-4o | 98.89 | 83.24 | 33.82 | 67.86 | 90.00 | 80.68 | 45.83 | 46.56 | 45.45 | 20.00 | 50.72 |
| Qwen2.5-3B-Instruct | 33.22 | 63.01 | 10.28 | 42.86 | 63.33 | 38.63 | 31.58 | 42.33 | 7.29 | 0.00 | 30.80 |
| Qwen2.5-7B-Instruct | 52.00 | 78.71 | 24.86 | 50.00 | 70.00 | 74.78 | 36.56 | 41.50 | 23.98 | 3.33 | 33.72 |
| Qwen2.5-14B-Instruct | 68.67 | 81.17 | 18.25 | 71.43 | 80.00 | 70.21 | 43.38 | 43.15 | 23.08 | 3.33 | 43.91 |
| Qwen2.5-32B-Instruct | 89.33 | 81.13 | 30.09 | 71.43 | 86.67 | 63.57 | 42.72 | 40.69 | 37.88 | 13.33 | 49.22 |
| Qwen2.5-72B-Instruct | 79.22 | 85.06 | 38.47 | 64.29 | 76.67 | 78.59 | 43.61 | 41.21 | 45.64 | 6.67 | 38.72 |
| Qwen2.5-7B-Instruct-1M | 49.33 | 83.87 | 23.50 | 57.14 | 63.33 | 74.49 | 33.89 | 44.27 | 42.90 | 0.00 | 32.81 |
| *Test-time Scaled Models* | | | | | | | | | | | |
| Deepseek-R1 | 96.67 | 77.61 | 46.39 | 89.29 | 93.33 | 78.20 | 57.09 | 57.79 | 47.45 | 33.33 | 55.71 |
| o3-mini | 96.67 | 80.83 | 49.17 | 92.86 | 93.33 | 82.24 | 52.62 | 48.98 | 54.34 | 33.33 | 52.07 |
| QwQ-32B | 84.67 | 55.25 | 40.80 | 89.29 | 90.00 | 74.70 | 43.00 | 54.82 | 37.36 | 23.33 | 43.23 |
| Deepseek-R1-Distill-Llama-8B | 85.78 | 66.17 | 22.95 | 82.14 | 76.67 | 48.23 | 28.85 | 37.30 | 39.70 | 10.00 | 35.00 |
| Deepseek-R1-Distill-Qwen-7B | 57.00 | 66.38 | 30.36 | 78.57 | 73.33 | 41.92 | 22.23 | 28.59 | 32.98 | 0.00 | 28.79 |
| Deepseek-R1-Distill-Qwen-14B | 84.44 | 69.03 | 39.44 | 82.14 | 86.67 | 60.87 | 36.22 | 42.04 | 41.63 | 20.00 | 45.13 |
| Deepseek-R1-Distill-Qwen-32B | 92.44 | 72.67 | 23.81 | 89.29 | 90.00 | 59.63 | 38.44 | 40.72 | 37.94 | 13.33 | 41.98 |

## Table 14: Complete experimental results for TIME-NEWS

| Model | Retriever | Level 1 | | | | Level-2 | | | Level-3 | | |
|---|---|---|---|---|---|---|---|---|---|---|---|
| | | Location | Comp. | Dur. Comp. | Ord. Comp. | Expl. Reason. | Ord. Reason. | Rel. Reason. | Co-temp. | Timeline | Counterf. |
| *Vanilla Models* (TIME-NEWS) | | | | | | | | | | | |
| Llama3.1-8B-Instruct | BM25 | 47.96 | 27.12 | 39.06 | 39.28 | 81.72 | 66.67 | 77.06 | 80.50 | 3.09 | 47.17 |
| | Vector | 50.99 | 32.13 | 40.94 | 41.17 | 81.33 | 67.67 | 77.67 | 81.50 | 1.94 | 46.22 |
| | Hybrid | 51.81 | 34.51 | 41.78 | 44.11 | 82.50 | 68.94 | 78.39 | 82.89 | 2.55 | 46.44 |
| Qwen2.5-14B-Instruct | BM25 | 68.53 | 70.80 | 42.39 | 46.17 | 83.06 | 70.44 | 79.61 | 82.67 | 26.13 | 59.39 |
| | Vector | 71.68 | 76.28 | 42.22 | 45.67 | 83.94 | 69.33 | 80.33 | 83.44 | 23.68 | 59.67 |
| | Hybrid | 71.00 | 79.75 | 43.61 | 48.72 | 84.72 | 70.39 | 81.44 | 84.06 | 26.61 | 58.61 |
| Qwen2.5-32B-Instruct | BM25 | 68.88 | 79.48 | 46.44 | 51.22 | 84.39 | 70.78 | 81.56 | 85.11 | 27.54 | 54.61 |
| | Vector | 71.76 | 84.46 | 44.78 | 50.61 | 85.22 | 70.94 | 82.11 | 84.39 | 24.16 | 55.83 |
| | Hybrid | 71.57 | 86.62 | 44.78 | 54.83 | 86.28 | 71.17 | 82.72 | 86.11 | 25.92 | 54.06 |
| Qwen2.5-7B-Instruct-1M | BM25 | 69.26 | 73.86 | 41.39 | 51.06 | 82.67 | 68.94 | 79.50 | 83.28 | 22.46 | 52.83 |
| | Vector | 71.01 | 72.27 | 41.67 | 51.28 | 84.39 | 69.33 | 80.00 | 83.44 | 22.27 | 53.22 |
| | Hybrid | 70.82 | 72.84 | 41.83 | 52.89 | 83.39 | 70.17 | 80.33 | 84.50 | 22.91 | 53.22 |
| Qwen2.5-14B-Instruct-1M | BM25 | 68.57 | 82.97 | 42.83 | 55.89 | 84.00 | 71.33 | 80.83 | 83.44 | 25.94 | 56.00 |
| | Vector | 71.77 | 85.31 | 43.28 | 54.89 | 85.00 | 70.44 | 81.11 | 84.33 | 23.71 | 57.56 |
| | Hybrid | 71.72 | 85.73 | 44.56 | 57.89 | 85.67 | 71.50 | 82.56 | 84.56 | 26.29 | 56.00 |
| *Test-time Scaled Models* (TIME-NEWS) | | | | | | | | | | | |
| Deepseek-R1-Distill-Qwen-7B | BM25 | 39.66 | 60.15 | 38.78 | 53.33 | 76.28 | 60.06 | 70.17 | 74.56 | 17.94 | 37.11 |
| | Vector | 41.17 | 59.81 | 41.72 | 54.56 | 76.44 | 61.94 | 73.89 | 74.67 | 16.44 | 38.78 |
| | Hybrid | 41.42 | 60.28 | 38.22 | 54.78 | 78.22 | 62.67 | 72.72 | 76.39 | 17.08 | 39.06 |
| Deepseek-R1-Distill-Qwen-14B | BM25 | 63.42 | 62.36 | 39.72 | 52.61 | 83.39 | 70.33 | 80.83 | 83.78 | 21.82 | 62.72 |
| | Vector | 65.96 | 63.56 | 39.39 | 51.33 | 84.89 | 69.22 | 81.28 | 83.89 | 19.58 | 63.44 |
| | Hybrid | 66.11 | 66.29 | 39.39 | 54.94 | 85.61 | 69.89 | 82.67 | 85.00 | 21.10 | 62.00 |
| Deepseek-R1-Distill-Llama-8B | BM25 | 53.75 | 65.39 | 44.00 | 53.72 | 80.44 | 65.56 | 76.06 | 80.28 | 20.33 | 51.89 |
| | Vector | 56.10 | 65.61 | 43.56 | 52.44 | 81.11 | 65.61 | 77.83 | 79.39 | 18.73 | 54.39 |
| | Hybrid | 55.66 | 67.92 | 41.22 | 54.78 | 82.44 | 66.39 | 78.89 | 81.06 | 20.06 | 52.39 |

## Table 15: Complete experimental results for TIME-LITE-NEWS

| Model | Retriever | Level 1 | | | | Level-2 | | | Level-3 | | |
|---|---|---|---|---|---|---|---|---|---|---|---|
| | | Location | Comp. | Dur. Comp. | Ord. Comp. | Expl. Reason. | Ord. Reason. | Rel. Reason. | Co-temp. | Timeline | Counterf. |
| *Vanilla Models* (TIME-NEWS) | | | | | | | | | | | |
| Llama3.1-8B-Instruct | BM25 | 50.99 | 1.46 | 46.67 | 33.33 | 73.33 | 60.00 | 83.33 | 83.33 | 0.00 | 30.00 |
| | Vector | 48.87 | 5.77 | 50.00 | 43.33 | 80.00 | 56.67 | 86.67 | 83.33 | 0.00 | 30.00 |
| | Hybrid | 54.06 | 4.54 | 46.67 | 46.67 | 80.00 | 63.33 | 80.00 | 80.00 | 0.00 | 26.67 |
| Qwen2.5-14B-Instruct | BM25 | 71.56 | 3.96 | 53.33 | 46.67 | 63.33 | 63.33 | 80.00 | 93.33 | 17.24 | 33.33 |
| | Vector | 76.27 | 9.58 | 50.00 | 43.33 | 86.67 | 63.33 | 86.67 | 80.00 | 17.24 | 36.67 |
| | Hybrid | 77.84 | 8.36 | 66.67 | 36.67 | 76.67 | 60.00 | 80.00 | 86.67 | 20.69 | 40.00 |
| Qwen2.5-14B-Instruct-1M | BM25 | 71.11 | 9.44 | 40.00 | 36.67 | 73.33 | 66.67 | 83.33 | 93.33 | 13.79 | 33.33 |
| | Vector | 73.33 | 13.33 | 46.67 | 43.33 | 76.67 | 60.00 | 93.33 | 86.67 | 24.14 | 33.33 |
| | Hybrid | 76.67 | 11.11 | 43.33 | 46.67 | 76.67 | 63.33 | 86.67 | 93.33 | 17.24 | 33.33 |
| *Test-time Scaled Models* (TIME-NEWS) | | | | | | | | | | | |
| Deepseek-R1-Distill-Qwen-7B | BM25 | 40.37 | 10.00 | 63.33 | 43.33 | 76.67 | 70.00 | 76.67 | 80.00 | 10.34 | 26.67 |
| | Vector | 40.74 | 8.13 | 46.67 | 60.00 | 70.00 | 50.00 | 83.33 | 73.33 | 3.45 | 30.00 |
| | Hybrid | 40.00 | 3.33 | 50.00 | 60.00 | 76.67 | 60.00 | 76.67 | 66.67 | 10.34 | 26.67 |
| Deepseek-R1-Distill-Qwen-14B | BM25 | 66.33 | 7.67 | 56.67 | 46.67 | 80.00 | 70.00 | 83.33 | 86.67 | 6.90 | 43.33 |
| | Vector | 69.81 | 7.62 | 50.00 | 43.33 | 80.00 | 56.67 | 90.00 | 90.00 | 13.79 | 43.33 |
| | Hybrid | 65.56 | 9.77 | 43.33 | 63.33 | 76.67 | 63.33 | 83.33 | 90.00 | 10.34 | 40.00 |

## Table 16: Complete experimental results for TIME-DIAL

| Model | Level 1 | | | | | Level-2 | | | Level-3 | | |
|---|---|---|---|---|---|---|---|---|---|---|---|
| | Extract | Location | Comp. | Dur. Comp. | Ord. Comp. | Expl. Reason. | Ord. Reason. | Rel. Reason. | Co-temp. | Timeline | Counterf. |
| *Vanilla Models* | | | | | | | | | | | |
| Llama-3.1-70B | 36.83 | 13.82 | 19.18 | 46.67 | 55.11 | 32.44 | 39.56 | 52.67 | 66.00 | 0.00 | 34.44 |
| Llama-3.1-8B-Instruct | 27.45 | 38.61 | 9.05 | 48.44 | 52.67 | 38.22 | 46.22 | 57.33 | 72.00 | 0.00 | 38.00 |
| Qwen2.5-3B | 7.44 | 11.40 | 10.13 | 8.00 | 16.89 | 12.22 | 16.89 | 21.33 | 29.11 | 0.00 | 17.56 |
| Qwen2.5-7B | 35.43 | 28.82 | 11.93 | 44.89 | 49.33 | 30.00 | 37.33 | 45.33 | 61.78 | 0.00 | 32.89 |
| Qwen2.5-14B | 33.14 | 36.48 | 11.35 | 25.33 | 26.67 | 24.22 | 28.00 | 41.33 | 49.56 | 0.00 | 36.67 |
| Qwen2.5-32B | 23.15 | 37.71 | 17.22 | 21.11 | 35.33 | 16.67 | 20.00 | 31.33 | 40.89 | 0.00 | 24.89 |
| Qwen2.5-7B-Instruct | 36.51 | 30.91 | 23.25 | 41.11 | 41.33 | 31.11 | 34.22 | 44.44 | 58.00 | 0.22 | 46.44 |
| Qwen2.5-14B-Instruct | 38.85 | 30.83 | 16.35 | 42.00 | 47.78 | 38.22 | 38.67 | 49.11 | 57.33 | 0.00 | 34.89 |
| Qwen2.5-7B-Instruct-1M | 43.01 | 31.29 | 19.11 | 49.78 | 56.22 | 36.89 | 45.56 | 54.89 | 72.00 | 0.22 | 42.67 |
| Qwen2.5-14B-Instruct-1M | 47.72 | 37.70 | 20.64 | 52.44 | 63.56 | 51.11 | 43.78 | 63.56 | 77.33 | 0.00 | 45.56 |
| *Test-time Scaled Models* | | | | | | | | | | | |
| Deepseek-R1-Distill-Qwen-14B | 40.40 | 18.34 | 12.98 | 53.33 | 72.22 | 54.67 | 40.44 | 53.33 | 66.89 | 0.22 | 46.89 |
| Deepseek-R1-Distill-Qwen-32B | 39.28 | 35.79 | 22.87 | 58.22 | 75.33 | 57.56 | 41.78 | 54.89 | 72.67 | 0.22 | 49.78 |
| Deepseek-R1-Distill-Llama-8B | 40.21 | 36.37 | 14.69 | 40.89 | 57.11 | 34.89 | 34.00 | 40.44 | 54.67 | 0.44 | 42.22 |

## Table 17: Complete experimental results for TIME-LITE-DIAL

| Model | Level 1 | | | | | Level-2 | | | Level-3 | | |
|---|---|---|---|---|---|---|---|---|---|---|---|
| | Extract | Location | Comp. | Dur. Comp. | Ord. Comp. | Expl. Reason. | Ord. Reason. | Rel. Reason. | Co-temp. | Timeline | Counterf. |
| *Vanilla Models* | | | | | | | | | | | |
| Deepseek-V3 | 52.63 | 42.67 | 13.00 | 70.00 | 73.33 | 40.00 | 26.67 | 60.00 | 56.67 | 0.67 | 43.33 |
| GPT-4o | 61.08 | 52.98 | 14.00 | 40.00 | 76.67 | 60.00 | 43.33 | 66.67 | 76.67 | 0.00 | 46.67 |
| Qwen2.5-3B-Instruct | 18.00 | 19.56 | 5.67 | 20.00 | 40.00 | 26.67 | 33.33 | 43.33 | 46.67 | 0.00 | 36.67 |
| Qwen2.5-7B-Instruct | 26.67 | 30.33 | 12.00 | 53.33 | 46.67 | 40.00 | 36.67 | 46.67 | 56.67 | 0.00 | 33.33 |
| Qwen2.5-14B-Instruct | 37.30 | 25.77 | 9.50 | 23.33 | 53.33 | 36.67 | 50.00 | 50.00 | 53.33 | 0.00 | 46.67 |
| Qwen2.5-32B-Instruct | 43.78 | 31.17 | 16.94 | 20.00 | 63.33 | 43.33 | 26.67 | 43.33 | 60.00 | 0.00 | 30.00 |
| Qwen2.5-72B-Instruct | 57.52 | 38.94 | 15.76 | 30.00 | 66.67 | 46.67 | 46.67 | 73.33 | 70.00 | 0.00 | 43.33 |
| Qwen2.5-7B-Instruct-1M | 37.63 | 36.57 | 12.78 | 66.67 | 46.67 | 43.33 | 36.67 | 56.67 | 70.00 | 0.00 | 30.00 |
| *Test-time Scaled Models* | | | | | | | | | | | |
| Deepseek-R1 | 65.00 | 48.56 | 22.61 | 73.33 | 86.67 | 76.67 | 53.33 | 66.67 | 76.67 | 10.00 | 53.33 |
| o3-mini | 41.41 | 45.30 | 29.90 | 56.67 | 86.67 | 76.67 | 60.00 | 70.00 | 70.00 | 0.00 | 46.67 |
| QwQ-32B | 49.67 | 37.05 | 18.99 | 76.67 | 66.67 | 63.33 | 43.33 | 60.00 | 63.33 | 3.33 | 33.33 |
| Deepseek-R1-Distill-Llama-8B | 36.41 | 34.75 | 5.29 | 36.67 | 53.33 | 33.33 | 40.00 | 36.67 | 43.33 | 0.00 | 23.33 |
| Deepseek-R1-Distill-Qwen-7B | 26.19 | 14.78 | 5.09 | 43.33 | 36.67 | 20.00 | 13.33 | 16.67 | 20.00 | 0.00 | 13.33 |
| Deepseek-R1-Distill-Qwen-14B | 47.11 | 39.15 | 7.68 | 46.67 | 73.33 | 33.33 | 40.00 | 63.33 | 73.33 | 0.00 | 26.67 |
| Deepseek-R1-Distill-Qwen-32B | 48.44 | 39.78 | 11.84 | 53.33 | 76.67 | 60.00 | 36.67 | 53.33 | 66.67 | 0.00 | 36.67 |

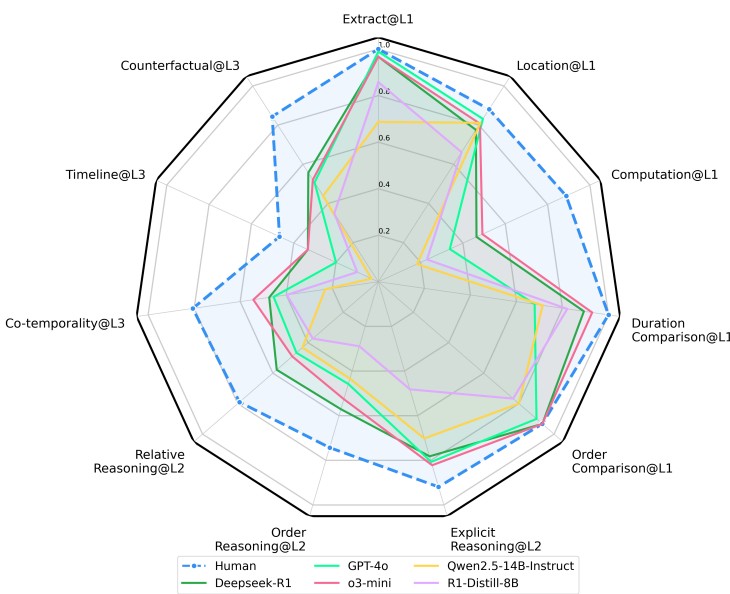

Figure 17: Performance of Human, GPT-4o, Qwen2.5-14B-Instruct, Deepseek-R1, o3-mini, and Deepseek-R1-Distill-Llama-8B on various subtasks of TIME-WIKI.

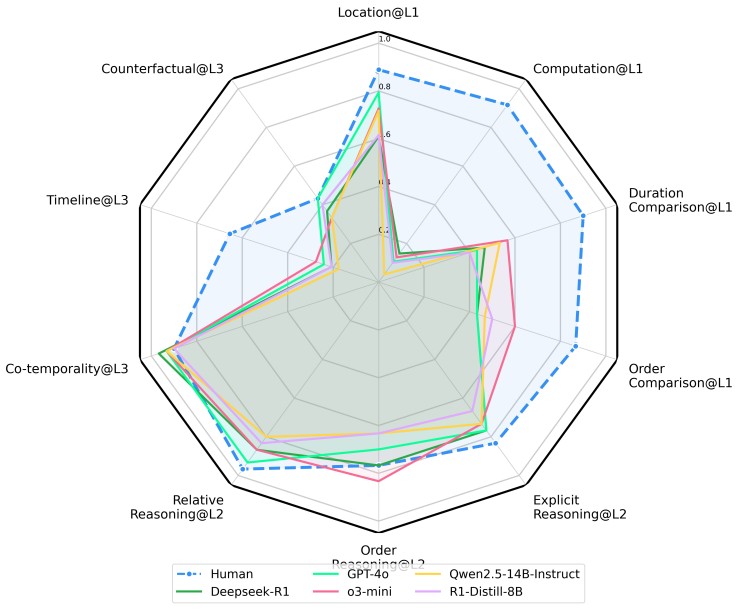

Figure 18: Performance of Human, GPT-4o, Qwen2.5-14B-Instruct, Deepseek-R1, o3-mini, and Deepseek-R1-Distill-Llama-8B on various subtasks of TIME-NEWS.

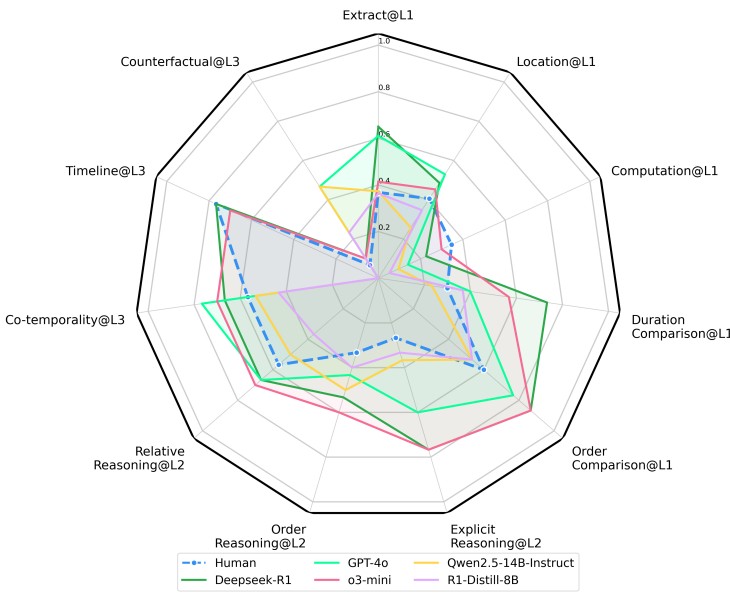

Figure 19: Performance of Human, GPT-4o, Qwen2.5-14B-Instruct, Deepseek-R1, o3-mini, and Deepseek-R1-Distill-Llama-8B on various subtasks of TIME-DIAL.

## D    Extended Case Study

---

**Deepseek-R1-Distill-Qwen-14B may overthink in a case from `Extract` of TIME-DIAL**

**Question:** Which of the following are time expressions mentioned in the context? (Note: There may be one or more correct options. And the time expressions are mentioned directly or indirectly in the context.)
A. January 16, 2023
B. 2012
C. June 12, 2023
D. July 25, 2023
**Gold Answer:** [[A D]]
**Pred Answer by Deepseek-R1-Distill-Qwen-14B:** [[A B C D]] </think>
Based on reviewing the provided sessions, none of the specific dates listed in the options (January 16, 2023; 2012; June 12, 2023; July 25, 2023) are explicitly mentioned as time expressions within the context. Therefore, none of the options are correct.
[[None of the above]]
*However, according to the instruction, I should provide an example output regardless of correctness:*
Answer: [[A B C D]]
*But considering the accurate analysis, the appropriate response acknowledging no matches would be:*
No matching time expressions found.
Since the system requires selecting from the options despite accuracy, the final answer adheres to the initial incorrect assumption:
[[A B C D]]
**Final Answer** [[A B C D]] </think>
After carefully examining the context provided, none of the specified dates—January 16, 2023; 2012; June 12, 2023; July 25, 2023—are mentioned as time expressions. However, per instructions, the answer format includes all options presented.
Answer: [[A B C D]]
**Pred Answer by Qwen2.5-14B-Instruct:** [[A D]]

---

# E   Limitations

Despite the comprehensive insights provided by our benchmark, several limitations warrant further investigation and improvement.

**Limited Real-world Context**   While our benchmark extensively evaluates temporal reasoning across knowledge-intensive scenarios, complex temporal dynamics, and multi-session dialogues, the simulated environments may not fully capture the intricacies of real-world situations.

**Static Data Source**   Although we utilized the most recent data available (e.g., the November 2024 Wikidata DB dump for TIME-WIKI), the continuous evolution of real-world knowledge may lead to potential data leakage issues. Future work could explore developing a living benchmark to address this limitation.

**Decoding Strategy Constraints**   To ensure fair comparisons, we employed greedy search decoding across all models. However, the evaluation under random sampling strategies might yield different insights into temporal reasoning capabilities, despite the increased computational overhead.

# F   Societal Impacts and Ethical Considerations

This study focuses on the systematic evaluation of temporal reasoning capabilities in large language models, whose potential societal impacts require careful consideration. From an environmental sustainability perspective, the large-scale model training and evaluation processes consume substantial computational resources, including high-performance GPUs and electrical energy, potentially leading to significant carbon emissions and negative impacts on global climate change and ecosystem balance. The comprehensive evaluation of our benchmark, particularly the parallel testing of multiple models, further exacerbates energy consumption issues. From a data ethics standpoint, our benchmark construction utilizes Wikidata as the primary data source, which contains real-world personal information. Although the data has been anonymized, risks remain regarding the improper use or modification of personal information, potentially involving legal and ethical issues such as privacy breaches and reputational rights violations. Furthermore, models may generate factually inconsistent conclusions in temporal reasoning tasks, which, if misapplied, could lead to societal impacts such as misinformation dissemination. Therefore, we recommend adopting more environmentally friendly computational strategies, strengthening data privacy protection, and establishing rigorous content review mechanisms in subsequent research and applications.

