# OpenReview forum: "TimE: A Multi-level Benchmark for Temporal Reasoning of LLMs in Real-World Scenarios"
_NeurIPS.cc/2025/Datasets_and_Benchmarks_Track — NeurIPS 2025 Datasets and Benchmarks Track spotlight_

### Official Review · Reviewer_cXYQ · 2025-07-02

**Rating:** 5
**Confidence:** 4

**Summary:**

This paper proposes a new TIME benchmark, a multi-level evaluation framework for assessing temporal reasoning capabilities of Large Language Models (LLMs) in real-world scenarios. It includes three sub-datasets: TIME-WIKI, TIME-NEWS, and TIME-DIAL, each addressing different temporal reasoning challenges. The benchmark features a hierarchical structure with three levels of tasks: 1) Level1: Basic Temporal Understanding and Retrieval includes tasks like Extract, Localization, Computation, DurationCompare, and OrderCompare. 2) Level2: Temporal Expression Reasoning involves Explicit Reasoning, Order Reasoning, and Relative Reasoning. 3) Level3: Complex Temporal Relationship Reasoning covers Co-temporality, Timeline, and Counterfactual Reasoning.

**Dataset Code Accessibility:**

Yes

**Ethical Considerations:**

No, there are no or only very minor ethics concerns

**Final Justification:**

After carefully considering the authors’ rebuttal and the follow-up discussions, I have revised my evaluation of the paper. There are two main factors that informed my updated recommendation:

First, authors clarified the main differentiation from existing temporal datasets using a table. One of my main previous concerns was the distinctiveness of the proposed evaluation framework compared to prior work such as TCELongBench because of many temporal reasoning datasets. The authors have addressed this point clearly in the rebuttal by highlighting the differences. Their explanation clarified the unique contributions of the proposed framework.

Second, I was worried about the temporal complexity. The authors provided helpful statistical metrics to explain the complexity of their dataset, including "an average time span of 405.87 days", "16.93 different entities per question", and "40.17 SRO triples". These metrics suggest that the dataset is sufficiently rich and capable of capturing complex temporal reasoning challenges.

**Limitations Weaknesses:**

Weaknesses:

1.	While this dataset is large and well-designed for temporal reasoning, the key differences from existing datasets across various dimensions are not clearly articulated. From the related work, it appears that the proposed TIME dataset serves as a unified framework for comprehensive evaluation, as it encompasses more real-world scenarios and multiple levels of reasoning. However, it remains unclear whether this dataset introduces new types of temporal reasoning tasks that differ significantly from those in previous datasets.

2.	The paper highlights the construction of a large-scale Complex Temporal Relationship Reasoning dataset containing multi-event interaction samples as a key contribution. While it's plausible that news data includes many events, it is not clearly explained how the dataset captures complex interactions among multiple events—such as interactions involving ten or more events. It would be helpful for the authors to elaborate on their methodology for modeling such interactions and to provide a quantitative analysis of the event interactions represented in the dataset.

**Strengths Contributions:**

Strengths:

1.	This paper presents a new multi-level evaluation framework for temporal reasoning. It can systematically evaluate temporal understanding capabilities from different granularities.

2.	The proposed TIME benchmark is a very reasonable data which captures temporal reasoning situations in diverse real-world scenarios.

3.	The depth analysis on this data provides critical insights for advancing temporal reasoning capabilities.

---

> ### Author Rebuttal · Authors · 2025-07-31
>
> # Response to Reviewer cXYQ
>
> We appreciate the reviewer's time and constructive feedback, which have helped us improve the clarity and rigor of our work. Below, we provide our responses to each comment.
>
>
> > **W1: Key differences between TIME dataset and previous datasets.**
>
> We sincerely appreciate your valuable feedback. We would like to clarify that the key differences between TIME dataset and previous datasets are as follows.
>
> In comparison to previous datasets, the TIME dataset introduces several significant differences, both in terms of evaluation scenarios and the design of reasoning tasks. Specifically, there are six key areas of differentiation:
>
> 1. **Realistic Evaluation Scenarios**: TIME presents long, complex texts with an average of over 10 entities per context, involving Subject-Relation-Object (SRO) triples and multiple interwoven timelines. Existing datasets either lack real-world data (e.g., TRAM[1]) or provide overly simplistic, short contexts that are not applicable for current time reasoning models (e.g., TimeBench[2]).
> 2. **Comprehensive and Fair Evaluation Framework**: TIME’s evaluation framework offers a fine-grained and comprehensive assessment of temporal reasoning abilities, measuring various capabilities within the same context. In contrast, datasets like TimeBench[2] assess sub-tasks using existing datasets with varying levels of difficulty and scale, leading to potential bias in model evaluation due to differing difficulty levels or scale, making the evaluation unfair.
> 3. **Hierarchical Reasoning Relationships**: The sub-tasks in TIME are directly related, offering a layered approach to evaluation. For example, in Figure 1 of our paper, when asking, *"Who was the most recent person to hold the position of titular bishop before Mauro Morelli stepped down as diocesan bishop?"* our framework can evaluate the model's understanding of temporal expressions like *"before Mauro Morelli stepped down"* at a finer level, assessing whether the model grasps fundamental time meanings such as *"Find the time when Mauro Morelli stepped down"* and the temporal relation *"before"*. This hierarchical reasoning provides more explainable results, which previous datasets do not address.
> 4. **Complex World Knowledge**: TIME evaluates models on their ability to reason temporally within vast amounts of real-world knowledge, posing a more challenging task than previous datasets.
> 5. **Dynamic Temporal Reasoning Challenges**: The dataset assesses models on their ability to accurately understand and reason about time within dynamically changing facts, a capability that is often neglected in other datasets.
> 6. **Long-Text Temporal Reasoning**: TIME evaluates the model’s ability to capture and reason about time-related information within long contexts, which is a more difficult challenge than the short-context evaluations seen in other datasets.
>
>
> **The differences outlined above are summarized in the table below. If the paper is accepted, we will include this table in the camera-ready version.**
>
> ### Table: Key differences between TIME dataset and previous datasets
> | Evaluation Scenario | Realistic Evaluation Scenarios | Comprehensive and Fair Evaluation Framework | Hierarchical Reasoning Relationships | Complex World Knowledge | Dynamic Temporal Reasoning Challenges | Long-Text Temporal Reasoning |
> |:----------------------------------|:--------------------------------|:---------------------------------------------|:-------------------------------------|:--------------------------|:---------------------------------------|:------------------------------|
> |TRAM [1]|✗|✓|✗|✗|✗|✗|
> |TimeBench [2]|✗|✗|✗|✓|✗|✗|
> |TCELongBench [3]|✓|✗|✗|✓|✓|✓|
> |TimeQA [4]|✗|✗|✗|✓|✗|✗|
> |FreshLLMs [5]|✓|✗|✗|✗|✓|✗|
> |LoCoMo [6]|✓|✗|✗|✗|✗|✓|
> |**TIME (Ours)**|✓|✓|✓|✓|✓|✓|
>
>
> In addition, the specific design of the reasoning tasks in TIME also differs from previous work:
>
> * **Task Design Focus**: The main goal of the TIME dataset is to simulate human-like use of time for understanding complex and dynamic world information. We specifically aim to evaluate how models use temporal concepts to better understand and solve real-world problems. Thus, we designed three levels of tasks:
>
>   * **Level 1**: Basic temporal understanding and retrieval tasks.
>
>   * **Level 2**: Tasks that require reasoning over implicit or vague temporal expressions to infer event details.
>
>   * **Level 3**: Tasks that focus on understanding complex temporal relationships between events, crucial for clarifying timelines and event causality.
>
> * **Task Variety**: We made detailed considerations in the design of each reasoning task. In addition to explicitly extracting temporal reasoning tasks, we included tasks involving *relative reasoning*, *implicit reasoning* (dealing with vague or range-based temporal expressions), *computation* (for precise time calculations), and *counterfactual* reasoning (dealing with abnormal temporal sequences). These types of tasks distinguish the TIME dataset from previous datasets, which do not incorporate such diverse reasoning challenges.
>
> In summary, the TIME dataset introduces a comprehensive framework with a more nuanced and dynamic approach to temporal reasoning, alongside a variety of tasks designed to evaluate a broader spectrum of temporal understanding abilities than previous datasets.
>
> ### References
>
> [1] Wang, Yuqing, and Yun Zhao. "TRAM: Benchmarking Temporal Reasoning for Large Language Models." ACL 2024 Findings.
>
> [2] Chu, Zheng, et al. "TimeBench: A Comprehensive Evaluation of Temporal Reasoning Abilities in Large Language Models." ACL 2024.
>
> [3] Zhang, Zhihan, et al. "Analyzing Temporal Complex Events with Large Language Models? A Benchmark towards Temporal, Long Context Understanding." ACL 2024.
>
> [4] Chen, Wenhu, Xinyi Wang, and William Yang Wang. "A Dataset for Answering Time-Sensitive Questions." NeurIPS 2021 Track on Datasets and Benchmarks.
>
> [5] Vu, Tu, et al. "FreshLLMs: Refreshing Large Language Models with Search Engine Augmentation." ACL 2024 Findings.
>
> [6] Maharana, Adyasha, et al. "Evaluating Very Long-Term Conversational Memory of LLM Agents." ACL 2024.
>
>
> > **W2: Clarification on multi-event complex interaction in TIME dataset.**
>
> **First, we would like to clarify that our focus is on temporal reasoning abilities.** Therefore, **when we emphasize multi-event interaction, we refer to the temporal interactions and relationships between events, rather than the interactions between the details of the events themselves.** This interaction refers to: (1) the chronological sequence of event occurrences, (2) the duration of multiple events, (3) the comparison of the intervals between events, and (4) the potential overlap or intersection of the timelines of two or more events.
>
> To facilitate such complex temporal interactions, **we carefully constructed timelines and, based on these timelines, designed various question formats to assess the complex temporal relationships between events**:
>
> * **Constructing Timelines:** For each set of temporal complex events, we extract the timeline based on curated articles, identifying key points from all timestamps. Each key point represents an informative and concise sentence that conveys the atomic event information, documenting the fact-triples for each entity. When we unfold these timelines, we can reconstruct the entire story.
> * **Designing Complex Interaction-Related Questions Based on Timelines:** For nearly every question, we design it around the temporal interactions between events. Specifically, we randomly select two or more related key points from the timeline of the same set of temporal complex events and design questions based on these key points. For example, in the *Timeline* task, we randomly shuffle multiple key points and design questions to test the model's ability to reorder them correctly. In the *Co-temporality* task, we choose two key points with overlapping temporal relations and use the time of one key point as a condition to inquire about the event details of an entity in the other key point. This design fully evaluates the model's capacity to handle complex interactions.
>
> **Next, we will provide statistical data from TIME-News regarding event interactions:**
>
> * **Data Source Statistics:** To ensure event complexity, we selected a set of events from the data sources that span a sufficiently long period of time and involve timelines with multiple dates. **As shown in Table 6 of our paper, each set of temporal complex events has an average time span of 405.87 days and an average of 7.45 dates (or timestamps) in the timeline.**
> * **Timeline Statistics:** We randomly selected 100 sets of temporal complex events from TIME-News using seed 42, and semi-automatically counted the number of Subject-Relation-Object (SRO) triples in their timelines. **The results showed that each set of temporal complex events contains an average of 16.93 different entities as subjects and an average of 40.17 SRO triples.** During the statistics process, we used the Deepseek-R1-0528 prompt to extract S, R, O triples for each temporal complex event, then employed three annotators with rich annotation experience to revise the extracted results. The final statistics are shown above.
>
> Based on the above methods of constructing complex event interactions and the statistical data, we ensured that the temporal interactions between events in TIME-News are complex and present a significant challenge to large language models in capturing these complex temporal relationships.

---

> ### Comment · Reviewer_cXYQ · 2025-08-05
> **Appreciation of Authors' Feedback**
>
> I greatly appreciate the authors' thorough responses to my questions, which have provided valuable clarification on several key points.
>
> **Regarding Question 1**:
> I initially worried about the differentiation between this paper's evaluation framework and existing works like TCELongBench. The provided table effectively addressed this concern by highlighting two key distinctions:
>
>    - The TIME dataset incorporates hierarchical reasoning relationships;
>    - The framework encompasses more complex temporal reasoning tasks compared to the limited scope of previous datasets.
>
> Furthermore, Table 10 demonstrates an appropriate sample size for Level 3 questions, which are particularly valuable given the challenge of generating complex temporal reasoning questions.
>
> **Regarding Question 2:**
> I appreciate the clarification about the creation of data with complex temporal relationships, including the timeline construction tasks involving multiple events. However, I would like to explore this aspect further, specifically regarding the quantification of temporal task complexity.  The reason I asked the question is to ask more analysis about the number of events in the proposed temporal reasoning data especially on the complex tasks.
>
> This complexity may be visualized as a temporal trajectory or graph, where events are nodes and temporal relationships are edges.  Then two potential dimensions to analyze the temporal task complexity can be:
>
>    - The number and nature of temporal interactions/relationships;
>    - The quantity of temporal events involved. More events in temporal reasoning increases complexity.
>
> Analyzing the number of nodes (events) and edges (temporal relationships) would provide quantitative metrics for assessing task difficulty. The analysis on TIME-News has partially covered this concern. Such analysis could offer valuable insights into measuring and understanding the complexity levels of temporal reasoning tasks. Please add these analysis about complexity in the new version of paper.
>
> In general, I really appreciate for the valuable feedback. Most of my concerns have been addressed. Hence I'd like to raise my score accordingly.

---

> > ### Author Response · Authors · 2025-08-08
> >
> > We thank the reviewer for the helpful suggestion. We agree that measuring the number of events and temporal relationships is a meaningful way to understand task complexity. We are committed to incorporating this analysis into the camera-ready version, should the paper be accepted.

---

### Official Review · Reviewer_GXAP · 2025-07-03

**Rating:** 5
**Confidence:** 4

**Summary:**

The paper introduces **TIME**, a large‐scale benchmark for evaluating temporal reasoning in large language models (LLMs). TIME spans three realistic domains—knowledge (TIME-WIKI), dynamic news (TIME-NEWS) and long multi-turn dialogue (TIME-DIAL)—and contains 38 522 question-answer (QA) pairs organised into three difficulty levels and 11 sub-tasks that progress from explicit temporal extraction to complex event reasoning. A 943-item human-verified subset (TIME-LITE) is also released. Experiments on 24 open-source and commercial models analyse the impact of retriever choice, decoding strategy and test-time scaling, revealing persistent weaknesses (e.g., timeline ordering, counterfactual reasoning).

**Dataset Code Accessibility:**

Yes

**Ethical Considerations:**

No, there are no or only very minor ethics concerns

**Final Justification:**

The authors provided a solid rebuttal which provided additional information. While I share the concerns of the reviewers for the original paper, I think that the paper can significantly improve if all comments and additional experiments from the rebuttal would be incorporated in the main paper and/or the supplementary. Given the rebuttal, I'm also confident that the authors are interested in improving the paper and will do this. I therefore increase my recommendation.

**Limitations Weaknesses:**

1. **Retrieval–reasoning confound** – In TIME-NEWS, simply switching from BM25 to a Hybrid retriever changes GPT-4o’s Timeline score by >10 pp.   Therefore low accuracy may stem from missing evidence, not reasoning failure.

2. **Static knowledge snapshot** – TIME-WIKI freezes a 1 Nov 2024 Wikidata dump, risking both model leakage and rapid obsolescence.

3. **Context-length bottleneck** – TIME-DIAL dialogues average 15.9 k tokens, so results partly reflect window limits rather than pure temporal reasoning.

**Strengths Contributions:**

1. **Diagnostic ladder** – The three-tier, eleven-subtask design cleanly isolates skills from date lookup to counterfactual reasoning, enabling precise error analysis.

2. **Diverse real-world scenarios** – Combines Wikidata facts, a news corpus of temporal complex events, and \~16 k-token multi-session chats.

3. **Human-verified subset (TIME-LITE)** – 943 QAs with 89 % inter-annotator agreement permit lightweight yet reliable evaluation.

4. **Reveals persistent weaknesses** – Even GPT-4o falls below 50 % on Timeline and Counterfactual tasks; small models hover near chance.

---

> ### Author Rebuttal · Authors · 2025-07-30
>
> # Response to Reviewer GXAP
>
> We thank the reviewer for the thoughtful comments and valuable insights. Your insights have been very valuable in refining our manuscript, and we are grateful for the time and effort you've invested in reviewing our work. We have carefully considered your feedback, and in the following responses, we have provided detailed clarifications and additional experiments to address the concerns raised. We hope these revisions demonstrate the significance of the improvements made and will help in elevating the quality of the paper.
>
> > **W1: Clarification on the concern about retrieval-reasoning confound.**
>
> **We would like to first clarify that the use of a retrieval mechanism is critical for effective evaluation.** As indicated in Table 6 of our paper, the average article length in the TIME-News retrieval corpus is approximately 527k tokens, which far exceeds the context window of almost all existing models. To address this, we adopt the methodology from prior research \[1] and utilize state-of-the-art retrieval techniques, including BM25-based, Vector-based (BGE-3 \[2]), and Hybrid retrieval approaches. These methods allow us to retrieve the top-3 articles as context, ensuring that we can effectively extract task-relevant content from lengthy documents. This retrieval mechanism is crucial for enabling the model to focus on the specific temporal reasoning tasks at hand.
>
>
> **In addition, we manually annotated and analyzed the responses generated by GPT-4o, calculating the error rates when the top-3 retrieved contexts were fully accurate and aligned with the gold answers.** Specifically, we set a random seed of 42 and randomly selected 100 questions, gold answers, and the corresponding GPT-4o predictions across all sub-tasks. These samples were then manually annotated by professional annotators. (*Note: For the human annotation process, we selected three annotators with at least an undergraduate degree and substantial experience in text annotation. All annotators possess excellent proficiency in English.*) Below are the error rates observed when the top-3 contexts retrieved by different retrieval methods were entirely correct and consistent with the gold answers:
>
>
> ### Table: Error rates of GPT-4o's answers when the top-3 contexts retrieved by different retrieval methods were entirely correct and consistent with the gold answers
> | Retrieval Method      | Error Rate (%) |
> |------------------------|----------------|
> | BM25-based Retrieval   | 29.51          |
> | Vector-based Retrieval | 33.33          |
> | Hybrid Retrieval       | 36.62          |
>
>
> **These results suggest that even when the retrieval system accurately fetches the relevant document content, GPT-4o's average reasoning error rate remains around one-third.**
>
>
> **Additionally, we have conducted experiments by increasing the top-k value from 3 to 5 in an effort to reduce the impact of limited relevant context.** The experimental results on the TIME-News dataset are presented in the table below. Abbreviations follow those in Table 2 of our paper. For consistency with the experiments in Table 3 (see Table 3 in our paper, where top-k=3 is used as the retrieval setting), we employed greedy search as the decoding strategy.
>
>
> ### Table: Results for TIME-News, top-k=5 for all retrieval methods
> |Model|Retriever(top-k=5)|Loc.|Comp.|DC.|OC.|ER.|OR.|RR.|Co-tmp.|TL.|CTF.|
> |:---------------------------------|:----------|:------|:------|:-----|:-----|:-----|:-----|:-----|:-------|:-----|:-----|
> |Llama-3.1-8B-Instruct|BM25|57.96|42.65|42.78|43.96|82.7|68.39|78.09|81.35|7.26|48.67|
> ||Vector|60.13|47.45|45.00|42.73|83.22|68.72|78.61|81.61|6.49|47.83|
> ||Hybrid|60.31|49.00|43.67|45.04|83.94|69.00|79.42|81.68|6.79|47.61|
> |Qwen2.5-14B-Instruct|BM25|70.72|74.55|47.94|48.61|85.17|71.31|80.44|83.84|23.63|60.11|
> ||Vector|71.11|77.10|49.06|45.70|86.22|77.61|79.22|83.57|20.81|60.31|
> ||Hybrid|72.31|80.22|47.39|48.57|87.13|76.23|80.82|81.72|23.15|59.89|
> |Deepseek-R1-Distill-Qwen-7B|BM25|54.34|71.28|41.33|53.33|76.00|60.94|72.44|73.28|20.84|40.28|
> ||Vector|56.22|71.51|42.67|56.94|77.33|62.11|72.83|75.61|19.66|40.50|
> ||Hybrid|56.07|71.21|42.78|57.28|77.39|63.33|75.33|74.33|19.24|39.28|
> |Deepseek-R1-Distill-Qwen-14B|BM25|68.11|72.34|44.00|58.72|84.50|70.50|81.39|84.39|27.59|65.56|
> ||Vector|69.31|72.54|42.33|58.89|84.00|70.22|82.33|84.61|24.11|64.89|
> ||Hybrid|69.09|72.57|43.72|61.56|86.11|71.50|83.89|85.67|27.20|67.39|
>
> From the table above, and by comparing the results with those in Table 3 of our paper, we observe that for certain tasks, such as *Location* and *Computation*, increasing the top-k value to 5 leads to a slight improvement in the model's accuracy. **However, for tasks like *Timeline*, raising the top-k value does not significantly enhance the model’s ability to reason about complex events, with accuracy remaining below 30%.** Therefore, we conclude that even with more relevant context provided, the model’s ability to handle complex event reasoning remains a limitation.
>
>
> **Furthermore, we will incorporate higher top-k retrieval results in future experiments to further mitigate the impact of context loss due to retrieval biases.** We will also continue to monitor advancements in retrieval technologies to ensure the reliability of our findings.
>
>
>
> ### References
>
> [1] Zhang, Zhihan, et al. "Analyzing Temporal Complex Events with Large Language Models? A Benchmark towards Temporal, Long Context Understanding." ACL 2024.
>
> [2] Chen, Jianlv, et al. "Bge m3-embedding: Multi-lingual, multi-functionality, multi-granularity text embeddings through self-knowledge distillation." arXiv preprint arXiv:2402.03216 (2024).
>
>
> > **W2: Clarification on the concern about static knowledge snapshot and potential data leakage.**
>
> We fully agree with the importance of maintaining the freshness of the knowledge facts in the dataset. To address the potential data leakage issue, we have taken the following steps:
>
> **First, during the dataset construction phase, we specifically focused on the potential issues arising from the static knowledge snapshot.** On one hand, when constructing TIME-Wiki, we carefully filtered out knowledge triples that involve future tense or rapidly changing information, with assistance from both manual curation and LLMs. On the other hand, when generating each sub-task's question, we employed code scripts to control the temporal range of the knowledge being queried, ensuring that questions did not involve future events or knowledge that could change dynamically. As a result, we have made every effort to mitigate model and data leakage issues stemming from static knowledge.
>
> **Additionally, we are committed to continuously updating the knowledge facts in the benchmark at appropriate intervals, removing outdated question-and-answer pairs from our dataset to ensure the answers remain up-to-date.** This approach will further address concerns regarding the static knowledge snapshot and potential data leakage issues.
>
>
> > **W3: Clarification on the concern about context-length bottleneck in evaluation.**
>
> **We would like to clarify that the context-length bottleneck is not a unique challenge for temporal reasoning tasks; rather, it is a general issue that impacts all tasks with long context.**
>
> **More importantly, the ability to handle long text is also a crucial aspect of a language model’s capacity to retrieve information over time.** Through evaluations on the TIME-Dial dataset, we aim to demonstrate the model’s proficiency in accurately retrieving time-related information from long dialogues, comprehending event details, and integrating this information to reason about event temporal relationships.
>
> Additionally, **modern LLMs** like newest Qwen series, Deepseek-V3/R1 and recent OpenAI models **have almost already surpassed the context window of 100k tokens.** This is not a significant challenge for most models, and LLMs should be able to handle such long contexts and handle temporal reasoning tasks with contexts that average around 15.9k tokens. **What we used in our experiments are all with context window of over 32k tokens. Therefore, we believe the context-window length will not pose any impact on effective evaluation of temporal reasoning tasks in our paper.**

---

> > ### Comment · Reviewer_GXAP · 2025-08-06
> >
> > I appreciate the authors’ efforts in addressing my earlier concerns. The revisions have improved the manuscript; however, I still have a few remaining points for consideration:
> >
> > 1. **On temporal reasoning** – While the experiments are extensive, it remains unclear whether the LLM exhibits independent temporal reasoning issues. If such issues exist, the benchmark might benefit from a quantitative causal analysis that isolates them from other confounding factors (e.g., long-context attention degradation, retrieval errors across the full dataset).
> >
> > 2. **On annotation reliability** – The authors note that “we selected three annotators with at least an undergraduate degree and substantial experience in text annotation,” and conclude that single-round human annotation is reliable. In evaluation practice, this assumption can be risky. Clarifying whether cross-validation or inter-annotator agreement checks were conducted would strengthen the reliability claim.
> >
> > 3. **On the inclusion of agentic frameworks** – Many agentic frameworks have been applied to similar QA tasks, often to avoid directly injecting retrieved content into the context, and have shown higher accuracy in some cases. It might be worth explaining why such mainstream frameworks were not considered in the evaluation.
> >
> > Addressing these points could further improve the work. If they remain unresolved, my overall assessment would remain unchanged.

---

> ### Author Response · Authors · 2025-08-08
> **Response to Reviewer GXAP's Comments (on Point 1: Temporal Reasoning Independence)**
>
> We are grateful to the reviewer for the careful reassessment and constructive comments. We take the remaining concerns seriously. Below, we respond to each point with additional analysis and clarifications.
>
> > **Point 1: Clarifying Temporal Reasoning Independence**
>
> Thank you for your comments. We appreciate the reviewer’s attention to the context settings in temporal reasoning evaluation. **We would like to clarify that using long and complex texts as evaluation contexts is both realistic and essential for capturing the true challenges of temporal reasoning in real-world scenarios.** This perspective is supported by prior studies such as TReMu [1] and TCELongBench [2], which have demonstrated the necessity of incorporating long-range dependencies when evaluating temporal reasoning capabilities.
>
> **Temporal reasoning inherently requires the model to process temporally dispersed information. Hence, the ability to handle long contexts is not merely an auxiliary skill—it is an integral part of the task.** For instance, TReMu [1] evaluates neuro-symbolic temporal reasoning on extremely long documents (average context length ~14,509.91 tokens measured using `cl100k_base` tokenizer) and reveals that existing LLMs struggle with temporal reasoning under such memory demands. In this sense, long-context processing deficiencies are not orthogonal to the task but rather reflect a core limitation of current models. Our TIME-Dial benchmark is designed to uncover exactly such limitations.
>
> **To isolate and analyze temporal reasoning skills in the absence of long-context noise, we additionally use TIME-Wiki, whose context length is moderate** (average ~1,155.99 tokens). In this setting, all temporal facts required to answer the question are explicitly given in the context. Therefore, issues observed on TIME-Wiki can be attributed more directly to genuine reasoning failures rather than to retrieval or memory limitations.
>
> Based on the performance of models such as Qwen2.5-72B-Instruct, QwQ-32B and GPT-4o on TIME-Wiki and TIME-Lite-Wiki, we identified and categorized several key types of independent temporal reasoning issues:
>
>
> * **Issue 1: Misunderstanding temporal order.** Even with moderately sized contexts, models often fail to correctly infer the temporal order of events.
>
> * **Issue 2: Temporal conversion and arithmetic errors.** These commonsense mistakes—e.g., converting 1 year into 10 months—were also reported in TimeBench [3] and TRAM [4], and significantly affect *Duration Compare* and *Computation* tasks.
>
> * **Issue 3: Misinterpretation of ordinal expressions.** Phrases like "the x-th event" often confuse models, which fail to retrieve or align the correct event. Since TIME-Wiki’s contexts are moderate, such failures suggest deeper issues in temporal localization rather than memory.
>
> * **Issue 4: Misunderstanding relative temporal expressions.** Expressions such as "the most recent ....(event) after/before" often result in misinterpretation due to the model's inability to accurately recall all relevant candidate events for comparison.
>
> To further support our findings, we manually annotated a subset of TIME-Lite-News samples where retrieval was verified to be complete (i.e., the retrieved passage fully covers the gold answer), thereby eliminating the impact of retrieval errors. We analyzed the predictions of Qwen2.5-7B-Instruct and R1-Distill-Qwen-7B, and categorized independent temporal reasoning issues accordingly. The annotation was conducted by three experienced annotators, achieving a raw agreement score of 0.89. Final labels were determined by majority vote and discussion.
>
> ### Table: Proportion of issue types observed in Qwen2.5-7B-Instruct and DeepSeek-R1-Distill-Qwen-7B (on selected subset of TIME-Lite-News) (%).
>
> |Models|Issue 1|Issue 2|Issue 3|Issue 4|
> |:------|:------|:------|:------|:------|
> |Qwen2.5-7B-Instruct|19.1|15.73|11.24|8.99|
> |DeepSeek-R1-Distill-Qwen-7B|14.6|13.48|4.49|3.37|
>
> Lastly, we acknowledge that our causal analysis is preliminary due to time constraints. In future work, we plan to conduct more extensive experiments—such as perturbing retrieval results to observe changes in temporal reasoning—to better isolate underlying failure modes.
>
> ### References
> [1] Ge, Yubin, et al. "Tremu: Towards neuro-symbolic temporal reasoning for llm-agents with memory in multi-session dialogues." ACL 2025.
>
> [2] Zhang, Zhihan, et al. "Analyzing Temporal Complex Events with Large Language Models? A Benchmark towards Temporal, Long Context Understanding." ACL 2024.
>
> [3] Chu, Zheng, et al. "TimeBench: A Comprehensive Evaluation of Temporal Reasoning Abilities in Large Language Models." ACL 2024.
>
> [4] Wang, Yuqing, and Yun Zhao. "TRAM: Benchmarking Temporal Reasoning for Large Language Models." ACL 2024 Findings.

---

> ### Author Response · Authors · 2025-08-08
> **Response to Reviewer GXAP's Comments (on Point 2 and 3)**
>
> > **Point 2: Clarification on Annotation Reliability and Quality Assurance**
>
> We appreciate the reviewer’s concern regarding the annotation procedure. We would like to clarify that all manual annotations presented in the rebuttal were independently performed by multiple annotators. The final labels were determined through majority voting and, when necessary, discussion—not by a single annotator's decision.
>
> To further support the reliability of our analysis, we report inter-annotator agreement metrics computed from the original annotations discussed in the rebuttal. These metrics include both raw agreement (i.e., the percentage of items on which annotators fully agree) and Fleiss' Kappa, which accounts for the agreement occurring by chance. The results are summarized below:
>
> ### Table: Raw Agreement (%)
>
> |Retrieval Method(top-k=3)|Answer Included|Answer Correct|
> |:----------------------|:------------:|:------------:|
> |BM25|91.00|95.00|
> |Vector|89.00|96.00|
> |Hybrid|88.00|95.00|
>
>
> ### Table: Fleiss' Kappa
>
> |Retrieval Method(top-k=3)|Answer Included|Answer Correct|
> |:----------------------|:------------:|:------------:|
> |BM25|0.8634|0.9483|
> |Vector|0.8439|0.9573|
> |Hybrid|0.8397|0.9307|
>
>
> In these tables:
>
> * **Answer Included** refers to whether the retrieved documents fully cover the gold reference answer (annotated as True/False).
> * **Answer Correct** refers to whether the model’s response (GPT-4o in this case) is judged to be fully correct (True/False).
>
> We believe these results demonstrate the high reliability of our annotations and reinforce the robustness of our findings regarding model performance and retrieval quality.
>
>
> > **Point 3: Rationale for Excluding Agentic Frameworks**
>
> **We would like to clarify that the primary objective of our paper is to propose a benchmark specifically designed to evaluate the *intrinsic temporal reasoning capabilities of LLMs*, rather than to assess performance under *agentic frameworks*. Evaluating agent-based systems is beyond the intended scope of our work.**
>
> **While agentic frameworks may potentially improve temporal reasoning, they also introduce additional confounding factors—such as error propagation from multi-agent interactions—that make it difficult to isolate and attribute errors to the LLM's core reasoning abilities.** More importantly, **improvements observed in agentic settings do not directly inform how to improve the LLM’s *internal* reasoning capabilities.** As such, including agentic evaluations would be misaligned with the purpose of our benchmark, which aims to diagnose and improve fundamental temporal reasoning in LLMs.
>
> Prior related works such as TCELongBench \[1] and TimeBench \[2] have also focused on evaluating LLMs' temporal reasoning skills without considering agentic setups, further reinforcing the validity of our evaluation scope.
>
> We sincerely appreciate the reviewer’s insightful suggestion. **Although our evaluation primarily targets LLMs in isolation, we conducted preliminary experiments under an agentic framework within the limited rebuttal period.** Specifically, we adopted the ReAct framework \[3] and tested it on the TIME-News subset. Using LangChain with Ollama for local inference, we employed the `nomic-embed-text` model to generate embeddings and stored them in a Chroma vector database. For each query, the agent system retrieved the top-k=3 relevant documents and was guided by descriptions to invoke the retrieval tool. We set the agent's maximum number of iterations to 15.
>
> We experimented with several models, including Llama3.1-8B-Instruct and Qwen2.5-14B-Instruct, within this agentic setup. However, due to the large dataset size, limited time, and the need for extensive hyperparameter tuning in agentic workflows, we were not yet able to produce reliable and fair evaluation results.
>
> We would also greatly appreciate any recommendations on widely adopted agentic frameworks that could be used to evaluate temporal reasoning performance in such settings, as this could serve as a valuable complement to our LLM-focused evaluation. Where necessary and appropriate, we would be glad to cite these frameworks and incorporate their evaluation results.
>
> ### References
>
> [1] Zhang, Zhihan, et al. "Analyzing Temporal Complex Events with Large Language Models? A Benchmark towards Temporal, Long Context Understanding." ACL 2024.
>
> [2] Chu, Zheng, et al. "TimeBench: A Comprehensive Evaluation of Temporal Reasoning Abilities in Large Language Models." ACL 2024.
>
> [3] Yao, Shunyu, et al. "React: Synergizing reasoning and acting in language models." ICLR 2023.
>
> ---
>
> **We sincerely thank you again for the valuable feedback. We genuinely hope that our clarifications have addressed your concerns and helped highlight the value and scope of our contribution. We welcome further suggestions and appreciate your thoughtful evaluation of our work.**

---

> ### Author Response · Authors · 2025-08-09
> **Follow-Up on Reviewer GXAP‘s Feedback**
>
> We sincerely appreciate the time and effort you have dedicated to reviewing our work. In response to your valuable feedback, we have provided detailed explanations for the issues raised.
>
> **As the discussion deadline is now less than one day away**, we are eager to hear your thoughts on our responses, including whether they have addressed your concerns. We would be grateful if our explanations and updates could be taken into consideration for your overall assessment.
>
> We are committed to incorporating all of your suggestions to further enhance the quality of our manuscript. We look forward to your further comments and discussion.

---

### Official Review · Reviewer_mwqW · 2025-07-06

**Rating:** 5
**Confidence:** 3

**Summary:**

This paper introduces TimE, a comprehensive, multi-level benchmark designed to evaluate the temporal reasoning capabilities of LLMs in realistic scenarios. The authors highlight the limitations of existing benchmarks, which often neglect real-world complexities such as intensive temporal information, fast-changing event dynamics, and intricate temporal dependencies in social interactions.
The core contribution of this work is the TimE benchmark, comprising 38,522 question-answer (QA) pairs across three levels and eleven fine-grained sub-tasks. These levels progressively assess LLMs' abilities from basic temporal understanding and retrieval (Level-1), to temporal expression reasoning (Level-2), and finally to complex temporal relationship reasoning (Level-3).

**Dataset Code Accessibility:**

Yes

**Ethical Considerations:**

No, there are no or only very minor ethics concerns

**Limitations Weaknesses:**

1. **Simulated Environments:** While the benchmark aims for real-world scenarios, the authors acknowledge that the simulated environments might not fully capture all the intricacies of actual real-world situations.

2. **Static Data Source and Potential Data Leakage:** The use of a static data source (e.g., November 2024 Wikidata DB dump) for TimE-WIKI poses a potential risk of data leakage due to the continuous evolution of real-world knowledge. This could affect the long-term relevance and freshness of certain parts of the dataset.

3. **Decoding Strategy Constraints:** The exclusive use of greedy search decoding for all models, while ensuring fair comparison, might not fully reveal the temporal reasoning capabilities under other decoding strategies like random sampling, which could offer different insights.

**Strengths Contributions:**

1. **Comprehensive and Multi-level Design:** The TimE benchmark addresses a significant gap in temporal reasoning evaluation by providing a multi-level framework that systematically assesses capabilities across different granularities and real-world complexities.

2. *Diverse Real-World Scenarios:** The inclusion of TimE-WIKI, TimE-NEWS, and TimE-DIAL effectively captures a wide range of real-world temporal challenges, from static knowledge to dynamic events and interactive dialogues.

3. **High-Quality Data with Human Annotation:** The creation of TimE-LITE, a meticulously human-annotated and verified subset, significantly enhances the benchmark's reliability and facilitates efficient evaluation.

4. **Extensive Experimental Evaluation:** The paper presents a thorough evaluation across a wide range of LLMs (24 models).

---

> ### Author Rebuttal · Authors · 2025-07-31
>
> # Response to Reviewer mwqW
>
> We are grateful to the reviewer for the positive review and detailed suggestions, which have contributed to strengthening our manuscript. Below, we provide our responses to each comment.
>
> > **W1: Concerns about simulated environments in TIME benchmark**
>
> We would like to clarify that **the tasks in our TIME benchmark have been carefully designed, taking into account the most critical real-world challenges**, including: (1) the density of temporal information embedded within world knowledge, (2) the rapid evolution of event details over time, and (3) the complexity of temporal dependencies in social interactions. The TIME dataset evaluates the model’s temporal reasoning ability in real-world scenarios by providing a vast array of dynamic, time-related facts that humans encounter, all embedded in complex contexts.
>
> **Moreover, the primary objective of the TIME benchmark is not to simulate every conceivable real-world scenario in exhaustive detail, but rather to evaluate the model's ability to reason about time within the constraints of available evaluation resources**. Enumerating all possible real-world scenarios exceeds the scope of this benchmark.
>
> **Additionally, we will continue to develop the TIME benchmark and update future real-world evaluation scenarios.** Our goal is to collaboratively build the time reasoning evaluation environment with the open-source community, so as to better reflect models' real-world temporal reasoning capabilities.
>
>
> > **W2: Concerns about static data source and potential data leakage**
>
> Thank you for your comment. We completely agree on the importance of ensuring the freshness of the knowledge in the dataset. To address potential data leakage, we have implemented the following measures:
>
> First, during the dataset construction, we carefully considered the risks associated with static knowledge snapshots. **Specifically, when building TIME-Wiki, we filtered out knowledge triples involving future events or rapidly changing information** with the help of both manual curation and LLMs. **Furthermore, when generating questions for each sub-task, we used code scripts to limit the temporal scope of the knowledge being queried, ensuring that questions did not involve future events or dynamic knowledge.** These steps were taken to minimize the risk of model and data leakage from static knowledge.
>
> Additionally, **we are committed to regularly updating the benchmark knowledge at appropriate intervals and removing outdated question-answer pairs** to ensure the answers remain current. This approach will further mitigate concerns related to static knowledge and potential data leakage.
>
> > **W3: Concerns about decoding strategy constraints in evaluation.**
>
> **We have conducted additional experiments beyond the greedy search strategy in the main text to ensure comprehensive evaluation. We set the temperature to 0.6, top-p to 0.9 (nucleus sampling) for all models**. All experiments were repeated three times, and the results are averaged. The experiments were conducted on an NVIDIA A8000 80GB. The evaluation results for the three sub-datasets of TIME are as follow. Abbreviations are the same as in Table 2 in the main text.
>
> ### Table: Results for TIME-Wiki, temperature=0.6, top-p=0.9 (nucleus sampling)
> |Model|Ext.|Loc.|Comp.|DC.|OC.|ER.|OR.|RR.|Co-tmp.|TL.|CTF.|
> |:-----------------------------------|:------------|:----------|:----------|:----------|:----------|:----------|:----------|:----------|:----------|:----------|:----------|
> |Llama-3.1-8B-Instruct|30.49±0.96|71.37±0.3|9.44±0.6|48.18±0.9|67.23±0.24|35.32±0.29|31.66±0.31|24.7±0.25|30.28±0.12|0.33±0.07|24.21±0.21|
> |Qwen2.5-7B-Instruct|57.46±0.26|63.92±0.28|31.59±0.5|46.87±0.17|67.87±0.1|44.32±0.01|38.89±0.29|26.5±0.71|34.09±0.3|0.46±0.11|34.61±0.49|
> |Qwen2.5-14B-Instruct|70.72±2.21|67.2±0.04|34.79±0.48|63.37±0.14|77.48±0.35|51.39±0.3|39.86±0.09|31.51±0.33|33.31±0.17|4.23±0.38|39.9±0.73|
> |Qwen2.5-72B-Instruct|78.25±0.63|82.38±0.04|76.69±0.42|65.82±0.22|79.36±0.21|55.36±0.21|40.47±0.62|38.48±0.13|45.28±0.15|9.46±0.71|45.78±0.19|
> |Deepseek-R1-Distill-Llama-8B|65.47±0.47|70.76±0.24|67.95±0.23|80.43±0.34|88.99±0.45|52.69±0.24|39.46±0.27|31.45±0.23|32.23±0.25|4.79±0.54|35.2±0.39|
> |Deepseek-R1-Distill-Qwen-7B|56.6±0.23|58.42±0.93|49.67±0.48|75.85±0.46|83.45±0.28|35.95±0.26|26.56±0.38|26.53±0.34|26.42±0.31|0.92±0.22|31.5±0.33|
> |Deepseek-R1-Distill-Qwen-14B|74.23±0.61|63.55±0.67|51.56±0.32|84.85±0.06|92.4±0.19|56.24±0.56|42.99±0.32|34.14±0.17|35.85±0.23|14.26±0.79|42.17±0.45|
> |QwQ-32B|74.69±0.09|77.3±0.2|79.88±0.21|87.8±0.42|93.32±0.03|60.38±0.16|48.15±0.19|39.35±0.48|43.04±0.28|24.97±0.28|46.76±0.08|
>
> ### Table: Results for TIME-News, temperature=0.6, top-p=0.9 (nucleus sampling)
> |Model|Retriever|Loc.|Comp.|DC.|OC.|ER.|OR.|RR.|Co-tmp.|TL.|CTF.|
> |:-----------------------------------------|:-----------|:--------|:--------|:-------|:-------|:-------|:-------|:-------|:---------|:-------|:-------|
> |Llama-3.1-8B-Instruct|BM25|47.63±0.3|31.86±0.19|37.7±0.67|38.72±0.62|83.98±0.19|66.31±0.46|77.69±0.38|81.72±0.71|8.38±0.38|48.89±0.75|
> ||Vector|50.93±0.08|37.11±0.42|37.37±0.54|39.63±0.32|84.2±1.02|70.11±1.04|74.11±0.95|82.93±0.3|7.58±0.14|47.78±0.35|
> ||Hybrid|50.06±0.48|39.24±0.71|37.56±1.28|40.94±0.66|85.46±0.56|69.59±1.03|74.22±0.43|83.78±0.67|8.2±0.4|48.33±0.47|
> |Qwen2.5-14B-Instruct|BM25|69.19±0.05|71.58±0.54|42.78±0.11|48.19±0.08|87.67±0.56|66.11±0.11|75.17±0.67|82.78±0.06|20.82±0.09|55.5±0.17|
> ||Vector|69.71±0.05|69.62±0.07|41.58±0.64|47.78±0.56|87.53±0.19|65.36±0.19|76.56±0.78|81.97±0.25|17.83±0.37|54.83±0.44|
> ||Hybrid|69.05±0.01|68.96±0.57|42.64±0.92|47.53±0.03|87.69±0.47|66.22±0.33|77.75±0.64|82.67±0.33|20.21±0.41|54.92±0.86|
> |Deepseek-R1-Distill-Qwen-7B|BM25|55.03±0.36|66.83±0.23|42.43±0.28|51.69±0.43|76.28±0.44|60.35±1.29|71.3±0.16|73.81±1.1|17.53±0.4|37.61±0.67|
> ||Vector|57.23±0.27|67.54±0.8|41.52±0.84|55.07±1.34|77.52±0.2|62.15±0.68|73.11±0.63|75.72±0.39|17.71±0.6|38.54±0.2|
> ||Hybrid|58.08±0.75|66.52±0.39|41.5±0.45|52.59±1.05|76.5±0.56|60.91±0.36|73.0±0.61|74.43±0.73|17.0±0.27|37.54±0.22|
> |Deepseek-R1-Distill-Qwen-14B|BM25|64.42±0.25|62.53±0.46|42.14±0.42|56.83±1.22|82.81±0.58|68.61±0.17|79.67±0.11|83.69±0.19|21.53±0.03|65.17±0.06|
> ||Vector|67.46±0.11|63.93±0.93|42.17±0.44|56.25±0.14|83.53±0.31|69.11±0.17|81.47±0.58|83.31±0.14|18.87±0.35|65.33±1.28|
> ||Hybrid|66.89±0.44|67.54±0.54|41.17±0.0|59.25±0.14|83.97±0.03|69.83±0.33|81.94±0.22|83.86±0.19|19.96±0.24|63.5±0.06|
>
> ### Table: Results for TIME-Dial, temperature=0.6, top-p=0.9 (nucleus sampling)
> |Model|Ext.|Loc.|Comp.|DC.|OC.|ER.|OR.|RR.|Co-tmp.|TL.|CTF.|
> |:-------------------------------------|:---------|:---------|:---------|:---------|:---------|:---------|:---------|:---------|:---------|:---------|:---------|
> |Llama-3.1-8B-Instruct|34.04±2.89|29.57±0.97|16.95±0.2|37.41±0.69|40.52±2.16|34.22±2.49|28.15±1.29|34.07±1.11|42.3±0.73|0.0±0.0|24.15±2.1|
> |Qwen2.5-14B-Instruct|36.61±0.21|24.86±1.07|18.17±0.24|37.22±1.89|40.0±0.22|38.33±0.56|22.0±1.33|24.33±0.78|35.0±1.22|0.11±0.11|29.11±0.0|
> |Qwen2.5-32B-Instruct|37.55±0.49|29.44±0.43|24.74±1.21|36.44±0.96|45.85±0.38|41.26±0.21|35.7±0.55|33.93±0.69|38.74±0.84|0.52±0.21|32.59±0.46|
> |Deepseek-R1-Distill-Llama-8B|36.22±1.4|33.45±0.47|14.25±0.35|42.3±0.21|61.26±0.21|36.89±2.27|34.67±1.84|39.78±2.2|55.85±2.06|0.0±0.0|34.52±0.73|
> |Deepseek-R1-Distill-Qwen-14B|42.22±0.55|40.09±0.44|20.07±0.1|48.33±0.33|72.44±1.78|50.33±0.56|39.33±3.56|54.56±1.22|66.89±0.67|1.22±0.11|47.67±0.56|
> |Deepseek-R1-Distill-Qwen-32B|39.37±0.82|47.65±0.35|27.68±0.23|59.19±2.19|75.85±1.51|56.07±0.76|42.52±0.46|56.3±1.48|72.07±0.58|1.63±0.1|46.52±1.92|
>
> Based on the experiments conducted, the overall conclusion aligns with the one presented in the main text. However, in the case of TIME-Dial, the random sampling configuration shows a slight decrease in performance for most non-reasoning models across various tasks compared to greedy search. In contrast, the performance of reasoning models remains largely unchanged.
>
> **Additionally, we conducted experiments with the following configurations: (1) temperature=0.7, top-p=1.0, top-k=50 and (2) temperature=0.9, top-p=0.95, top-k=50.** These experiments yielded similar results to those presented earlier. However, due to space limitations, we are unable to include the details of these experiments in this rebuttal. In the future, we are committed to providing more random sampling configurations to more comprehensively reflect the models' real-world temporal reasoning capabilities.

---

> ### Author Response · Authors · 2025-08-09
> **Rebuttal Follow-Up**
>
> We sincerely thank you for the time and effort you have devoted to reviewing our work. In light of your valuable feedback, we have provided detailed responses and clarifications to each of the issues raised.
>
> **As the discussion deadline is now less than one day away**, we would greatly appreciate your thoughts on whether our revisions and explanations have sufficiently addressed your concerns.
>
> We remain fully committed to incorporating all of your suggestions to further improve the quality of our manuscript, and we look forward to your continued comments and discussion.

---

### Note · Authors · 2025-08-14

# Author Final Remarks

We sincerely thank the AC and all reviewers for their thoughtful feedback and constructive input during their rebuttal period. We appreciate the opportunity to improve our work. Below, we summarize the major improvements made and key reviewer concerns addressed.

---

## Major Rebuttal Discussions & Resolutions

1. **Improved Temporal Reasoning Analysis** (Reviewer GXAP):

   We manually annotated a subset of TIME-Lite-News to **isolate temporal reasoning issues** and conducted a statistical analysis of the key challenges.
2. **Clarified Key Differentiations of TIME Dataset** (Reviewer cXYQ):

   We **highlighted 6 key differences between TIME and existing datasets in a table** for clarity.
3. **Enhanced Event Interaction Analysis** (Reviewer cXYQ):

   We quantified event interaction complexity in TIME-News, emphasizing its complex temporal relationships. Further analysis will follow.
4. **Optimized Retrieval Analysis in TIME-News** (Reviewer GXAP):

   We tested retrieval strategies like top-k=5, noting slight improvements. **Errors were primarily due to temporal reasoning limitations rather than retrieval issues.**
5. **Ensured Freshness and Data Integrity** (Reviewer GXAP & Reviewer mwqW):

   We clarified **exclusions of future events and dynamic data** to prevent leakage and planned updates for freshness.
6. **Clarified Long-Context Impact** (Reviewer GXAP)

   We clarified that long-context ability is crucial for temporal reasoning. Our experiments confirmed temporal reasoning tasks can still be effectively evaluated.
7. **Multi-Sample Evaluation** (Reviewer mwqW):

   We explored **3 non-greedy decoding strategies** with temperature-based sampling **on the whole TIME dataset, repeating 3 times for robust results.**

---

## Reviewer-Specific Responses & Improvements

* Reviewer mwqW: Clarified data filtering methodology, updated dataset freshness, and new extensive experiments with extra decoding strategies.

* Reviewer GXAP: Addressed temporal reasoning independence, provided error analysis, and emphasized long-context processing's role in temporal reasoning with additional performance evidence.

* Reviewer cXYQ: Added a comparison table with previous datasets, clarified complex event interactions with statistical analysis, and committed to further analysis in the final version.

---

We believe these revisions significantly improve the clarity, rigor, and utility of **TIME**. Thank you again for your valuable feedback.

---

### Decision · Program_Chairs · 2025-09-18

**Decision:**

Accept (spotlight)

**Comment:**

**Summary**
The paper introduces TimE, a large-scale, multi-level benchmark designed to systematically evaluate temporal reasoning capabilities of LLMs across realistic domains, including static knowledge, dynamic news, and long dialogues. It consists of 38,522 QA pairs organized into three levels and eleven sub-tasks, with a human-verified subset (TIME-LITE) to ensure evaluation reliability. Experiments with 24 models highlight persistent challenges in complex temporal reasoning, such as timeline ordering and counterfactual reasoning.

**Strengths of the paper**

* Comprehensive multi-level framework that systematically isolates temporal reasoning skills from simple extraction to complex counterfactual reasoning.
* Coverage of diverse real-world scenarios through TIME-WIKI, TIME-NEWS, and TIME-DIAL, spanning static facts, evolving events, and long dialogues.
* Human-verified subset (TIME-LITE) with high annotation agreement ensures reliable lightweight evaluation.
* Thorough experimental evaluation across a wide range of LLMs, revealing clear insights into their temporal reasoning limitations.

**Weaknesses of the paper**

* Potential confounding factors such as retrieval quality (e.g., retriever choice significantly impacting results) may blur reasoning vs. evidence access.
* Reliance on a static knowledge snapshot (Wikidata dump) raises concerns of data leakage and obsolescence over time.
* Context-length limitations in TIME-DIAL make it difficult to disentangle temporal reasoning ability from model window constraints.
* Insufficient articulation of novelty relative to existing temporal reasoning datasets, especially regarding how complex multi-event interactions are uniquely captured.

**Major reason to accept**
The reviewers reached a consensus on the merits of the paper, with strengths being described above outweighing minor limitations.

**Summary of rebuttal and discussions**
The three reviews of the papers were initially positive. The authors' rebuttal addressed most of the reviewers' concerns, and the scores were further raised.